# Geometric Algorithms for Neural Combinatorial Optimization with Constraints

**Nikolaos Karalias**[*]
MIT
stalence@mit.edu

**Akbar Rafiey**[*]
NYU
ar9530@nyu.edu

**Yifei Xu**
NYU
yx3590@nyu.edu

**Zhishang Luo**
UCSD
zluo@ucsd.edu

**Behrooz Tahmasebi**
MIT
bzt@mit.edu

**Connie Jiang**
MIT
conniej@mit.edu

**Stefanie Jegelka**
TUM and MIT
stefje@mit.edu

## Abstract

Self-Supervised Learning (SSL) for Combinatorial Optimization (CO) is an emerging paradigm for solving combinatorial problems using neural networks. In this paper, we address a central challenge of SSL for CO: solving problems with discrete constraints. We design an end-to-end differentiable framework that enables us to solve discrete constrained optimization problems with neural networks. Concretely, we leverage algorithmic techniques from the literature on convex geometry and Carathéodory's theorem to decompose neural network outputs into convex combinations of polytope corners that correspond to feasible sets. This decomposition-based approach enables self-supervised training but also ensures efficient quality-preserving rounding of the neural net output into feasible solutions. Extensive experiments in cardinality-constrained optimization show that our approach can consistently outperform neural baselines. We further provide worked-out examples of how our method can be applied beyond cardinality-constrained problems to a diverse set of combinatorial optimization tasks, including finding independent sets in graphs, and solving matroid-constrained problems.

## 1 Introduction

Combinatorial Optimization (CO) encompasses a broad category of optimization problems where the objective is to find a configuration of discrete objects that satisfies specific constraints and is optimal given a prescribed criterion. It has a wide range of real-world applications [64, 58, 31, 7, 94, 73] while also being of central importance in complexity theory and algorithm design. These problems are often non-convex, and solving them involves exploring large-scale discrete combinatorial spaces of configurations. To that end, neural networks have emerged as a powerful tool for designing CO algorithms as they can learn to exploit patterns in real-world data and discover high-quality solutions efficiently. A compelling proposal in that direction is self-supervised CO because it eschews the need to acquire labeled data, which can be computationally expensive for those problems. This usually involves training a neural network to produce solutions to input instances by minimizing a continuous proxy for the discrete objective of the problem [35, 84, 1, 36, 90]. While the efficiency and scalability of this approach are appealing, it also harbors significant challenges.

Enforcing constraints on the output of the neural network is one of the primary obstacles. When training, this is often done by including a weighted term in the loss function that penalizes solutions that do not comply with the constraints. There are two key considerations when doing this. Having

---

[*]Equal contribution.

39th Conference on Neural Information Processing Systems (NeurIPS 2025).

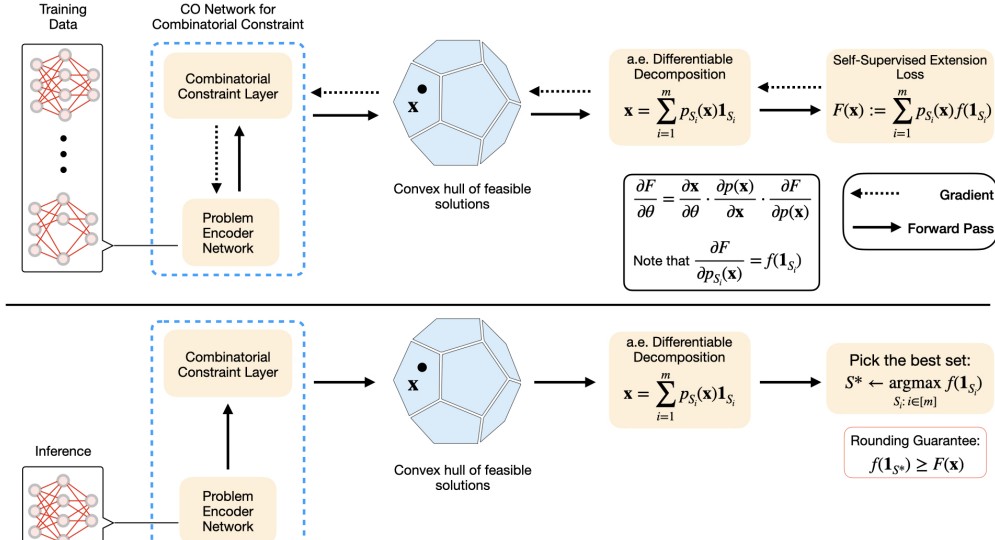

Figure 1: Overview of our framework. During training (top), the model learns to output a point in the convex hull of feasible solutions. A self-supervised extension loss is computed by decomposing this point and evaluating the objective, which is then used to backpropagate. During inference (bottom), the model outputs a relaxed solution, which is decomposed and the best feasible set is selected with a rounding guarantee.

access to a differentiable function that encodes constraint violation for the discrete problem, and carefully tuning its contribution to the overall loss. Furthermore, at inference, it is important to guarantee that the output of the neural network is discrete and it complies with the constraints. In practice, this is usually achieved with an algorithm that rounds the continuous output of the neural network to a feasible discrete solution. Such algorithms are often heuristics and treat inference differently from training which may hurt performance. Despite recent developments that have enabled tackling new combinatorial problems in the self-supervised setting, combining loss function design and rounding in a coherent fashion can still be highly nontrivial.

In this work, building on ideas from the literature on self-supervised CO and techniques from geometric algorithms and combinatorial optimization, we propose an approach to loss function design that seamlessly integrates training and inference while also providing a rounding guarantee. The main idea is to *learn to map input instances to distributions of feasible solutions* that maximize (or minimize resp.) the objective. Given an input instance, we use the neural network to predict a continuous vector in the convex hull of feasible solutions. Using an iterative algorithm, this vector is decomposed into a distribution of discrete feasible sets. This algorithm draws from the literature on Carathéodory's theorem [13] to express points in the interior of a convex polytope as a sparse convex combination of the corners of the polytope. This decomposition yields a distribution that allows us to tractably calculate the expected value of the discrete objective. We use this expectation as our loss and minimize it with a neural net and standard automatic differentiation in a self-supervised fashion. By backpropagating derivatives from this expectation, the neural network learns to generate outputs in the feasible set that optimize the expected objective. At inference time, the iterative algorithm generates candidate feasible solutions to the problem. The pipeline is summarized in Figure 1.

**Our contribution.** Our method generalizes previous work, and offers strong empirical benefits in efficiency and performance on large-scale CO instances. Our contributions can be summarized as follows:

- We use a class of geometric decomposition algorithms to design end-to-end learning pipelines for constrained neural CO. Our main result establishes the applicability of our method to any problem whose feasible set polytope admits a fast linear optimization oracle.

- Our geometric algorithms effectively tackle both the challenge of loss function design for training with constraints, and the challenge of rounding neural network outputs at inference.

- We show strong results in cardinality constrained CO on large-scale instances with hundreds of thousands of nodes, often surpassing previous neural baselines and heuristics.

- We conduct ablation studies to test the effectiveness of optimizing the extension with gradient descent and the contribution of the model in the overall performance.

## 2 Related work

**Self-Supervised CO and continuous extensions.** Our work follows a long line of work on unsupervised CO [1, 2, 35, 84, 36, 82, 87, 30, 72, 24, 63, 78, 43, 60, 90]. Crucial components of those pipelines include the choice of model, the use of specific input features (positional encodings), the loss function, and the rounding algorithm. There has been extensive research on neural net architectures for combinatorial problems [57, 85, 74, 93] and the role of input features for well known classes of models [91, 53, 38]. Our primary focus will be on loss function design and rounding. We build on previous work that proposed using continuous extensions of discrete functions as losses for neural CO [36]. Following this blueprint, bespoke extensions have been used for learning on permutations [62], hierarchical clustering [43] and interpretable graph learning [16]. However, there is no general extension design paradigm that can accommodate different constraints. We address this issue by proposing a template for extension design and show how it enables building the constraints into the distribution. This streamlines the process of building extensions and dispenses with the need to tune constraint terms in the loss function that is present in other approaches to self-supervised CO.

**Loss function design and enforcing constraints.** The design of specific loss functions in self-supervised CO to enable smoother optimization, integration with different powerful architectures, and/or better rounding has been studied over the past few years. A common approach is to adopt a probabilistic perspective and parametrize a distribution of outputs with a neural network that learns to optimize the expected cost and the probability of constraint violation [60, 87, 36, 72, 62]. Deriving and/or computing those expectations and probability terms for more complex problems can be difficult, which motivated the work by [10] that provides derivations and fast rounding techniques for cases such as spanning trees and cardinality constraints. In our work, the constraints are built into the distribution and the loss only has to handle the objective function. It is challenging to craft such losses which has led to the development of techniques that improve training dynamics via annealed training [82, 33]. Other approaches to loss function design leverage ideas from physics [78, 79], and semidefinite programming [93, 32]. Those cases also involve tunable constraint terms. Another major consideration is that of enforcing constraints on the outputs of the neural networks. Techniques in the literature typically involve calculating the projection of the neural net output onto the feasible set. This has been done for cardinality constraints using ideas from the theory of optimal transport [89], and general linear constraints with gradient-based techniques [18, 95]. In the continuous (potentially non-convex) optimization setting, several projection techniques have also been developed [51, 52, 50, 48].

**Convex geometry and learning over polytopes.** Our approach relies on fundamental results from convex geometry and geometric algorithms. Specifically, the Carathéodory Theorem states that any point in a polytope $P \subseteq \mathbb{R}^d$ can be expressed as a convex combination of at most $d + 1$ points. The constructive versions of this result [25] have been adapted to yield efficient algorithms in polytopes for ranking [42] and scheduling problems [29, 21, 8]. Approximate versions of the decomposition have been developed that do not explicitly depend on the ambient dimension $d$ [4] and admit efficient algorithms with fast convergence rates [59, 17]. In the context of machine learning, polyhedral methods have been used extensively to impose constraints on the outputs of models for generative modelling [22], for learning with end-to-end differentiable sampling from exponential families using perturb-and-map techniques [67], for applications to differentiable sorting [6, 71] and learning to optimize over permutations [26, 56, 62]. For additional discussion, please refer to the appendix.

## 3 Problem formulation and learning setup

Let $V = \{v_1, \ldots, v_n\}$ be a set of variables, each assigned a value from a discrete domain $D = \{1, \ldots, d\}$. A combinatorial optimization problem consists of a universe of instances $\mathcal{T}$, where each instance $T \in \mathcal{T}$ can be represented by a tuple $T = (V, D, \mathcal{C}, f)$. The set $\mathcal{C} \subseteq D^n$ defines the feasible solutions to the instance. The real-valued function $f : D^n \to \mathbb{R}$ is called the objective function.

The problem is then to find an optimal feasible vector $\mathbf{x}^*$, i.e., to find a feasible vector $\mathbf{x} \in D^n$ that maximizes the objective function $f$:

$$\max f(\mathbf{x}), \quad \text{s.t.} \quad \mathbf{x} \in \mathcal{C}. \tag{1}$$

In this paper, we focus on CO problems with Boolean domain, i.e., $D = \{0, 1\}$, and $\mathcal{C} \subseteq \{0, 1\}^n$. This domain choice corresponds to combinatorial optimization problems that involve set functions, i.e., $\mathbf{x}$ indicates a subset $S \subseteq V$ of a ground set $V = \{v_1, \ldots, v_n\}$ of elements and $x_i = 1$ if element $i$ is in the subset, and $f$ maps each subset to a real number. The subsets can be, for instance, nodes or edges in a graph, and are central to many problems in applied mathematics and operations research [9]. The set $\mathcal{C} \subseteq \{0, 1\}^n$ includes all subsets of $V$ that obey the constraints of the problem. Each subset $S$ is identified with its indicator vector $\mathbf{1}_S \in \{0, 1\}^n$ and we use $f(S)$ and $f(\mathbf{1}_S)$ interchangeably throughout this paper. The objective function is the set function $f : \{0, 1\}^n \to \mathbb{R}$. The goal is to select an optimal feasible subset so Equation (1) takes the form $\max_{S \in \mathcal{C}} f(S)$.

**Background on polyhedral geometry.** Define the encoding length of a number as the length of the binary string representing the number. The encoding length of an inequality $\mathbf{a}^\top \mathbf{x} \leq \mathbf{b}$, is the sum of encoding lengths of the entries of the vectors $\mathbf{a}$ and $\mathbf{b}$. Let $\mathcal{P} \subseteq \mathbb{R}^n$ be a polyhedron and let $\varphi$ be a positive integer. We say that $\mathcal{P}$ has *facet–complexity* at most $\varphi$ if there exists a system of linear inequalities with rational coefficients whose solution set is $\mathcal{P}$, and such that the encoding length of each inequality in the system is at most $\varphi$. A *well–described polytope* is a triple $(\mathcal{P}; n, \varphi)$ where $\mathcal{P} \subseteq \mathbb{R}^n$ is a polytope with facet–complexity at most $\varphi$. The *strong optimization problem* for a given rational vector $\mathbf{x} \in \mathbb{Q}^n$ and any well–described $(\mathcal{P}; n, \varphi)$ is

$$\max_{\mathbf{c} \in \mathcal{P}} \mathbf{c}^\top \mathbf{x}. \tag{2}$$

A strong optimization oracle returns a maximizer $\mathbf{c}^*$ for (2). Polytopes can be described as convex hulls of finitely many vectors or as systems of linear inequalities. Even though a description may require exponentially many vectors or inequalities, the polytope may admit a fast optimization oracle. This is the case for submodular polyhedra [3] or the matching polytope [77, Chapters 25, 26].

**Self-Supervised CO.** Given a problem instance $T$ and input features $\mathbf{Z}_T \in \mathbb{R}^{n \times d}$, we use a neural network $\text{NN}_\theta$, which we will often refer to as the encoder network, to map the instance data to an output prediction $\mathbf{x} \in \mathbb{R}^n$ by computing $\mathbf{x} = \text{NN}_\theta(\mathbf{Z}_T; T)$. For example, the input instance could be a graph, the input features could be positional encodings for the graph, and the neural net a Multi-Layer Perceptron (MLP). The output $\mathbf{x}$ may not correspond to a discrete feasible point. In those cases, a rounding algorithm will be used to map $\mathbf{x}$ to a feasible indicator vector of a set. The goal is for the neural net to learn to predict the optimal solution $\mathbf{x}^*$. In self-supervised CO, this is done by training $\text{NN}_\theta$ on a collection of instances $T_1, T_2, \ldots, T_m$. The neural net minimizes the problem-specific loss function $\mathcal{L}_\mathcal{T}$ computed for each instance $\mathcal{L}_\mathcal{T}(\text{NN}_\theta(\mathbf{Z}_{T_i}; T_i))$ and averaged over a batch (or the entire training set). Since there are no labels, how the loss function is designed to fit the constraints of the problem is of critical importance. At inference time, a test instance is processed through the neural network to obtain a prediction. At this step, rounding heuristics are typically used to ensure that the solution is feasible and discrete. It should be noted that, it is possible to directly optimize the model with gradient descent on the test instances since no labels are leveraged during training. In this case, the additional computational cost of backpropagation is incurred at inference time.

## 4 Proposed approach

First, we present an overview of the key components in the pipeline and then we discuss each component in more detail. Our proposed method focuses on the design of $\mathcal{L}_\mathcal{T}(\mathbf{x})$ that is used to train the model. The objective $f$ is a set function with discrete domain and cannot be directly optimized. Following the literature, we construct a continuous extension of $f$, denoted by $F(\mathbf{x}) : [0, 1]^n \to \mathbb{R}$, and use it as our loss function, i.e. $\mathcal{L}_\mathcal{T}(\mathbf{x}) = F(\mathbf{x})$. The extension is defined as

$$F(\mathbf{x}) := \mathbb{E}_{S \sim \mathcal{D}_\mathcal{C}(\mathbf{x})}[f(S)], \tag{3}$$

where $\mathcal{D}_\mathcal{C}(\mathbf{x})$ is a distribution parametrized by the output of the neural network and $S \in \mathcal{C}$ for all $S$ in the support of $\mathcal{D}_\mathcal{C}(\mathbf{x})$. The key challenge lies in how to tractably construct and parameterize such a distribution on feasible sets in a way that enables end-to-end learning. We achieve this using an

iterative algorithm that ensures $\mathcal{D}_{\mathcal{C}}(\mathbf{x})$ is supported on $O(n)$ feasible sets and the probability of each set is an a.e. differentiable function of $\mathbf{x}$. This allows us to efficiently compute the exact expectation in Equation (3) and to calculate its derivatives with standard automatic differentiation packages. At inference time, the geometric algorithm can be used as a rounding algorithm for $\mathbf{x}$ since $\mathcal{D}_{\mathcal{C}}(\mathbf{x})$ has a small number of sets in its support. Specifically, we can round $\mathbf{x}$ to the best set in the support and obtain a feasible integral solution whose quality is at least as good as $F(\mathbf{x})$.

## 4.1 A general decomposition algorithm for extension design

For a feasible set $\mathcal{C}$, consider the polytope $\mathcal{P}(\mathcal{C}) = \text{Conv}(\{\mathbf{1}_S : S \in \mathcal{C}\})$, i.e., the convex hull of the indicator vectors of the sets in $\mathcal{C}$. Any point $\mathbf{x} \in \mathcal{P}(\mathcal{C})$ is a convex combination $\mathbf{x} = \sum_{S \in \mathcal{C}} \alpha_S \mathbf{1}_S$ of feasible solutions from $\mathcal{C}$. In fact, from Carathéodory's theorem we know that this convex combination requires at most $n + 1$ points where $n$ is the ambient dimension of the space. The coefficients of the convex combination can be viewed as the probabilities of a distribution $\mathcal{D}_{\mathcal{C}}(\mathbf{x})$ and hence $\mathbf{x} = \mathbb{E}_{S \sim \mathcal{D}_{\mathcal{C}}(\mathbf{x})}[\mathbf{1}_S]$. If we can *uniquely* and *efficiently* decompose each point $\mathbf{x} \in \mathcal{P}(\mathcal{C})$ into such a convex combination, we can leverage that distribution to compute the extension in Equation (3). In Algorithm 1, we propose a general template for an iterative algorithm that yields such decompositions and forms the foundation for our approach.

Intuitively, the algorithm decomposes $\mathbf{x}$ iteratively by removing a corner $\mathbf{1}_S$ weighted by a suitably chosen coefficient in each iteration. This is done until the coefficients and the corresponding corners can be used to reconstruct $\mathbf{x}$ up to some small error $\epsilon$.

Let $\mathbf{x} \in \mathcal{P}(\mathcal{C})$ and $\mathbf{x}_0 = \mathbf{x}$. In each iteration $t$, the algorithm finds a feasible solution in $\mathcal{C}$ that is both an extremal point of the convex hull and lies in the support of $\mathbf{x}_t$. Here by the support of $\mathbf{x}_t$ we mean $supp(\mathbf{x}_t) := \{S \in \mathcal{C} : \mathbf{x}_t(i) \neq 0 \ \forall i \in S\}$, that is all the feasible subsets $S \in \mathcal{C}$ such that every corresponding entry of $\mathbf{x}_t$ indexed by $i \in S$ is nonzero. Let this point be $\mathbf{1}_{S_t} \in supp(\mathbf{x}_t) \subseteq \{0,1\}^n$. Let $a_t \in [0,1]$ be the coefficient at iteration $t$. We express each iterate $\mathbf{x}_t$ as

$$\mathbf{x}_t = a_t \mathbf{1}_{S_t} + (1 - a_t)\mathbf{x}_{t+1}, \qquad (4)$$

**Algorithm 1** General decomposition algorithm
**Require:** $\mathbf{x} \in [0,1]^n$ in $\mathcal{P}(\mathcal{C})$.
1: $\mathbf{x}_0 \leftarrow \mathbf{x}, t \leftarrow 0$.
2: **repeat**
3: $\quad \mathbf{1}_{S_t} \leftarrow$ vertex from $\mathcal{P}(\mathcal{C})$ in the support of $\mathbf{x}_t$.
4: $\quad a_t \leftarrow g(\mathbf{x}_t, \mathbf{1}_{S_t})$
5: $\quad \mathbf{x}_{t+1} \leftarrow (\mathbf{x}_t - a_t \mathbf{1}_{S_t})/(1 - a_t)$
6: $\quad p_{\mathbf{x}_t}(S_t) \leftarrow a_t \prod_{i=0}^{t-1}(1 - a_i)$
7: $\quad t \leftarrow t + 1$
8: **until** $\|\mathbf{x} - \sum_t p_{\mathbf{x}_t}(S_t)\mathbf{1}_{S_t}\| \leq \epsilon$
9: **return** All $\{p_{\mathbf{x}_t}(S_t), \mathbf{1}_{S_t})\}$ pairs.

where $a_t = g(\mathbf{x}_t, \mathbf{1}_{S_t})$ is a function of $\mathbf{x}_t$ and $\mathbf{1}_{S_t}$, chosen such that $\mathbf{x}_{t+1}$ remains in $\mathcal{P}(\mathcal{C})$. We can recursively apply the same process to $\mathbf{x}_{t+1}$ until $\mathbf{x}$ has been decomposed into a convex combination of feasible sets. After the algorithm terminates, we can obtain a distribution $\mathcal{D}_{\mathcal{C}}(\mathbf{x})$ which is completely characterized by the sets $S_t$ and probabilities $p_{\mathbf{x}_t}(S_t) = a_t \prod_{i=0}^{t-1}(1 - a_i)$ of the decomposition. Determining the specific form of steps 3 and 4 in a way that yields a polynomial-time algorithm that is usable in an end-to-end learning setting is a non-trivial task that depends on the polytope. We prove the following theorem which relies on a constructive version of the Carathéodory theorem, also known in the literature as the GLS method (from Grötschel-Lovász-Schrijver) [29, 42]. The GLS method establishes a decomposition for a large class of polytopes. Our result builds on top of the GLS method to show how this decomposition is also almost everywhere differentiable with respect to $\mathbf{x}$.

**Theorem 4.1.** *There exists a polynomial-time algorithm that for any well-described polytope $\mathcal{P}$ given by a strong optimization oracle, for any rational vector $\mathbf{x}$, finds vertex-probability pairs $\{p_{\mathbf{x}_t}(S_t), \mathbf{1}_{S_t}\}$ for $t = 0, 1, \ldots, n-1$ such that $\mathbf{x} = \sum_{t=0}^{n-1} p_{\mathbf{x}_t}(S_t)\mathbf{1}_{S_t}$ and all $p_{\mathbf{x}_t}(S_t)$ are almost everywhere differentiable functions of $\mathbf{x}$.*

Intuitively, at each iteration the decomposition algorithm leverages the strong optimization oracle to obtain a vertex for a minimal face of the polytope that contains the current iterate. This corresponds to step 3 of our algorithm. At step 4, the largest possible $a_t$ is chosen that yields a new iterate that remains in the polytope. This means that $\mathbf{x}_{t+1}$ will intersect the boundary of the polytope. At each iteration, the iterate lies on a lower-dimensional face of the polytope than the previous one, and hence the algorithm terminates in polynomial time, in at most $n + 1$ iterations.

**End-to-end learning with the decomposition.** Drawing insights from the GLS method, we aim to design a decomposition that can be integrated in gradient-based optimization pipelines. Here,

we summarize the basic requirements for the general decomposition to yield an extension that can be optimized with standard automatic differentiation, which will enable end-to-end learning. To obtain consistent tie breaks at the boundaries for the strong optimization oracle, we adopt a standard lexicographic ordering of the coordinates. This also ensures that the steps of the decomposition are deterministic and that the coefficients of the decomposition are uniquely determined for each point in the polytope. For our approach to be useful for end-to-end learning, we require:

1. A differentiable method that ensures our starting point $\mathbf{x}$ lies in the polytope $\mathcal{P}(\mathcal{C})$.

2. A routine that efficiently finds a vertex $\mathbf{1}_{S_t} \in \mathcal{P}(\mathcal{C})$ in the support of $\mathbf{x}_t$.

3. An (almost everywhere) differentiable function $g(\mathbf{x}_t, \mathbf{1}_{S_t})$ that yields a valid decomposition.

Satisfying these conditions will allow us to tractably construct a distribution of feasible sets $\mathcal{D}_{\mathcal{C}}(\mathbf{x})$, which will enable efficient calculation of the expectation in Equation (3). It also means that we can update the parameters of the neural network that is used to predict $\mathbf{x}$ by optimizing the extension via standard automatic differentiation packages. Our proof of Theorem 4.1 establishes that a strong optimization oracle is sufficient to handle steps 2 and 3. See Appendix B.1 and Appendix B.2 for detailed comments on the applicability of the method.

**Practical considerations for step 3 and 4.** Our GLS-based construction offers a canonical template that can instantiate Algorithm 1, but deviating from GLS can confer additional benefits. For example, $g$ in Step 4 can be chosen so that the next iterate does not intersect a lower dimensional face of the polytope. The difference now is that, instead of choosing the maximum coefficient possible, we will rescale it in each step by some fixed constant $b \in (0, 1]$. Termination of the algorithm is ensured by the parameter $\epsilon$ in step 8 of Algorithm 1 and a lower bound $l$ on the minimum value of the coefficient.

**Proposition 4.2.** *Let*

$$\tilde{a}_t = \begin{cases} b \ a_t, & b \ a_t \geq \ell, \\ a_t, & b \ a_t < \ell \end{cases} \tag{5}$$

*be the coefficient we pick for step 3 in Algorithm 1. Then the reconstruction error of step 8 Algorithm 1 decays exponentially in $k$.*

This approach allows for controlled approximation of $\mathbf{x}$ and the introduction of more vertices into the decomposition which can help exploration when optimizing the extension.

**Neural predictions in the feasible set polytope.** Suppose our encoder network outputs $\mathbf{x} \in \mathbb{R}^n$. Since Algorithm 1 requires a point in the polytope $\mathbf{x} \in \mathcal{P}(\mathcal{C})$, we need a way to enforce this constraint on the output of our neural network. This type of problem has received significant attention (see Section 2) in both the optimization and machine learning communities. Commonly, this can involve projection-based techniques such as Sinkhorn's algorithm [80] and its various extensions [88] or other gradient-based approaches [95]. This can be a viable strategy but, depending on the polytope, may be difficult to integrate in a differentiable pipeline. We will leverage a projection-based approach when appropriate but we also introduce a simple efficient alternative. We interpret the neural encoder output as a perturbation of an interior point in $\mathcal{P}(\mathcal{C})$. More concretely, the neural net predicts a perturbation vector $\mathbf{z} \in \mathbb{R}^n$ such that $\mathbf{x} = \mathbf{z} + \mathbf{u}$, where $\mathbf{u} \in \mathbb{R}^n$ is an interior point of $\mathcal{P}(\mathcal{C})$. While this approach circumvents the need for projection, it requires access to an interior point of the polytope. It also requires care so that the perturbed point $\mathbf{z} + \mathbf{u}$ remains in the polytope. Nonetheless, we show that for several fundamental constraint families–including cardinality constraints, partition matroids, and the spanning tree base polytope–it is possible to do this efficiently in a differentiable way (e.g., Proposition 4.4).

**Extension function, training, and inference.** Once the decomposition algorithm provides $\mathcal{D}_{\mathcal{C}}(\mathbf{x})$, we can use it to compute our loss function. For the extension to be well defined, the decomposition needs to deterministically map each point in the polytope to vertex-probability pairs. This is guaranteed in our constructions. It is also straightforward to compute derivatives of the extension with respect to the neural network parameters. For simplicity, suppose the encoder network, parameterized by weights $\theta$, outputs $\mathbf{x} \in \mathcal{P}(\mathcal{C})$. Then, we have $\frac{\partial F}{\partial \theta} = \frac{\partial F}{\partial p(\mathbf{x})} \cdot \frac{\partial p(\mathbf{x})}{\partial \mathbf{x}} \cdot \frac{\partial \mathbf{x}}{\partial \theta}$ where $p(\mathbf{x})$ is the vector of probabilities returned by Algorithm 1.

The extension $F(\mathbf{x}) = \mathbb{E}_{S \sim \mathcal{D}_\mathcal{C}(\mathbf{x})}[f(S)] = \sum_t p_{\mathbf{x}_t}(S_t)f(S_t)$ has several desirable properties that have been studied in the literature [36]. Since we are considering convex combinations of the objective over the feasible set, the extreme points are preserved i.e., $\max_{\mathbf{x} \in \mathcal{P}(\mathcal{C})} F(\mathbf{x}) = \max_{S \in \mathcal{C}} f(S)$; moreover we have $\operatorname{argmax}_{\mathbf{x} \in \mathcal{P}(\mathcal{C})} F(\mathbf{x}) \in \operatorname{Conv}(\{\mathbf{1}_S : S \in \operatorname{argmax}_{S \in \mathcal{C}} f(S)\})$. The expectation formulation also provides a rounding guarantee. There exists at least one feasible solution $S^*$ in the decomposition of $\mathbf{x}$ such that $f(S^*) \geq F(\mathbf{x})$. This property enables fast inference on the output of the encoder network without any degradation in solution quality.

## 4.2 Case studies

In this section we discuss how the decomposition can be generated for various constraints. We provide a detailed treatment of the cardinality constraint to build intuition and then discuss additional examples including spanning trees and independent sets. All proofs, additional details and additional experiments can be found in the Appendix.

### 4.2.1 Cardinality constraint

Here, we focus on combinatorial problems that involve optimizing a set function under an exact cardinality constraint. This setting is sufficiently general to encompass many fundamental and practical combinatorial optimization problems. Formally, let $f : 2^V \rightarrow \mathbb{R}$ be a set function that assigns a real value to each subset of $V = \{v_1, \ldots, v_n\}$. For a given positive integer $k$, the set of feasible solutions is defined as $\mathcal{C} = \{S : |S| = k, S \subseteq V\}$, which includes all subsets of $V$ of size exactly $k$. The resulting constrained combinatorial optimization problem is:

$$\max_{\mathbf{x} \in \{0,1\}^n, \|\mathbf{x}\|_1 = k} f(\mathbf{x}). \tag{6}$$

The associated convex polytope, $\mathcal{P}(\mathcal{C}) = \operatorname{Conv}(\{\mathbf{1}_S : |S| = k\})$, is the convex hull of Boolean vectors with exactly $k$ ones and $n - k$ zeros and the *matroid base polytope* corresponding to a uniform matroid on $n$ elements of rank $k$. It is known as the (k,n)-hypersimplex and is denoted by $\Delta_{n,k}$ [70].

**Designing the decomposition algorithm.** For now, we assume that our predictions lie in the polytope of the feasible set, i.e. $\mathbf{x} \in \Delta_{n,k}$. To construct a decomposition, we need to ensure that steps 3 and 4 in Algorithm 1 comply with our conditions for efficient end-to-end learning. To obtain a vertex of the polytope in the support of $\mathbf{x}_t$ for Step 3 in Algorithm 1, we set

$$\mathbf{1}_{S_t} = \operatorname*{argmax}_{\mathbf{c} \in \{0,1\}^n, \|\mathbf{c}\|_1 = k} \mathbf{x}_t^\top \mathbf{c}. \tag{7}$$

For step 4, we use Equation (4) to calculate the largest possible coefficient that keeps the iterate in the polytope. This yields

$$a_t = \min \left\{ \min_{i \in S_t} \mathbf{x}_t(i), 1 - \max_{i \notin S_t} \mathbf{x}_t(i) \right\}. \tag{8}$$

The following result shows that our decomposition algorithm produces a valid convex decomposition of points in the hypersimplex and is suitable for end-to-end learning.

**Theorem 4.3.** *For $\mathbf{x} \in \Delta_{n,k}$, Algorithm 1 with Step 3 according to Equation (7) and Step 4 according to Equation (8), terminates in at most $n$ iterations and returns probability-vertex pairs $\{(p_{\mathbf{x}_t}(S_t), \mathbf{1}_{S_t})\}$ such that every $S_t$ is a set of size $k$ and $\mathbf{x} = \sum_t p_{\mathbf{x}_t}(S_t)\mathbf{1}_{S_t}$. Moreover, each $p_{\mathbf{x}_t}(S_t)$ is an a.e. differentiable function of $\mathbf{x}$.*

At a high level, the result follows a similar strategy as Theorem 4.1. Our choice of coefficient ensures that the next iterate intersects the boundary of the polytope and lies in a lower-dimensional face. This in turn guarantees that the algorithm terminates in at most $n + 1$ steps.

**Generating neural predictions in the hypersimplex.** The last concern to address is that of obtaining a vector $\mathbf{x} \in \Delta_{n,k}$ in an efficient and differentiable manner. We follow the perturbation strategy that we outlined earlier and use the neural network to predict a perturbation vector $\mathbf{z} \in [0, 1]^n$ for an interior point. The following result ensures that the perturbed point $\mathbf{x}$ remains in the polytope.

**Proposition 4.4.** *Let $\mathbf{z} \in [0, 1]^n$ and $\mathbf{u} = [\frac{k}{n}, \frac{k}{n}, \ldots, \frac{k}{n}]$. Let $\mu$ be the mean of $\mathbf{z}$, and $\tilde{\mathbf{z}}$ be the mean-centered version of $\mathbf{z}$. For $s = \min(\frac{k/n}{\mu}, \frac{(n-k)/n}{1-\mu})$ and $\mathbf{x} = s\tilde{\mathbf{z}} + \mathbf{u}$, we have that $\mathbf{x} \in \Delta_{n,k}$.*

This simple transformation is efficient to compute and maintains differentiability of $\mathbf{x}$ with respect to the neural network parameters.

### 4.2.2 Decomposition for Partition Matroids, Spanning Trees, and Independent Sets

**Partition Matroids.** We extend our result to general constraints beyond cardinality. Let $V_1, \ldots, V_c$ be disjoint subsets with $V = \cup_{i=1}^{d} V_i$. The CO problem is $\max_{S:|S \cap V_i|=k_i} f(S)$. When $c = 1$, this reduces to the cardinality case. In step 3 of Algorithm 1, we obtain a vertex in the support of $\mathbf{x}_t$ by setting $\mathbf{1}_{S_t} = \text{argmax}_{\mathbf{c} \in \{0,1\}^n, \|\mathbf{c}(V_i)\|_1 = k_i} \mathbf{x}_t^\top \mathbf{c}$. Step 4 matches Equation (8). The encoder can similarly be constrained to the polytope, with scaling factor $s$ being handled per partition.

**Spanning Trees.** Let $G = (V, E)$ be a graph with $n$ nodes and $m$ edges. The CO problem is $\max_{S \subseteq E: S \in \mathcal{F}} f(S)$, where $\mathcal{F}$ is the set of full *spanning forests* of $G$. The convex hull of feasible sets is the *graphical matroid base polytope*. In Step 3 of Algorithm 1, we obtain a vertex in the support of $\mathbf{x}_t$ by finding a maximum spanning forest. The encoder learns edge weights to perturb the uniform spanning tree distribution, where we use Kirchhoff's Matrix–Tree Theorem [41] and results from spectral graph theory.

**Independent Sets.** The maximum independent set problem is an NP-hard problem where the goal is to find the largest subset of nodes $S$ in a graph $G = (V, E)$ such that no pair of nodes in $S$ is adjacent. The strong optimization problem for the independent set polytope is NP-Hard. Furthermore, there is no compact description of the polytope so this is a particularly challenging case. We circumvent those difficulties by considering its relaxation, which admits a compact description and is known as the *fractional stable set polytope* [55]. We use a fast gradient-based approach to project to the polytope. Steps 3 and 4 are chosen according to the standard template set by the GLS decomposition.

## 5 Experiments

### 5.1 Experimental setup

We consider the Maximum Coverage problem [39], a fundamental combinatorial problem with many variants that have been extensively studied. It appears in numerous applications, including document summarization [54] and identifying influential users in social networks [37, 15]. Let $U = \{e_1, \ldots, e_m\}$ be a set of $n$ elements and $V = \{T_1, \ldots, T_n\}$ be $n$ subsets of $U$. Given $k \leq n$, the `Max Coverage` problem asks for $k$ subsets whose union covers the largest number of elements. In the weighted version, each $e_i$ has a weight, and the goal is to maximize the total weight of covered elements. This problem can be represented as a bipartite graph $G = (U, V, E)$, where $(e_j, T_i) \in E$ iff $e_j \in T_i$. A node $u \in U$ is covered by $S \subseteq V$ if it is adjacent to some vertex in $S$. Set $f(S, u) = 1$ if $u$ is covered by $S$, and $f(S, u) = 0$ otherwise. The optimization problem is $\max_{S \subseteq V, |S|=k} \sum_{u \in U} f(S, u)$. Next we provide details on the experimental results that are presented in Figure 2 and the code can be found in this link.

*Problem Encoder:* To encode the bipartite graph, we use three layers of GraphSage [27] followed by a fully-connected layer with our *noise perturbation layer* to map the output vector to the hypersimplex $\Delta_{n,k}$. We add normalization and residual connections in between the GraphSage layers.

**Baselines.** We adopt the same baselines as prior work, including: `random`, which samples $k$ candidates uniformly at random over multiple trials and selects the best within 240 seconds; `Greedy algorithm` [61] that iteratively adds the element with the largest marginal gain and achieves the well-known $(1 - \frac{1}{e})$-approximation guarantee; `Gurobi` for exact MIP-based solutions, with time limited to 120 seconds for each graph; `CardNN` [88] and its variants (with and without test-time optimization, following [10] approach); a naive version of `EGN` with a naive probabilistic objective construction and iterative rounding [35]; `UCOM2` with variants [10]; and a Reinforcement Learning (`RL`) baseline. For the RL baseline, we follow the instructions in [45, 69] to use GraphSage layers [28] as policy network with Actor-Critic [44] algorithm to train on the same problem instances. We classify `UCOM2` as a non-learning method because their incremental greedy de-randomization, when applied to a uniform noise or all-zero input vector, performs as well as—or in some cases better than—when applied to the output of their encoder network suggesting that its performance is mostly due to the greedy module.

**Dataset and training.** We use synthetic graphs for training and real-world graphs for testing. Two $k$ values are chosen per dataset to assess method generalizability. (Due to space constraints, only one value of $k$ is shown; the full set is in the appendix.) `Random graphs:` the number in front of "Random" indicates the size of $U$. Each group contains 100 graphs from either a uniform or Pareto

distribution. `Real-world graphs:` Built from set systems like Twitch and Railway, each group includes multiple graphs from the same source for broader coverage. Due to limited large-scale data, real-world datasets are used only for testing, with training done on synthetic data. We point out that scaling the probability weights in Equation (8) generates diverse decompositions, exposing the model to more sets during training and inference. We use a list of *scaling factors* to produce these decompositions in *parallel*. See Appendix C.1 for details.

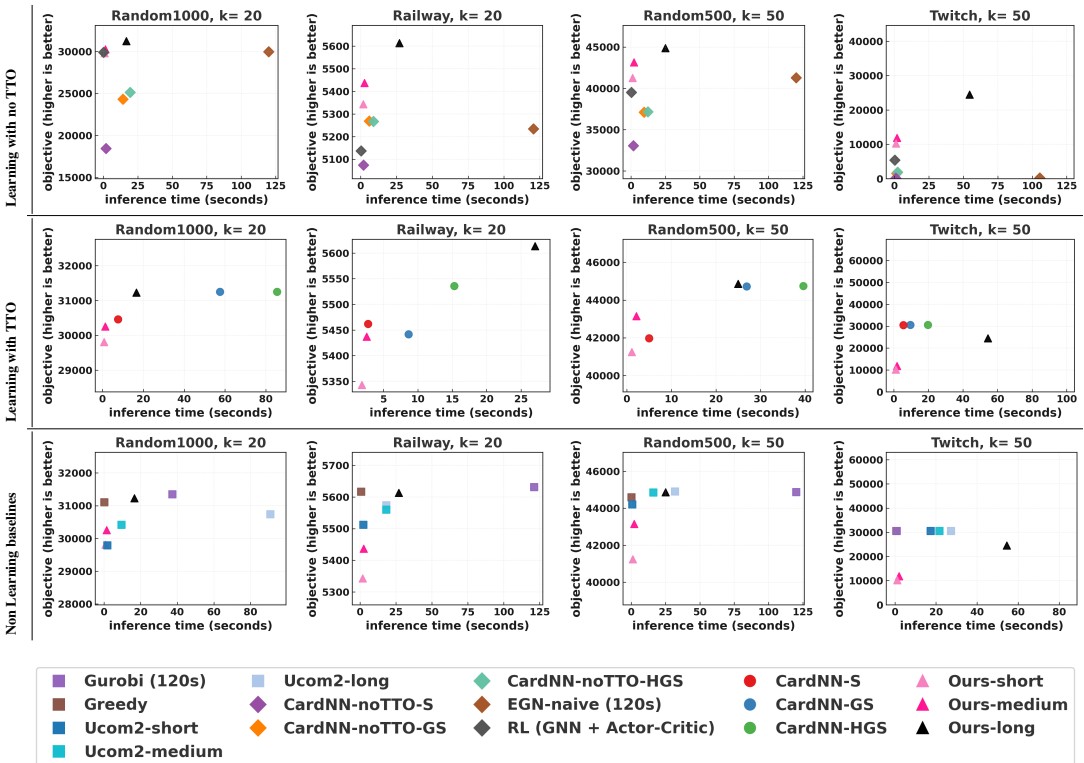

Figure 2: Performance comparison of our method against baseline approaches across three evaluation settings—Learning without Test-Time Optimization (TTO), Learning with TTO, and non-learning/traditional baselines—on multiple datasets. Metrics used are inference time (lower is better) and objective value (higher is better). In the Learning without TTO setting, our short version consistently outperforms all baselines in both inference time and objective value, demonstrating strong learning capability. When extended to medium and long versions, our method surpasses most TTO-based baselines across datasets, with the exception of the Twitch dataset. It also outperforms the greedy algorithm on multiple benchmarks. (Random(240s) is not included in the plots due to space constraints. Detailed result numbers can be found in Appendix G.6.)

## 5.2 Discussion

**Learnability results.** Our end-to-end pipeline shows a very strong learning ability in comparison to the existing learning baselines. In the setting where no method performs optimization at inference time, the `short` version of our method already shows strong performance compared to existing baselines. In this variant, we apply our decomposition procedure to the output of the neural encoder and select the best set from the decomposition. Based on the rounding guarantees of our extension, the resulting discrete solution is provably at least as good as the encoder's output. Compared to other baselines and `CardNN-noTTO` (across all three variants), our inference method achieves superior efficiency in both runtime and output quality. Note that we do not include the `UCOM2` method in this comparison, as it falls outside the scope of this setting; more details are provided later in the paper. Regarding the Twitch dataset, although our `short` method performs significantly better than other approaches in this setting, we observe that all learning-based methods perform poorly overall on this dataset. As mentioned in the introduction, this underperformance is largely attributed to the

architecture of the encoder network and the specific characteristics of the data distribution. This highlights a need for further investigation and suggests that adapting the model architecture to better suit such data is an important direction for future work.

**Test-Time Optimization (TTO).** In this setting, additional computational time is allowed during inference to improve solution quality. Specifically, we perform a simple optimization procedure: after selecting the best set from the decomposition of the encoder output, we apply local improvement by swapping a few elements with those from other sets in the decomposition's support. This lightweight optimization step demonstrates that the encoder has successfully learned meaningful structure from the data, as reflected in the quality of the decomposition's support. This is the `medium` variant of our method. In the `long` version we moreover perform data augmentation similar to [34]. These procedures yield noticeable improvements on the `Random500` and `Random1000` datasets, and produce reasonably good results on the Railway dataset. On the twitch dataset, the `CardNN` methods directly optimize the neural network output on the test sample with Adam, therefore overcoming the model architecture problem. For our methods, performance remains poor on the Twitch dataset, consistent with observations discussed earlier.

**Comparison with Non-Learning methods.** While greedy is an efficient baseline with a strong approximation guarantee, our method performs competitively and is capable of outperforming it on datasets like `Random500` for larger values of $k$. We are also able to outperform UCOM in several cases (e.g., `Random1000` and `Railway`), though there are instances where greedy and/or UCOM perform the best, such as the twitch dataset.

**Ablation.** We conduct an ablation study on the NP-hard problem of finding a cardinality-constrained maximum cut. Given a graph $G = (V, E)$, the goal is to find a set $S$ of $k$ nodes such that the number of edges with exactly one endpoint in $S$ is maximized. We compare the following baselines on two datasets and two settings for the value $k$. Our decomposition-based method consists of two main parts: a NN to predict a point in the polytope, and a loss function to optimize it and aid in learning. To probe the effect of the loss function, we compare the quality of solutions obtained by the decomposition when the input is *i)* a random point projected onto the polytope *ii)* a random point optimized with Adam on the loss. To probe the effect of the neural network, we test the solutions obtained by the decomposition of the predictions of a neural network that has optimized the loss on the same data. Finally, to assess generalization, we compare the decomposed sets obtained from the predictions of a neural net that has been trained on a separate training set. As a sanity check, we also include a greedy algorithm. The table and more details of the experiment are provided in Appendix H. We consistently observe that directly optimizing the extension on the test set with Adam significantly improves the objective compared to decomposing a randomly chosen point and reporting the result of its best performing set. Furthermore, the neural net optimized directly on the test set improves consistently over direct optimization with Adam and is competitive with greedy pointing to the benefits of parametrization. Finally, the performance of the SSL approach is close to the competing neural baseline, suggesting that the model generalizes well.

# 6 Conclusion

We proposed a novel geometric approach to neural combinatorial optimization with constraints that effectively tackles the challenges of loss function design and rounding. Our method achieved strong empirical results against both neural baselines and classical methods, and opens up promising avenues for further research on incorporating classical geometric algorithms into neural net pipelines. It is important to note that the model architecture can play an important role in results; we have not focused on this aspect here but it merits further exploration in future work. Our framework is applicable whenever an oracle for the strong maximization problem exists, but its efficiency will vary based on the polytope. The structure of the polytope influences our ability to obtain interior points, to solve the strong optimization problem, and to compute intersections with its boundary. Nevertheless, we believe that the blueprint described in this work paves the way towards new hybrid algorithms that can effectively tackle complex combinatorial problems.

# 7   Acknowledgements

Stefanie Jegelka acknowledges support from NSF AI Institute TILOS and the Alexander von Humboldt Foundation. Nikolaos Karalias acknowledges support from the SNSF, in the context of the project "General neural solvers for combinatorial optimization and algorithmic reasoning" (SNSF grant number: P500PT_217999) and from NSF AI Institute TILOS.

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

## A  Additional discussion of related work

Reinforcement learning (RL)-based construction heuristics have been one of the leading paradigms in neural CO. The first application to use RL with neural networks are [5], where they combined a Pointer Network [85] with actor-critic reinforcement algorithm [44] to generate solutions for the Traveling Salesman Problem. Later works including [45], [47], and [69] have gained great success in solving larger-scale CO problems with RL algorithms, and also showed the ability to generalize to large scale problem instances. There are similarities between our approach and the general approach followed in RL algorithms like policy gradient [83] that are worth discussing. In RL, one learns a distribution that maximizes the expected reward. This is done by sampling solutions from the outputs of a neural network which are treated as the parameters of a probability distribution, then calculating their reward and weighing it by their log probability, and finally backpropagating to the parameters of the neural network. Stochasticity is often at the core of RL and the ways that one may sample solutions for a given problem differ. Our goal is to propose a structured approach that relies on methods from polyhedral combinatorics and convex geometry. Concretely, the output of the neural network is transformed so that it yields a point in the convex hull of feasible solutions of the problem. Then, instead of sampling, we leverage the decomposition algorithm to explicitly construct the support of the distribution that is parametrized by the neural net output. The extension we maximize can be viewed as the expected 'reward' (objective), which is similar to how RL operates. In terms of differentiability, our approach is slightly different since our extension is a.e. differentiable and we can directly use automatic differentiation while RL deals with this using techniques like the log-derivative trick. Importantly, extensions can be used to optimize any black-box objective which is also the case for RL.

Other related lines of work include the Predict then Optimize paradigm [20] which combines learned predictions with a deterministic optimization problem that is parametrized by them. The mathematical relationship to extensions originates from the fact that extensions can be viewed as feasible solutions to a linear program for the convex envelope that is parametrized by some neural net predictions [36]. Another line of work that combines predictive and algorithmic components is the work on backpropagating through solvers [68, 65, 66]. For a more complete reference on combinatorial optimization with (graph) neural networks we refer the reader to [12].

## B  Decomposition Theorem: extended discussion

We will provide an extended discussion of the GLS result and its implications for our decomposition approach. For that we will need some definitions as presented in [25] in order to provide a self-contained discussion of the theorem, its proof, and its implications for our approach.

**Definition B.1** (Defining properties of polyhedra). Let $\mathcal{P} \subseteq \mathbb{R}^n$ be a polyhedron and let $\varphi$ and $\nu$ be positive integers.

1. We say that $\mathcal{P}$ has **facet-complexity** at most $\varphi$ if there exists a system of inequalities with rational coefficients that has solution set $\mathcal{P}$ and such that the encoding length of each inequality of the system is at most $\varphi$. In case $\mathcal{P} = \mathbb{R}^n$ we require $\varphi \geq n + 1$.

2. We say that $\mathcal{P}$ has **vertex-complexity** at most $\nu$ if there exist finite sets $V, E$ of rational vectors such that
$$\mathcal{P} = \operatorname{conv}(V) + \operatorname{cone}(E)$$
and such that each of the vectors in $V$ and $E$ has encoding length at most $\nu$. In case $\mathcal{P} = \emptyset$ we require $\nu \geq n$.

3. It can be shown that the vertex-complexity and facet-complexity are equivalent from the perspective of polynomial-time algorithm design, i.e., one can be written as a polynomial of the other [25, Lemma 6.2.4]. Therefore, we will only use the facet complexity to simplify discussions.

4. A **well-described polyhedron** is a triple $(\mathcal{P}; n, \varphi)$ where $\mathcal{P} \subseteq \mathbb{R}^n$ is a polyhedron with facet-complexity at most $\varphi$. The encoding length $\langle \mathcal{P} \rangle$ of a well-described polyhedron $(\mathcal{P}; n, \varphi)$ is
$$\langle \mathcal{P} \rangle = \varphi + n.$$
$\square$

**Definition B.2** (Strong optimization problem)**.** The strong optimization problem for a given rational vector $\mathbf{x}$ in $n$ dimensions and any well described polytope $(\mathcal{P}; n, \varphi)$ is given by

$$\max \mathbf{c}^\top \mathbf{x}, \quad \mathbf{c} \in \mathcal{P}. \tag{9}$$

Next, we state the GLS result which forms the foundation for Theorem 4.1.

**Theorem B.3.** *(GLS decomposition) There exists an oracle polynomial-time algorithm that for any well-described polyhedron $(\mathcal{P}; n, \varphi)$ given by a strong optimization oracle, for any rational vector $\mathbf{x}$, finds affinely independent vertices $\mathbf{1}_{S_0}, \mathbf{1}_{S_1}, \ldots, \mathbf{1}_{S_k}$ and positive rational numbers $\lambda_0, \lambda_1, \ldots, \lambda_k$ such that $\mathbf{x} = \sum_{t=0}^{k} \lambda_t \mathbf{1}_{S_t}$.*

*Proof.* Set $\mathbf{x}_0 = \mathbf{x}$. The algorithm begins by finding the smallest face that contains $\mathbf{x}$ and then obtain a vertex $\mathbf{1}_{S_0}$ of that face. If that vertex is $\mathbf{x}_0$ then the algorithm terminates. Otherwise, a half-line is drawn from $\mathbf{1}_{S_0}$ through $\mathbf{x}_0$ until the boundary of the polytope is intersected. At the intersection point, we obtain a new iterate $\mathbf{x}_1$. Notice that the previous iterate $\mathbf{x}_0$ can be written as a convex combination of $\mathbf{x}_1$ and $\mathbf{1}_{S_0}$ since it is part of a line with endpoints $\mathbf{x}_1$ and $\mathbf{1}_{S_0}$. Again, we find the smallest face containing $\mathbf{x}_1$ and obtain a vertex $\mathbf{1}_{S_1}$ from it. If it matches the iterate, we terminate, otherwise we repeat the process. Since each iterate belongs to the faces of all the previous iterates but not the subsequent ones, the iterates are affinely independent and hence the process has to terminate in at most $k \leq n$ steps. Given the starting vector and the vertices it is straightforward to compute the coefficients with linear algebra. Furthermore the encoding length of each iterate is polynomial since it is the unique point of intersection of affine subspaces spanned by $\mathbf{x}$ and the vertices of the polytope. □

### B.1 Constructing a polynomial time and a.e. differentiable decomposition

Based on the GLS proof, we can identify two conditions that yield a valid decomposition when satisfied. Assume we already have a starting point in the interior of the polytope, then:

1. For each iterate, finding a minimal face that contains it and retrieving a vertex from it.

2. The ability to find intersection points of half-lines with the boundary of the polytope.

Those requirements can be fulfilled via the strong maximization oracle.

**Obtaining a polytope vertex for Step 3 of Algorithm 1.** The proof requires a vertex of a face containing the iterate $\mathbf{x}_t$. The oracle may return any maximizer of the optimization problem that is not a vertex. The face $F$ containing $\mathbf{x}_t$ is the set of vectors attaining $\max_{\mathbf{c} \in \mathcal{P}} \mathbf{x}_t^\top \mathbf{c}$ [75]. Given access to the vectors of the face $F$, we need to be able to retrieve a vertex of that face. This can be done by a simple technique, where we make a call to the oracle for the perturbed program

$$\max \bar{\mathbf{c}}^\top \mathbf{x}_t, \quad \mathbf{c} \in \mathcal{P}, \tag{10}$$

with $\bar{\mathbf{c}} = \mathbf{c} + [\epsilon, \epsilon^2, \ldots, \epsilon^n]^\top$ for some small $\epsilon$ [25][Remark 6.5.2].

**Obtaining boundary intersections.** It is a well-known fact that an optimization oracle can be converted to a separation oracle in polynomial time. The separation oracle can be used to perform line search across the ray sent from the vertex obtained in Step 3 that passes through the current iterate. The line search can be used to calculate the coefficient that yields the next iterate which lies on a lower-dimensional face of the polytope and this process can be performed repeatedly until the full decomposition is computed.

**Almost everywhere differentiable decomposition.** To enable learning-based approaches with the decomposition, the coefficients at each iteration have to be almost everywhere differentiable functions of the iterates. We show how this is possible in the context of the recurrence Equation (4) by building on the main idea behind the GLS theorem just by leveraging calls to the strong optimization oracle.

**Theorem 4.1.** *There exists a polynomial-time algorithm that for any well-described polytope $\mathcal{P}$ given by a strong optimization oracle, for any rational vector $\mathbf{x}$, finds vertex-probability pairs $\{p_{\mathbf{x}_t}(S_t), \mathbf{1}_{S_t}\}$ for $t = 0, 1, \ldots, n-1$ such that $\mathbf{x} = \sum_{t=0}^{n-1} p_{\mathbf{x}_t}(S_t) \mathbf{1}_{S_t}$ and all $p_{\mathbf{x}_t}(S_t)$ are almost everywhere differentiable functions of $\mathbf{x}$.*

*Proof.* Let $\mathcal{P} \subset \mathbb{R}^n$, current iterate $\mathbf{x}_t \in \mathcal{P}$, and the vertex $\mathbf{1}_{S_t}$ given by the oracle in Step 3. Consider the recursive decomposition

$$\mathbf{x}_t \;=\; \alpha_t\,\mathbf{1}_{S_t} \;+\; (1-\alpha_t)\,\mathbf{x}_{t+1}, \qquad \mathbf{x}_{t+1} \in \mathcal{P}, \quad 0 \le \alpha_t < 1.$$

First, we will focus on differentiability. Recall that we can view each iteration above as intersecting a ray that passes through the vertex given by the strong optimization oracle and the current iterate with the boundary of the polytope. To find the intersection, set the ray direction $\mathbf{d}_t := \mathbf{x}_t - \mathbf{1}_{S_t} \ne \mathbf{0}$ and let

$$\mathbf{y}(\tau) \;:=\; \mathbf{1}_{S_t} + \tau\,\mathbf{d}_t, \qquad \tau \ge 0,$$

so that $\mathbf{x}_t = \mathbf{y}(1)$. Note that we can recover the original iteration equation from $\mathbf{y}(\tau)$ via

$$\tau = \frac{1}{1-\alpha_t}.$$

Furthermore, note that the constraint $\mathbf{x}_t - \mathbf{1}_{S_t} \ne \mathbf{0}$ implies that the iterate cannot be a vertex of the polytope. Indeed, the proof of the GLS result states that the decomposition terminates at a vertex, which is consistent with this constraint. The intersection point with the boundary is given by the largest coefficient allowed that can yield a point in the polytope

$$\tau^*(\mathbf{x}_t; \mathbf{1}_{S_t}) \;=\; \max\{\,\tau \ge 0 : \mathbf{y}(\tau) \in \mathcal{P}\,\}.$$

To derive the maximum step that yields an intersection with the boundary we will leverage the strong optimization oracle. For any $\mathbf{c} \in \mathbb{R}^n$, the support function $h_{\mathcal{P}}(\mathbf{c}) := \max_{\mathbf{z} \in \mathcal{P}} \mathbf{c}^\top \mathbf{z}$ which can be computed using the strong optimization oracle satisfies

$$\mathbf{c}^\top \mathbf{y}(\tau) = \mathbf{c}^\top(\mathbf{1}_{S_t} + \tau\,\mathbf{d}_t)$$
$$\le\; h_{\mathcal{P}}(\mathbf{c}).$$

This implies

$$\tau \;\le\; \frac{h_{\mathcal{P}}(\mathbf{c}) - \mathbf{c}^\top \mathbf{1}_{S_t}}{\mathbf{c}^\top \mathbf{d}_t}, \qquad \text{whenever } \mathbf{c}^\top \mathbf{d}_t > 0.$$

Taking the best bound yields

$$\tau^*(\mathbf{x}_t; \mathbf{1}_{S_t}) = \min_{\mathbf{c}:\,\mathbf{c}^\top \mathbf{d}_t > 0} \frac{h_{\mathcal{P}}(\mathbf{c}) - \mathbf{c}^\top \mathbf{1}_{S_t}}{\mathbf{c}^\top \mathbf{d}_t}. \tag{11}$$

Rewriting in terms of $a^* = 1 - \frac{1}{\tau^*}$ we obtain

$$a^*(\mathbf{x}_t; \mathbf{1}_{S_t}) = 1 - \max_{\mathbf{c}:\,h_{\mathcal{P}}(\mathbf{c}) > \mathbf{c}^\top \mathbf{1}_{S_t}} \frac{\mathbf{c}^\top(\mathbf{x}_t - \mathbf{1}_{S_t})}{h_{\mathcal{P}}(\mathbf{c}) - \mathbf{c}^\top \mathbf{1}_{S_t}}. \tag{12}$$

Note that for a positive denominator, the maximum of the ratio is always between zero and one. To see why that is the case, notice that $\mathbf{c} = \mathbf{x}_t - \mathbf{1}_{S_t}$ is always feasible and it always yields a positive ratio. The ratio is upper bounded by 1 because $h_{\mathcal{P}}$ is the support function, so we have for $\mathbf{x} \in \mathcal{P}$ that

$$h_{\mathcal{P}}(\mathbf{c}) \ge \mathbf{c}^\top \mathbf{x} \;\implies\; h_{\mathcal{P}}(\mathbf{c}) - \mathbf{c}^\top \mathbf{1}_{S_t} \ge \mathbf{c}^\top \mathbf{x} - \mathbf{c}^\top \mathbf{1}_{S_t}.$$

The optimizer $a^*$ is a piecewise linear function of $\mathbf{x}_t$ as a maximum over linear functions. Piecewise linear functions are locally Lipschitz, and therefore differentiable almost everywhere. Note that $\mathbf{1}_{S_t}$ is the maximizer of the strong optimization problem from Equation (9). The maximizer depends on $\mathbf{x}_t$ and its gradients are zero almost everywhere because it is a piecewise constant function of $\mathbf{x}_t$. Given an optimal $\mathbf{c}^*$, we can calculate $a^*$ and hence the probabilities in step 6 of Algorithm 1. Recall that the probabilities are given by $p_{\mathbf{x}_t}(S_t) = a_t \prod_{i=0}^{t}(1 - a_i)$. Since each $a_i$ is an a.e. differentiable function of its iterate and each iterate depends on the preceding iterate through the differentiable recursive update rule, Equation (4), we can calculate the derivatives of each $p_{\mathbf{x}_t}(S_t)$ with respect to the starting point in the polytope $\mathbf{x}$ with standard automatic differentiation packages.

Finally, we want to show that the a.e. differentiable decomposition can be computed in polynomial time. Step 3 requires solving an LP and can be done in polynomial time. If we show that Step 4 is also computable in polynomial time, we are done. To do so, we need to show that the maximum coefficient for the ray-boundary intersection is computable in polynomial time. We will show that the optimal

$\mathbf{c}^*$ can be found by solving a standard convex optimization problem. Observe that Equation (11) can be written as

$$\min_{\mathbf{c} \in \mathbb{R}^n} \; h_{\mathcal{P}}(\mathbf{c}) - \mathbf{c}^\top \mathbf{1}_{S_t}, \text{ subject to } \mathbf{c}^\top \mathbf{d}_t = 1, \tag{13}$$

by a simple change of variables $\mathbf{c} \to \mathbf{c}/(\mathbf{c}^\top \mathbf{d}_t)$ because the ratio in Equation (11) is invariant to rescalings of $\mathbf{c}$. Therefore, the objective is a sum of the support function of the polytope (convex) and $-\mathbf{c}^\top \mathbf{1}_{S_t}$ which is just a linear function (convex). Hence, the sum is also convex, so we have a convex minimization problem in $\mathbf{c}$ subject to a linear constraint. This is a standard linear program that we can efficiently solve with any of the well known algorithms to obtain the optimal $\mathbf{c}^*$. $\qquad\square$

**Special case: Compact polytope description.** An important special case that simplifies the design of efficient decompositions for end-to-end learning is when the polytope admits a compact description in terms of polynomially many inequalities. Recall that for a general polytope

$$\mathcal{P} = \big\{ \mathbf{x} \in \mathbb{R}^n : \mathbf{z}_i^\top \mathbf{x} \le b_i, \; i = 1, \ldots, m \big\},$$

Again, we pick the convex combination coefficient as

$$a_t^{\max} = \max\big\{ a_t \in [0, 1) : \; \mathbf{x}_{t+1}(a_t) \in \mathcal{P} \big\},$$

This choice of coefficient leads to the next iterate being as far as possible from the current one. This means the coefficient will be pushed until some constraints of the polytope are tight for the new iterate and hence the boundary will be intersected. At iteration $t$, with current iterate $\mathbf{x}_t$ and chosen corner $\mathbf{1}_{S_t}$, we may determine the coefficient by enforcing each face inequality. In particular:

$$
\begin{aligned}
\mathbf{z}_i^\top \mathbf{x}_{t+1} \;\le\; b_i \quad &\Longleftrightarrow \quad \mathbf{z}_i^\top \frac{\mathbf{x}_t - a_t \mathbf{1}_{S_t}}{1 - a_t} \;\le\; b_i \\
&\Longleftrightarrow \quad a_t \;\le\; \frac{b_i - \mathbf{z}_i^\top \mathbf{x}_t}{b_i - \mathbf{z}_i^\top \mathbf{1}_{S_t}}, \quad \text{whenever } b_i - \mathbf{z}_i^\top \mathbf{1}_{S_t} > 0.
\end{aligned}
$$

Hence

$$a_t^{\max}(\mathbf{x}_t) = \min_{i: \, b_i - \mathbf{z}_i^\top \mathbf{1}_{S_t} > 0} \; \frac{b_i - \mathbf{z}_i^\top \mathbf{x}_t}{b_i - \mathbf{z}_i^\top \mathbf{1}_{S_t}}, \tag{14}$$

which shows that the maximum feasible weight $a_t$ is a well-defined differentiable function of the current iterate $\mathbf{x}_t$. This is crucial because it allows us to backpropagate through the coefficient and leverage the decomposition in gradient-based optimization.

Equation (14) and Equation (10) are sufficient for our decomposition. Specifically, given access to a fast algorithm for Equation (14) and a fast algorithm for Equation (10) we can obtain a tractable decomposition that allows us to build $\mathcal{D}_{\mathcal{C}}(\mathbf{x})$ in a differentiable manner. It should be noted that if the polytope is defined by exponentially many inequalities, then iterating over the inequalities to find the minimum becomes intractable.

## B.2   On the applicability of our approach to different combinatorial problems

**Computational complexity vs applicability.** The existence of a strong optimization oracle for the feasible set polytope is a sufficient condition for the applicability of our approach. Superficially, this may seem to imply that our approach is only applicable to problems that are easy from a computational complexity standpoint. However, as we will explain, the computational complexity of the problem we are trying to solve is not sufficient to determine the applicability of our method.

Consider the Traveling Salesperson Problem (TSP) Polytope. The strong optimization problem for that polytope is not solvable in polynomial time unless P = NP. Having such an oracle would be equivalent to having an oracle for the problem itself. Fortunately, we can still apply our method to TSP. In combinatorial optimization, one can reformulate a problem to "offload" some of its difficulty from the constraints to the objective. While TSP can be viewed as a linear optimization problem over the TSP polytope, it can also be viewed as a quadratic optimization problem over a permutation (Birkhoff) polytope. Linear maximization can be done in polynomial-time in Birkhoff polytopes,

which means we can efficiently construct an extension of the quadratic objective over the Birkhoff polytope.

**Polytope description vs applicability.** Another important consideration is the size of the description of the polytope. Indeed, as we showed in Appendix B.1, if a polytope admits a compact description in terms of a polynomial number of inequalities, then we can straightforwardly proceed with our decomposition. On the other hand, even when that is not the case our method can still be applied using just the optimization oracle as it can be seen in the proof of Theorem 4.1. An intuitive example is the base polytope of a submodular function. It is defined by exponentially many inequalities but linear maximization over the polytope can be done efficiently with the greedy algorithm. The optimal value of the linear maximization problem in the polytope is in fact given by the Lovász extension of the submodular function.

# C  Decomposition with controlled approximation

One of the limitations of the decomposition as presented in the approach is that the support of the distribution we obtain depends exactly on the ambient dimension of the space. There are a few reasons why this might not be desirable. First, an exact decomposition like this fixes at most $n$ sets in the support. That may be too restrictive in problems with 'sparse rewards', i.e., when good objective values are achieved on only a few points. It would be preferable if we had more control over the number of sets in the support of the distribution, which could in turn lead to better outcomes for optimization. We show how we can achieve this with a simple tweak that allows us to control the reconstruction quality, which we also utilize in our max coverage experiments.

## C.1  Decomposition via rescaling

Start with $\mathbf{x}_0 = \mathbf{x}$. For each iteration $t = 0, 1, 2, \ldots, k$, we first pick the corner $\mathbf{1}_{S_t}$ and compute the iteration coefficient

$$\tilde{a}_t = \min\big\{\min_{i \in S_t} \mathbf{x}_t(i), \ \min_{j \notin S_t}(1 - \mathbf{x}_t(j))\big\}.$$

The difference now is that instead of picking the maximum coefficient ,we will rescale the $\tilde{a}_t$ in each step by some fixed constant $b \in (0, 1]$. This means that the next iterate will not be intersecting the boundary and hence the algorithm will require more steps to terminate, hence requiring a tolerance parameter $\epsilon$ as we have described in Algorithm 1. Additionally, we fix some lower bound $\ell$ on the rescaling so that

$$a_t = \begin{cases} b\,\tilde{a}_t, & b\,\tilde{a}_t \geq \ell, \\ \tilde{a}_t, & b\,\tilde{a}_t < \ell. \end{cases}$$

Using this coefficient we compute the new iterate using Equation (4).

**Proposition C.1.** *Suppose we pick step 3 in Algorithm 1 according to Equation* (5). *Then the reconstruction error of the algorithm (step 8 in Algorithm 1) decays exponentially in $k$.*

*Proof.* After $k$ iterations we may express $\mathbf{x}$ as

$$\mathbf{x} = \sum_{t=0}^{k} \underbrace{\Big(\prod_{i=0}^{t-1}(1 - a_i)\Big) a_t}_{p_{\mathbf{x}_t}(S_t)} \mathbf{1}_{S_t} + \Big(\prod_{i=0}^{k}(1 - a_i)\Big) \mathbf{x}_{k+1}.$$

Recall from step 8 of Algorithm 1 we have that the stopping criterion (given by the reconstruction error) is

$$\Big\|\mathbf{x} - \sum_{t=0}^{k} p_{\mathbf{x}_t}(S_t)\mathbf{1}_{S_t}\Big\| \leq \epsilon.$$

Clearly, the term inside the norm on the left hand side of the inequality is the residual

$$\mathbf{r}_{k+1} = \Big(\prod_{i=0}^{k}(1 - a_i)\Big) \mathbf{x}_{k+1}.$$

For the reconstruction error we just need to compute the norm of the residual and show that it decays exponentially in $k$. Hence, we calculate

$$\|\mathbf{r}_{k+1}\|_2 = \left(\prod_{i=0}^{k}(1 - a_i)\right)\|\mathbf{x}_{k+1}\|_2$$
$$\leq (1 - \ell)^k \|\mathbf{x}\|_2$$
$$\leq (1 - \ell)^k n,$$

which completes the proof. □

Given some tolerance parameter $\epsilon$, this allows us to control the number of sets considered in the iteration. We observed that using this approach helped with the training dynamics of the model as well as with achieving better approximate solutions.

## D  Deferred proofs from Section 4.2

**Proof of Theorem 4.3**

*Proof.* The proof follows the same argument as the proof in [25][Theorem 6.5.11]. To make the result more intuitive we may assume without loss of generality that the entries of $\mathbf{x}_t$ are sorted, i.e. $\mathbf{x}_t(1) \geq \mathbf{x}_t(2) \geq \cdots \geq \mathbf{x}_t(n)$. Under this ordering,

$$\mathbf{1}_{S_t} = \arg \max_{\substack{\mathbf{c} \in \{0,1\}^n \\ \|\mathbf{c}\|_1 = k}} \mathbf{x}_t^\top \mathbf{c},$$

selects exactly the top $k$ coordinates. Since $\|\mathbf{x}_t\|_1 = k$ and $\|\mathbf{1}_{S_t}\|_1 = k$, taking the $\ell_1$-norm of both sides of Equation (4) gives

$$k = a_t\, k + (1 - a_t)\, \|\mathbf{x}_{t+1}\|_1,$$

so $\|\mathbf{x}_{t+1}\|_1 = k$ and the sum-to-$k$ constraint is preserved exactly. Apart from the norm constraint, we also have to ensure that each coordinate remains in the hypercube. To keep each coordinate in the hypercube we solve for the coefficient $a_t$ that satisfies $0 \leq \mathbf{x}_{t+1} \leq 1$ in our recurrence equation. First, we rearrange Equation (4)

$$\mathbf{x}_{t+1} = \frac{\mathbf{x}_t - a_t \mathbf{1}_{S_t}}{1 - a_t}$$

and note that subtraction at the numerator affects only the top $k$ coordinates. It is clear that for $i \in S_t$ the coordinates cannot possibly exceed one because the denominator is always larger, but they may drop below 0. So we need to ensure

$$\frac{\mathbf{x}_t(i) - a_t}{1 - a_t} \geq 0, \tag{15}$$

which implies $a_t \leq \min_{i \in S_t} \mathbf{x}_t(i)$ and corresponds to a constraint on the k-th coordinate in this ordering. On the other hand, for $j \notin S_t$ they can only increase in magnitude, so they may exceed 1, which leads us to

$$\frac{\mathbf{x}_t(j)}{1 - a_t} \leq 1, \tag{16}$$

and hence $a_t \leq 1 - \max_{j \notin S_t} \mathbf{x}_t(j)$ which constrains the $(k+1)$-th coordinate. Ensuring that both Equation (15) and Equation (16) are satisfied, and following the GLS proof, we pick the largest possible coefficient which leads us to Equation (8):

$$a_t = \min\Big\{\min_{i \in S_t} \mathbf{x}_t(i),\ 1 - \max_{j \notin S_t} \mathbf{x}_t(j)\Big\}.$$

If $a_t = \min_{i \in S_t} \mathbf{x}_t(i)$, the $k$-th entry is set to zero; otherwise the $(k+1)$-th coordinate is set to 1. Note that from Equation (4), once a coordinate has been fixed to either 0 or 1, it cannot change in subsequent iterations. Therefore, the process will terminate when the next iterate becomes a corner of $\Delta_{n,k}$. Since there are $k$ zeros and $n - k$ ones at the final iterate and each iteration fixes exactly

one coordinate, the process terminates after at most $k + (n - k) = n$ iterations. In line with the GLS proof, the next iterate intersects the boundary of the polytope at a face of lower dimension because the minimum is guaranteed to fix a coordinate either to 0 or to 1.

Finally, because each $a_t$ is defined via pointwise minima and maxima of the entries of $\mathbf{x}_t$, it is almost everywhere differentiable in $\mathbf{x}_t$. By the chain rule, the resulting probability weights

$$p_{\mathbf{x}_t}(S_t) = a_t \prod_{i<t}(1 - a_i)$$

are almost-everywhere differentiable in the original input $\mathbf{x}_0$. This completes the proof. $\qquad\square$

**Proof of Proposition 4.4**

*Proof.* Let $\mathbf{z} \in [0, 1]^n$ and $\mathbf{x} = s\widetilde{\mathbf{z}} + \mathbf{u}$, and define

$$\mu = \frac{1}{n}\sum_{i=1}^{n} z_i, \qquad \widetilde{\mathbf{z}} = \mathbf{z} - \mu\mathbf{1}, \qquad \mathbf{u} = \frac{k}{n}\mathbf{1}.$$

Since $\sum_i \widetilde{z}_i = \sum_i (z_i - \mu) = 0$, we have

$$\sum_{i=1}^{n} x_i = s\sum_{i=1}^{n} \widetilde{z}_i + \sum_{i=1}^{n} u_i = 0 + k,$$

so $\|\mathbf{x}\|_1 = k$. Next, for each coordinate

$$x_i = s\,(z_i - \mu) + \frac{k}{n}.$$

Because $0 \le z_i \le 1$, we have $-\mu \le z_i - \mu \le 1 - \mu$. Therefore

$$x_i \ge -s\,\mu + \frac{k}{n} = \frac{k}{n}\left(1 - s\,\frac{n\mu}{k}\right) \ge 0$$

by $s \le \frac{k}{n\mu}$, and

$$x_i \le s\,(1 - \mu) + \frac{k}{n} = 1 - \frac{n - k}{n}\left(1 - s\,\frac{n(1 - \mu)}{n - k}\right) \le 1$$

by $s \le \frac{n-k}{n(1-\mu)}$. Hence $0 \le x_i \le 1$ for all $i$, and we conclude $\mathbf{x} \in \Delta_{n,k}$. $\qquad\square$

## E   Case study: Partition Matroids and Spanning Trees

### E.1   Partition Matroid

Partition matroids provide a powerful and flexible framework for modeling constraints in set function optimization problems. They capture scenarios where elements are divided into categories, and we are allowed to select only a limited number from each category — a structure that arises naturally in many real-world applications. This structure naturally arises in applications such as sensor placement, where sensors are partitioned by geographic regions and each region has an independent budget [46]; job allocation and welfare maximization, where tasks are categorized by required skills and only a bounded number of tasks from each skill category can be assigned [11, 86]; and many other theoretical and practical settings [23, 14, 39]. Partition matroids thus provide a flexible abstraction for modeling structured selection problems with heterogeneous constraints.

A partition matroid or partitional matroid is a matroid that is a direct sum of uniform matroids. It is defined over a ground set in which the elements are partitioned into different categories. For each category, there is a cardinality constraint–a maximum number of allowed elements from this category. Formally, let $V_1, \ldots, V_c$ be disjoint subsets such that $V = \cup_{i=1}^{d} V_i$. Let $k_i$ be integers with $0 \le k_i \le |V_i|$. Consider the following combinatorial optimization problem:

$$\max_{S:|S\cap V_i|=k_i} f(S). \tag{17}$$

Note that when $c = 1$, this reduces to the cardinality case. The convex polytope of feasible solutions is known as the partition matroid base polytope and is described as follows:

$$\mathcal{B} = \left\{ \mathbf{x} : \forall S \subseteq E \sum_{e \in S} \mathbf{x}(e) \leq r(S), \text{ and } \|\mathbf{x}\|_1 = \sum_{i=1}^{c} k_i \right\} \tag{18}$$

$$r(S) = \sum_{i=1}^{c} \min(|S \cap E_i|, k_i) \ \forall S \subseteq E. \tag{19}$$

The vertices of the matroid base polytope $\mathcal{B}$ are the indicator vectors of sets whose intersection with every block $V_i$ has size exactly $k_i$. Hence, any vector $\mathbf{x} \in \mathcal{B}$ can be written as a convex combination of the indicator vectors of feasible sets. We prove that Algorithm 2, given $\mathbf{x} \in \mathcal{B}$, returns in polynomial time a distribution over feasible sets which is a.e. differentiable with respect to $\mathbf{x}$. Note that Algorithm 2 is an explicit version of our generic Algorithm 1, with steps 3 and 4 specified.

**Theorem E.1.** *Given $\mathbf{x} \in \mathcal{B}$, Algorithm 2 terminates after at most $O(n)$ iterations and returns $\{(p_{\mathbf{x}_t}(S_t), S_t)\}$ such that for every $S_t$ and $V_i$ we have $|S_t \cap V_i| = k_i$, and $\sum_t p_{\mathbf{x}_t}(S_t) = 1$ with $0 \leq p_{\mathbf{x}_t}(S_t) \leq 1$. Moreover, each $p_{\mathbf{x}_t}(S_t)$ is an almost everywhere differentiable function of $\mathbf{x}$.*

*Proof of Theorem E.1.* Suppose $\mathbf{x}_0 = \mathbf{x} \in \mathcal{B}$. We prove the correctness of our algorithm for every iteration $t$. To obtain a vertex of the polytope in the support of $\mathbf{x}_t$, we set

$$\mathbf{1}_{S_t} = \underset{\mathbf{c} \in \{0,1\}^n, \|\mathbf{c}(V_i)\|_1 = k_i}{\operatorname{argmax}} \mathbf{x}_t^\top \mathbf{c}$$

The optimal solution for the above problem is $S_t = \cup_{i=1}^{c} T_i$ where for each block $V_i$, $T_i$ is the indices of the top $k_i$ coordinates within $\mathbf{x}_t(V_i)$. Define

$$a_t = \min \left\{ \min_{i \in S_t} \mathbf{x}_t(i), 1 - \max_{i \notin S_t} \mathbf{x}_t(i) \right\}, \text{ and} \tag{20}$$

$$\mathbf{x}_{t+1} = \tfrac{1}{1-a_t} \left( \mathbf{x}_t - a_t \mathbf{1}_{S_t} \right) \tag{21}$$

For the correctness of Algorithm 2 it suffices to show $\mathbf{x}_{t+1}$ is in fact inside the matroid base polytope $\mathcal{B}$.

First, note that $\mathbf{x}_{t+1} \in [0,1]^n$ by the definition of $a_t$ and the fact that $\mathbf{x}_t \in [0,1]^n$. Second, for both $\mathbf{x}_t$ and $\mathbf{1}_{S_t}$, we have $\|\mathbf{x}\|_1 = \|\mathbf{1}_{S_t}\|_1 = \sum_{i=1}^{c} k_i$. Hence, by the definition $\mathbf{x}_t = a_t \mathbf{1}_{S_t} + (1 - a_t)\mathbf{x}_{t+1}$, it follows that $\|\mathbf{x}_{t+1}\|_1 = \sum_{i=1}^{c} k_i$. It remains to show for all $S \subseteq V$ it holds that $\sum_{e \in S} \mathbf{x}_{t+1}(e) \leq r(S)$. Recall that $r(S) = \sum_{i=1}^{c} \min(|S \cap V_i|, k_i)$ and

$$\sum_{e \in S} \mathbf{x}_{t+1}(e) = \sum_{e \in S \cap V_1} \mathbf{x}_{t+1}(e) + \cdots + \sum_{e \in S \cap V_c} \mathbf{x}_{t+1}(e) \tag{22}$$

We show each term $\sum_{e \in S \cap V_i} \mathbf{x}_{t+1}(e) \leq \min(|S \cap V_i|, k_i)$. First suppose $\min(|S \cap V_i|, k_i) = |S \cap V_i|$. Then since $\mathbf{x}_{t+1}(e) \in [0,1]$ we have $\sum_{e \in S \cap V_i} \mathbf{x}_{t+1}(e) \leq \sum_{e \in S \cap V_i} 1 = |S \cap V_i|$. Second, suppose $\min(|S \cap V_i|, k_i) = k_i$. Note $\sum_{e \in S \cap V_i} \mathbf{x}_{t+1}(e) \leq \sum_{e \in V_i} \mathbf{x}_{t+1}(e)$, hence it suffices to show $\sum_{e \in V_i} \mathbf{x}_{t+1}(e) \leq k_i$.

In this case we use the induction hypothesis that $\mathbf{x}_t \in \mathcal{B}$. $\mathbf{x}_t$ being in $\mathcal{B}$ implies that $\sum_{e \in V_i} \mathbf{x}(e) \leq r(V_i) = k_i$. By definition we have

$$a_t \sum_{e \in V_i} \mathbf{1}_{S_t}(e) + (1 - a_t) \sum_{e \in V_i} \mathbf{x}_{t+1}(e) = \sum_{e \in V_i} \mathbf{x}_t(e) \leq k_i \tag{23}$$

$$\Rightarrow a_t |S_t \cap V_i| + (1 - a_t) \sum_{e \in V_i} \mathbf{x}_{t+1}(e) \leq k_i \tag{24}$$

$$\Rightarrow a_t k_i + (1 - a_t) \sum_{e \in V_i} \mathbf{x}_{t+1}(e) \leq k_i \qquad (|S_t \cap V_i| = k_i \text{ by the choice of } S_t) \tag{25}$$

$$\Rightarrow (1 - a_t) \sum_{e \in V_i} \mathbf{x}_{t+1}(e) \leq (1 - a_t)k_i \tag{25}$$

$$\Rightarrow \sum_{e \in V_i} \mathbf{x}_{t+1}(e) \leq k_i \tag{26}$$

This finishes the proof that $\sum_{e \in S} \mathbf{x}_{t+1}(e) \leq r(S)$, thus $\mathbf{x}_{t+1} \in \mathcal{B}$.

Algorithm 2 terminates in at most $O(n)$ iterations. At each iteration $t$, $\mathbf{x}_{t+1}$ differs from $\mathbf{x}_t$ by fixing one additional coordinate to either 0 or 1. Once a coordinate is fixed to 0/1, it remains unchanged in all future iterates. The process terminates when $\mathbf{x}_t$ has $n - \sum_{i=1}^{c} k$ zeros and $\sum_{i=1}^{c} k_i$ ones, meaning all coordinates are 0/1.

Finally, because each $a_t$ is defined via pointwise minima and maxima of the entries of $\mathbf{x}_t$, it is almost everywhere differentiable in $\mathbf{x}_t$. By the chain rule, the resulting probability weights

$$p_{\mathbf{x}_t}(S_t) = a_t \prod_{i < t} (1 - a_i)$$

are almost-everywhere differentiable in the original input $\mathbf{x}_0$. This completes the proof.

$\square$

---

**Algorithm 2** Decomposition for partition matroid

---

**Require:** $\mathbf{x} \in \mathcal{B}$.
1: $\mathbf{x}_0 \leftarrow \mathbf{x}$
2: **repeat**
3:     $\mathbf{1}_{S_t} = \operatorname{argmax}_{\mathbf{c} \in \{0,1\}^n, \|\mathbf{c}(V_i)\|_1 = k_i} \mathbf{x}_t^\top \mathbf{c}$
4:     $a_t = \min\{\min_{i \in S_t} \mathbf{x}_t(i), 1 - \max_{i \notin S_t} \mathbf{x}_t(i)\}$
5:     $\mathbf{x}_{t+1} \leftarrow (\mathbf{x}_t - a_t \mathbf{1}_{S_t}) / (1 - a_t)$
6:     $p_{\mathbf{x}_t}(S_t) \leftarrow a_t \prod_{i=0}^{t-1} (1 - a_i)$
7:     $t \leftarrow t + 1$
8: **until** $\mathbf{x}_t \in \{0, 1\}^n$
9: **return** All $\{(p_{\mathbf{x}_t}(S_t), S_t)\}$ pairs.

---

### E.1.1 Generating neural predictions in the Partition Matroid Base Polytope

Similar to the cardinality constraint case, we need to map the output of the neural network to the polytope in a differentiable and computationally efficient way so that it can be passed to Algorithm 2. The general idea is similar to the one we proposed in Section 4.2.1, however it requires a more careful way of dealing with blocks.

**Proposition E.2** (Block-wise scaling into a partitioned simplex). *Let the index set $[n] = \{1, \ldots, n\}$ be partitioned into disjoint blocks $V_1, \ldots, V_c$ with $|V_i| = n_i$ and $\sum_{i=1}^{c} n_i = n$. Given any vector $\mathbf{z} \in [0, 1]^n$, set*

$$m_i := \frac{1}{n_i} \sum_{e \in V_i} \mathbf{z}(e), \qquad \mathbf{m}(e) := m_i \ (e \in V_i), \qquad \mathbf{u}(e) := \frac{k_i}{n_i} \ (e \in V_i),$$

$$s_i := \min\left(\frac{k_i/n_i}{m_i}, \frac{(n_i - k_i)/n_i}{1 - m_i}\right), \qquad \mathbf{s}(e) := s_i \ (e \in V_i),$$

*and define*

$$\mathbf{x} := \mathbf{s} \odot (\mathbf{z} - \mathbf{m}) + \mathbf{u}.$$

*Then $\mathbf{x} \in \mathcal{B}$; that is, every coordinate of $\mathbf{x}$ lies in $[0, 1]$ and each block $V_i$ has the prescribed sum $k_i$.*

*Proof.* For each block $V_i$ and every $e \in V_i$ we have

$$\mathbf{x}(e) = s_i\big(\mathbf{z}(e) - m_i\big) + \frac{k_i}{n_i}.$$

Since $\sum_{e \in V_i}(\mathbf{z}(e) - m_i) = 0$, summing over $V_i$ gives

$$\sum_{e \in V_i} \mathbf{x}(e) = k_i.$$

Because $-m_i \leq \mathbf{z}(e) - m_i \leq 1 - m_i$,

$$\mathbf{x}(e) \geq \frac{k_i}{n_i} - s_i m_i, \qquad \mathbf{x}(e) \leq \frac{k_i}{n_i} + s_i(1 - m_i).$$

By the definition $s_i = \min\big(\frac{k_i/n_i}{m_i}, \frac{(n_i-k_i)/n_i}{1-m_i}\big)$ both right–hand sides lie in $[0, 1]$, so $0 \leq \mathbf{x}(e) \leq 1$.

Thus every block sums to $k_i$ and every coordinate is in $[0, 1]$; hence $\mathbf{x} \in \mathcal{B}$. $\qquad\square$

## E.2   Graphical Matroid

In this subsection we turn our attention to *graphical matroids* and show that our framework can be extended beyond cardinality constraints. Graphical matroids are core combinatorial objectives that have been studied for decades in combinatorial optimization. Some of the most basic combinatorial problems such as finding a minimum spanning tree in a connected graph to more complex problems like "does a graph $G$ contain $k$ disjoint spanning trees?" which appears in many applications in Network Theory.

Let $G = (V, E)$ be a graph with $n$ nodes and $m$ edges. The type of optimization problems that we consider here are expressed as follows

$$\max_{S \subseteq E : S \in \mathcal{F}} f(S) \tag{27}$$

where $\mathcal{F}$ denotes the set of full *spanning forests* of $G$. We focus on the convex hull formed by the feasible sets edges in $\mathcal{F}$. That is, let $\mathbf{1}_T$ be the indicator vector of $T \in \mathcal{F}$ and define

$$\mathcal{B}(G) = \mathtt{conv}\{\mathbf{1}_T : T \in \mathcal{F}\} \tag{28}$$

The combinatorial object $\mathcal{B}(G)$ is known as the *graphical matroid base polytope*. For any set of edges let the rank be $r(S) = n - c(S)$ where $c(S)$ is the number of connected components of the subgraphs formed by the edges in $S$. For instance, if $G$ is connected and $S$ forms a spanning tree then $r(S) = n - 1$. It is known that $\mathcal{B}(G)$ can be defined in an equivalent way as follows

$$\mathcal{B}(G) = \{\mathbf{x} \in [0, 1]^m : \forall S \subseteq E, \ \mathbf{x}(S) \leq r(S), \ \text{and} \ \|\mathbf{x}\|_1 = n - c(V)\}, \tag{29}$$

where we use the notation $\mathbf{x}(S) := \sum_{e \in S} \mathbf{x}(e)$ for every subset $S$. Note that any vector $\mathbf{x} \in \mathcal{B}(G)$ can be written as a convex combination of the indicator vectors of feasible sets i.e, indicator vectors of spanning forests. We follow our general template to design a decomposition algorithm for the graphical matroid (Algorithm 3). We can then prove the following.

**Theorem E.3.** *Given $\mathbf{x} \in \mathcal{B}(G)$, Algorithm 3 terminates after at most $O(m)$ iterations and returns $\{(p_{\mathbf{x}_t}(S_t), S_t)\}$ such every $S_t$ corresponds to a spanning forest in $G$ and $\sum_t p_{\mathbf{x}_t}(S_t) = 1$ with $p_{\mathbf{x}_t}(S_t) \in [0, 1]$. Moreover, each $p_{\mathbf{x}_t}(S_t)$ is an almost everywhere differentiable function of $\mathbf{x}$.*

*Proof.* The algorithm follows a similar template as our main result, the only difference is on selecting an appropriate vertex of the polytope at each step and selecting an appropriate coefficient $a_t$. Specifically in each iteration $t$ we define

$$\mathbf{x}_{t+1} \leftarrow (\mathbf{x}_t - a_t \mathbf{1}_{S_t}) / (1 - a_t)$$

---

**Algorithm 3** Decomposition for graphical matroid

---

**Require:** Graph $G = (V, E)$ and $\mathbf{x} \in \mathcal{B}(G)$.

1: $\mathbf{x}_0 \leftarrow \mathbf{x}$
2: **repeat**
3:    $S_t \leftarrow$ A maximum spanning forest using $\mathbf{x}_t$ as the weights for edges (treat zero entries as non-edges.)
4:    $a_t = \min \left\{ \min_{e \in S_t} \mathbf{x}(e),\, 1 - \max_{e \notin S_t} \mathbf{x}(e),\, \min_{F \subseteq E} \frac{r(F) - \mathbf{x}_t(F)}{r(F) - |S_t \cap F|} \right\}$
5:    $\mathbf{x}_{t+1} \leftarrow (\mathbf{x}_t - a_t \mathbf{1}_{S_t}) / (1 - a_t)$
6:    $p_{\mathbf{x}_t}(S_t) \leftarrow a_t \prod_{i=0}^{t-1} (1 - a_i)$
7:    $t \leftarrow t + 1$
8: **until** $\mathbf{x}_t \in \{0, 1\}^m$
9: **return** All $\{(p_{\mathbf{x}_t}(S_t), S_t)\}$ pairs.

---

where $S_t$ is a maximum spanning forest using $\mathbf{x}_t$ as the weights for edges (treat zero entries as non-edges). In order to guarantee that $\mathbf{x}_{t+1}$ is in the polytope $\mathcal{B}(G)$ it must satisfy the rank inequalities and be in $[0, 1]^m$. To guarantee $\mathbf{x}_{t+1} \in [0, 1]^m$ we enforce that $a_t \leq \min \{\min_{e \in S_t} \mathbf{x}(e), 1 - \max_{e \notin S_t} \mathbf{x}(e)\}$. To satisfy the rank constraints, for every $F \subseteq E$ we must guarantee $\mathbf{x}_{t+1}(F) \leq r(F)$. That is

$$
\begin{aligned}
\mathbf{x}_{t+1}(F) \leq r(F) &\iff (\mathbf{x}_t(F) - a_t \mathbf{1}_{S_t}(F)) / (1 - a_t) \leq r(F) \\
&\iff \mathbf{x}_t(F) - r(F) \leq a_t(|S_t \cap F| - r(F)) \\
&\iff a_t \leq \frac{r(F) - \mathbf{x}_t(F)}{r(F) - |S_t \cap F|}
\end{aligned}
$$

Note that $r(F) > |S_t \cap F|$ and we can dismiss the case where $r(F) = |S_t \cap F|$. This must hold for all subsets $F \subseteq E$. We therefore set

$$
a_t = \min \left\{ \min_{e \in S_t} \mathbf{x}(e),\, 1 - \max_{e \notin S_t} \mathbf{x}(e),\, \min_{F \subseteq E} \frac{r(F) - \mathbf{x}_t(F)}{r(F) - |S_t \cap F|} \right\} \tag{30}
$$

It is not immediate if we can compute $a_t$ in polynomial-time as the third term in Equation (30) is troublesome and involves minimization over all the $2^m$ subsets. There however exists a polynomial time algorithm to compute $a_t$. For a fixed $\lambda \in [0, 1]$ define the set function $g_{(\lambda, t)} : 2^m \to \mathbb{R}$

$$
g_{(\lambda, t)}(F) := (1 - \lambda) r(F) - \mathbf{x}_t(F) + \lambda |F \cap S_t| \tag{31}
$$

Then $\mathbf{x}_{t+1}$ is in the polytope iff:

1. $a_t \leq \min \{\min_{e \in S_t} \mathbf{x}(e), 1 - \max_{e \notin S_t} \mathbf{x}(e)\}$, and

2. $\min_{F \subseteq E} g_{(a_t, t)}(F) \geq 0$.

To this end, our goal is to find the largest $\lambda^* \in [0, \max\{\min_{e \in S_t} \mathbf{x}(e), 1 - \max_{e \notin S_t} \mathbf{x}(e)\}]$ for which $\min_{F \subseteq E} g_{(\lambda^*, t)}(F) \geq 0$. Then $a_t = \min \{\min_{i \in S_t} \mathbf{x}(i), 1 - \max_{i \notin S_t} \mathbf{x}(i), \lambda^*\}$.

We find such $\lambda^*$ by doing a binary search over the interval. To justify our binary search we need two things. First is monotonicity of $\phi(\lambda) := g_{(\lambda, t)}(F)$ w.r.t $\lambda$ and any fixed $F$. Second is computing $\min_{F \subseteq E} g_{(\lambda, t)}(F)$ for a fixed $\lambda$.

**Claim E.4.** *For a fixed $F$, $\phi(\lambda) := g_{(\lambda, t)}(F)$ is a nonincreasing function in $\lambda$.*

*Proof.* Note that $(1 - \lambda) r(F) - \mathbf{x}_t(F) + \lambda |F \cap S_t| = r(F) - \mathbf{x}_t(F) + \lambda(|F \cap S_t| - r(F))$ is an affine function of $\lambda$ with negative slope since $r(F) > |S_t \cap F|$. This completes the proof. $\qquad \square$

**Claim E.5.** *There is a polynomial time algorithm, i.e. efficient oracle, that for a fixed $\lambda$ returns $\min\limits_{F \subseteq E} g_{(\lambda, t)}(F)$.*

*Proof.* We point out that the set function $g_{(\lambda, t)}$ for a fixed $\lambda$ is a submodular function. This is because the rank function $r(F)$ is submodular, $-\mathbf{x}_t(F)$ and $\lambda |F \cap S_t|$ are modular functions. Hence, $\min_{F \subseteq E} g_{(\lambda, t)}(F)$ can be found in strongly polynomial time by any submodular-function minimization (SFM) algorithm, for example the one provided in [76]. $\square$

In the binary search, at a fixed $\lambda$ using the efficient oracle we compute $\min_{F \subseteq E} g_{(\lambda, t)}(F)$. If this value is greater than zero, we increase $\lambda$ and continue the binary search in the right half interval. Else, we decrease $\lambda$ and continue the binary search in the left half integral. (Note that this approach finds $\lambda^*$ up to a desired precision $\varepsilon \geq 0$.)

Finally, we argue that $a_t$ as defined in Equation (30) is an a.e. differentiable function with respect to $\mathbf{x}_t$. The proof is similar to what we have seen in the cardinality cases. First note that for a fixed maximum spanning tree $S_t$, $a_t$ is the minimum of finitely many affine functions of $\mathbf{x}_t$; hence, it is a piecewise linear function and almost everywhere differentiable. Second, the map that outputs a maximum weight spanning tree with respect to $\mathbf{x}_t$ is almost everywhere constant (the resulting spanning tree changes only if some edge weights are exactly equal.). These two together yield that $a_t$ as defined in Equation (30) is an a.e. differentiable function with respect to $\mathbf{x}_t$.

$\square$

### E.2.1 Generating neural predictions in the Graphical Base Matroid

We need to massage the output of the neural network in a differentiable and computationally efficient way so that it obeys a relaxed constraint and can be passed to Algorithm 3. Algorithm 3 requires $\mathbf{x}$ to lie within the graphical matroid base polytope $\mathcal{B}(G)$, which represents a relaxed version of the original constraints. Without loss og generality we assume $G$ is a connected graph. Every point inside the graphical matroid base polytope can be seen as a distribution over spanning trees. We first define the uniform distribution over the set of all spanning trees and forces our encoder network to generate a perturbation to this uniform distribution. Some notations are in order.

Fix an arbitrary orientation over the edges of $G$. For an edge $e = (u, v)$, i.e., oriented from $u$ to $v$, let

$$b_e = \mathbf{1}_u - \mathbf{1}_v \text{ with } \mathbf{1}_u(v) = \begin{cases} 1 \text{ if } v = u \\ 0 \text{ otherwise} \end{cases} \tag{32}$$

The Laplacian of (positively) weighted graph $G$ is $L_G = \sum_{e \in E} w(e) b_e b_e^\top$. Let $L_G^\dagger$ denote the Moore–Penrose pseudo-inverse of $L_G$.

**Generating a perturbation vector.** Let $\mathcal{T}$ denote the set of all spanning trees of $G$ and $\mu$ be a uniform distribution of all spanning trees of $G$. A uniform spanning tree distribution is a uniform distribution over all spanning trees of a given graph. If the graph is weighted, then we can study the weighted uniform distribution of spanning trees where the probability of each tree is proportional to the product of the weight of its edges. For $w : E \to \mathbb{R}_+$, we say $\mu(w)$ is a $w$-uniform spanning tree distribution, if for any spanning tree $T \in \mathcal{T}$,

$$\Pr[T] \propto \prod_{e \in T} w(e).$$

Let $\mu_e(w) := \Pr_T \sim \mu[e \in T]$ be the marginal probability of edge $e$. The following result gives us a way of writing an analytical expression for $\mu_e$.

**Theorem E.6** ([81]). *For any edge $e$*

$$\mu_e = b_e^\top L_G^\dagger b_e.$$

*Moreover, for weighted graph $G$ with $w : E \to \mathbb{R}_+$, if $w(e)$ is the weight of $e$, then*

$$\mu_e(w) = w(e) b_e^\top L_G^\dagger b_e.$$

The above Theorem E.6 suggests the following way of generating a point inside the graphical matroid base polytope. Let $w \in \mathbb{R}_+^E$ be the output of an encoder network that assigns a positive weight to every edge of graph $G$. We think of $w$ as a perturbation to the uniform distribution over the spanning trees. Then obtain vector $\mathbf{x}$ whose $e$-th entry is $\mu_e(w)$ according to the weight vector $w$ and Theorem E.6. The vector $\mathbf{x}$ lies inside the graphical matroid base polytope $B(G)$. Recall that $\mathcal{B}(G) = \mathrm{conv}\{\mathbf{1}_T : T \in \mathcal{T}\}$ and $\mathbf{x} = \mathbb{E}_{T \sim \mu(w)}[\mathbf{1}_T]$ is a convex combination of indicator vector of the spanning trees. We show that $\mu(w)$ is continuous and differentiable with respect to $w$.

**Theorem E.7.** *Let $G = (V, E, w)$ be a connected, weighted graph with non–negative edge weights $w : E \to \mathbb{R}_{\geq 0}$ and let*

$$\mu_e(w) \;=\; \mathbb{P}_{T \sim \mu(w)}[\,e \in T\,] \;=\; w(e)\, b_e^\top L_G^\dagger(w)\, b_e, \qquad e \in E,$$

*where $b_e$ is the signed incidence vector of $e$ and $L_G(w) = \sum_{f \in E} w(f)\, b_f b_f^\top$ is the weighted Laplacian (with Moore–Penrose pseudoinverse $L_G^\dagger$). Then*

   *i. $\mu_e : \mathbb{R}_{\geq 0}^E \to \mathbb{R}$ is continuous on the closed orthant;*

   *ii. $\mu_e$ is $C^\infty$ (indeed analytic) on the open orthant $\mathbb{R}_{>0}^E$;*

   *iii. $\mu_e$ is differentiable everywhere on $\mathbb{R}_{\geq 0}^E$.*

*Proof.* Let $\widetilde{L}_G(w)$ denotes the reduced Laplacian with any single row/column removed. By Kirchhoff's Matrix–Tree Theorem [41] we have

$$\tau(w) \;=\; \det\!\big(\widetilde{L}_G(w)\big) \;=\; \sum_{\text{spanning trees } T} \; \prod_{f \in T} w(f),$$

For an edge $e$ we have,

$$\mu_e(w) \;=\; 1 - \frac{\tau(w - e)}{\tau(w)} \tag{33}$$

where $w - e$ denotes the weight vector with $w(e) = 0$. Both numerator and denominator in (33) are (multivariate) polynomials; moreover $\tau(w) > 0$ whenever at least one spanning tree has positive total weight, which is guaranteed by the connectivity of $G$. Hence the ratio is continuous on $\mathbb{R}_{\geq 0}^E$, proving part (i).

On the interior $\mathbb{R}_{>0}^E$ the reduced Laplacian $\widetilde{L}_G(w)$ is positive definite (PD). The maps

$$w \longmapsto \widetilde{L}_G(w) \quad \text{(affine map)}, \qquad A \longmapsto A^{-1} \quad (C^\infty \text{ on PD}),$$

compose smoothly, so the formula $\mu_e(w) = w(e)\, b_e^\top \widetilde{L}_G(w)^{-1} b_e$ is infinitely differentiable i.e., is $C^\infty$ and indeed real-analytic on $\mathbb{R}_{>0}^E$, giving (ii).

It remains to show (iii) at boundary points where some coordinates of $w$ equal 0. Let $w_0$ be a weight vector with at least one zero entry, and $Z$ be the set of zero weight edges. There are two cases:

Case A: $G - Z$ is connected. In this case the reduced Laplacian $\widetilde{L}_G(w_0)$ positive definite since $G - Z$ has at least one spanning tree, so the same smoothness argument used for (ii) applies; thus $\mu_e(w_0)$ is differentiable (in fact is $C^\infty$).

Case B: $G - Z$ is disconnected. Then every spanning tree of any nearby weight vector must contain the cut–edge whose weight vanished, so the term in Equation (33) equals 1 in a whole neighbourhood; hence $\mu_e$ is locally constant and therefore differentiable at $w_0$ with derivative 0.

In either case one-sided (classical) derivatives exist and coincide with the interior derivative, completing (iii). $\qquad \square$

# F   Case study: Maximum Independent Set

For the maximum independent set problem, we want to find the largest set $S$ of nodes in a graph $G = (V, E)$, where no pair of nodes in the set is adjacent. Unfortunately, there is no known

compact description in terms of inequalities for the independent set polytope [55]. This immediately presents a challenge for the design of the decomposition. Furthermore, linear optimization over the independent set polytope is NP-Hard since it amounts to solve the the maximum independent set problem. Therefore, in this case, we will consider a relaxation of the polytope which has a known description in terms of inequalities and for which the optimization problem can be solved efficiently.

First, let us assume that $\emptyset \in \mathcal{C}$, and that the graph $G$ has no isolated nodes. In order to be able to optimize over independent sets we are going to need access to a polytope description. The constraints $0 \leq x_i \leq 1$ for all $i$ and $\mathbf{x}(i) + \mathbf{x}(j) \leq 1$ for $(i, j) \in E$ are defining the polytope referred to in the literature as FSTAB [55]. FSTAB is a relaxation of the independent set polytope since it includes half-integral vertices. We will explain how we will leverage FSTAB to explore the space of independent sets.

For step 3 we use Equation (10) to obtain a vertex of the polytope FSTAB. Note that now the vertex may contain half-integral coordinates. This will affect our coefficient selection. We need to select a coefficient that does not violate the polytope inequalities. The constraints $0 \leq \mathbf{x} \leq 1$ imply the same rule for the coefficient $a_t$. However, we have an additional consideration. If we just choose the coefficients as in the cardinality constraint case, when the coordinates $i, j \notin S$ and $(i, j) \in E$ get rescaled by $1 - a_t$, we may break the constraint for $\mathbf{x}(i) + \mathbf{x}(j) \leq 1$. This leads us to the following result:

**Theorem F.1.** *Let*

$$a_t = \min \left\{ \underbrace{\min_{i \in S_t} \mathbf{x}_t(i)}_{\mathbf{x}(i) \geq 0}, \quad \underbrace{\min_{j \notin S_t} \left(1 - \mathbf{x}_t(j)\right)}_{\mathbf{x}(j) \leq 1}, \quad \underbrace{\min_{(u,v) \in E: \, u,v \notin S_t} \left(1 - \mathbf{x}_t(u) - \mathbf{x}_t(v)\right)}_{\mathbf{x}(u) + \mathbf{x}(v) \leq 1} \right\} \tag{34}$$

*be the coefficient chosen at each iteration in the decomposition. Algorithm 1 on FSTAB using coefficients from Equation (34) and Equation (10) for step 3, obtains an exact decomposition in at most $n + 1$ steps, and each coefficient is an a.e. differentiable function of its corresponding iterate in the decomposition.*

*Proof.* At each step, a coordinate is set to either 0, 1, or an edge inequality is tightened. Since we use the same recurrence as with the cardinality constraint, if at any iteration a coordinate is set to 0 or 1, then it cannot change. For the edge inequalities, suppose we have $\mathbf{x}_{t+1}(u) + \mathbf{x}_{t+1}(v) = 1$, and $\mathbf{v}_{t+1}(u) + \mathbf{v}_{t+1}(v) = 1$ where $\mathbf{v}_{t+1}$ is the vertex we obtain by the oracle at iteration $t + 1$. Thus

$$\mathbf{x}_{t+2}(u) + \mathbf{x}_{t+2}(v) = \frac{\mathbf{x}_{t+1}(u) + \mathbf{x}_{t+1}(v) - a_{t+1}\left(\mathbf{v}_{t+1}(u) + \mathbf{v}_{t+1}(v)\right)}{1 - a_{t+1}} \tag{35}$$

$$= \frac{1 - a_{t+1} \cdot 1}{1 - a_{t+1}} \tag{36}$$

$$= 1, \tag{37}$$

so the edge equality persists. Each time we tighten an inequality, the dimension of the minimal face containing the current iterate is decreased by one. Thus, in at most $n$ steps the algorithm will land on a vertex (face of dimension 0) of the polytope, at which point the algorithm terminates. Finally, the coefficients are always a.e. differentiable functions of $\mathbf{x_t}$ since the minimum function over $\mathbf{x}_t$ is also everywhere differentiable. $\square$

**Obtaining a point on FSTAB.** Since we would like to build a support that avoids half-integral points as much as possible we follow a simple gradient-based projection scheme for FSTAB. We begin by defining the total edge-violation of a point $\mathbf{x} \in \mathbb{R}^n$ with slack parameter $s \geq 0$ as

$$V(\mathbf{x}) = \sum_{(i,j) \in E} \max\left\{0, \, \mathbf{x}(i) + \mathbf{x}(j) + s - 1\right\},$$

which measures how much each edge $(u, v)$ exceeds the bound $\mathbf{x}(i) + \mathbf{x}(j) + s \leq 1$. To correct all violations at once, we compute $\nabla V(\mathbf{x}) = (d_1, \ldots, d_n)$, where

$$d_i = \sum_{\{j: \, (i,j) \in E\}} \mathbf{1}\left[\mathbf{x}(i) + \mathbf{x}(j) + s - 1 > 0\right]$$

counts how many incident edges on node $i$ are violated. Next, for each violated edge $(u, v)$ with violation amount

$$b_{uv} = \mathbf{x}(u) + \mathbf{x}(v) + s - 1 > 0,$$

we require a step-size $\eta$ satisfying $b_{uv} - \eta \, (d_u + d_v) \leq 0$, which implies

$$\eta \geq \frac{b_{uv}}{d_u + d_v}.$$

Taking the maximum of these ratios over all violated edges gives the minimal $\eta$ that fixes every violation:

$$\eta = \max_{(u,v) \, : \, \mathbf{x}(u) + \mathbf{x}(v) + s > 1} \frac{\mathbf{x}(u) + \mathbf{x}(v) + s - 1}{d_u + d_v}.$$

Finally, we compute $\mathbf{x}' = \sigma(\mathbf{x} - \eta \, \nabla V(\mathbf{x}))$, where $\sigma$ is a ReLU. This guarantees $\mathbf{x}'(u) + \mathbf{x}'(v) + s \leq 1$ for every $(u, v) \in E$ and $\mathbf{x}' \geq 0$, which ensures that we have a point in FSTAB.

**Dealing with half-integral corners.** The decomposition may still yield several half-integral corners which can affect the performance of our extension, since we need the support sets to be independent. There are two ways of dealing with half integral corners. One approach is to penalize them in the objective i.e., the function evaluated at half-integral corners returns 0. Another approach is by tightening the relaxation with additional inequalities commonly found in the literature, including clique constraints and odd cycle constraints [55]. In practice, many of those come down to imposing an L1 norm constraint on the point in the polytope. We may enforce both constraints (FSTAB and L1) by alternating projections. It is easy to see that if the norm is 1 and the constraints are all satisfied, then we can always obtain a decomposition into independent sets, so a valid decomposition always exists.

## G   Experiments

Here, we provide detailed experimental settings and some additional experimental results. All the datasets are public and our code is available at `https://github.com/frankx2023/Neural_Combinatorial_Optimization_with_Constraints`.

**Hardware.**   All experiments are run on 16 cores (32 threads) of Intel(R) Xeon(R) Platinum 8268 CPU (24 cores, 48 threads in total), 32 GB ram, with a single Nvidia RTX8000 48GB GPU.

**Datasets.**   Following previous work [10, 88], we evaluate our methods on both synthetic and real-world bipartite graphs $(U, V, E)$, $V$ is the ground set and the goal is to select $k$ nodes from $V$ so that we cover maximum number of nodes from $U$.

- **Random Uniform.** The *Random Uniform* datasets include both the `Random500` and `Random1000` settings, and are used for both training and testing. Each dataset consists of 100 independently generated bipartite graphs, where $|V| = 500$, $|U| = 1000$ (`Random500`) or $|V| = 1000$, $|U| = 2000$ (`Random1000`). Each $u \in U$ is assigned a weight uniformly at random from 1 to 100. Each $v \in V$ covers a random subset of $U$, with the number of covered nodes for each $v$ chosen uniformly at random between 10 and 30.

- **Random Pareto.** The `Random Pareto` dataset consists of 100 independently generated bipartite graphs with $|U| = |V| = 1000$, used exclusively for training in the Twitch experiments. In each graph, the number of covered nodes per covering node $v \in V$ follows a Pareto distribution, with the $\alpha$ parameter randomly selected between 1 and 2, resulting in a heavy-tailed degree distribution (typically, 20% of covering nodes account for 80% of the edges). Each $u \in U$ is assigned a weight uniformly at random from 1 to 100, and every $u \in U$ is covered by at least one $v \in V$.

- **Twitch.** The Twitch datasets model social networks of streamers grouped by language, with $|U| = |V|$ set to the number of streamers. The dataset includes: DE ($|U| = |V| = 9498$), ENGB (7126), ES (4648), FR (6549), PTBR (1912), and RU (4385). The objective is to maximize the sum of logarithmic viewer counts over $U$.

- **Railway.** The Railway datasets are derived from real-world Italian railway crew assignments. We evaluate on three graphs: rail507 ($|V| = 507$, $|U| = 63009$), rail516 ($|V| = 516$, $|U| = 47311$), and rail582 ($|V| = 582$, $|U| = 55515$).

**Baselines.** We adopt the same baselines as prior work, including:

- **Random.** Samples $k$ candidates uniformly at random over multiple trials, selects the best within 240 seconds.

- **Greedy algorithm [61].** Iteratively adds the element with the largest marginal gain, achieving a $(1 - 1/e)$ approximation.

- **Gurobi.** Exact MIP solver, time-limited to 120 seconds per instance. The version used is Gurobi 12.01.

- **EGN [35].** Optimizes a probabilistic objective with naive rounding; does not guarantee feasibility during optimization.

- **CardNN [88].** One-shot neural solvers for cardinality-constrained problems. Main variants: CardNN-S (Sinkhorn), CardNN-GS (Gumbel-Sinkhorn), CardNN-HGS (Homotopy Gumbel-Sinkhorn). For each, a CardNN-noTTO variant omits test-time optimization.

- **UCOM2 [10].** Combinatorial optimization using greedy incremental derandomization. Three variants differ only by test-time augmentation and running time: UCOM2-short (no augmentation, fastest), UCOM2-middle (moderate augmentation, medium time), UCOM2-long (maximal augmentation, longest time).

- **RL.** For the RL baseline, we follow the instructions in [45, 69] to use GraphSage layers [28] as policy network with Actor-Critic [44] algorithm to train on the same problem instances.

## G.1 Training

To enhance optimization stability and solution quality, we incorporate several training techniques.

A rescaling coefficient in Appendix C.1 is referred to as a *scaling factor*. A set of *scaling factors* :$[1.0, 0.8, 0.6, 0.4, 0.2, 0.1, 0.05, 0.02, 0.01]$ are used in the training and inference parallelly, helping to get stable decompositions across different data distributions.

An *entropy regularization* term is applied with a cosine-decayed weight $\lambda_t$, initialized at $0.05$ and annealed to zero by the 30th epoch, promoting exploration within the hypersimplex.

Convergence speed is controlled via a *sharpness factor* applied to the noise perturbation layer when the sigmoid activation function is used in that layer (when using min-max scaling, it is ignored). The sharpness factor is linearly increased from $0.3$ at the onset of training to $1.0$ by epoch 80.

Gaussian noise is injected into the model logits throughout training to foster exploration and enhance robustness. The standard deviation of this noise is initially set to $0.05$ and is reduced to zero according to a cosine decay schedule as training progresses.

The other hyperparameters are listed below:

- Dropout: $0.1$ (applied only during training).
- Optimizer: AdamW.
- Learning Rate: $5 \times 10^{-3}$.
- Learning Rate Scheduler: `warmup_cosine`, with $50$ warmup epochs, decaying to a minimum of $5 \times 10^{-5}$.
- Weight Decay: $1 \times 10^{-4}$.
- Epochs: $80$.
- Batch Size: $4$.
- Seed: $42$.
- Workers: 16 CPU worker threads for data loading, preprocessing, and decomposition with multiple scaling factors.

## G.2 Inference strategies and local improvement

We evaluate our model using three distinct inference strategies—*Short*, *Medium*, and *Long*—each balancing computational efficiency and local improvement:

**Local improvement.** To further refine the candidate solution, we employ a local search algorithm based on iterative element replacement. At each step, the algorithm considers replacing a single element in the current solution set with one from outside the set (within a defined candidate pool), accepting only replacements that yield a strictly improved objective value. The procedure terminates when no improving replacements are found or when the maximum iteration count is reached.

- **Short:** For each instance, 50 decomposed sets are generated from each scaling factor and the best among them is chosen as the result, with no further local improvement applied.

- **Medium:** For each instance, 100 decomposed sets are generated from each scaling factor and the best among them is chosen as the base set. The base set then undergoes the local improvement procedure , performed for up to 10 iterations. The candidate pool consists of the first $k$ nodes produced by the scaling factor $0.1$ that is not present in the base set .

- **Long:** For each instance, 5 additional augmented graphs are generated (with 0–30% feature noise and 0–30% random edge dropout). For each graph, decomposed sets are generated from each scaling factor and the best among them is chosen as the base set. Local improvement is performed for up to $k$ iterations on the original graph, with the candidate pool includes all nodes that appears in decomposed sets across from the scaling factor where the base set is selected. The best solution found among all augmented graphs is selected as the final result for the instance.

### G.3 Ablation study: local improvement candidate pool selection

To demonstrate that the strategic selection of nodes in our candidate pool provides meaningful benefits, we conducted an ablation study comparing our decomposition-based candidate pool against a random alternative of equal size. In this ablation test, we maintain the same pre-local improvement solution (the base set) as the starting point for both approaches. The original candidate pool is constructed using our medium and long methods. The alternative candidate pool retains the same base set but replaces the remaining candidates with randomly selected nodes, ensuring both pools have identical size for fair comparison.

Table 1: Ablation study comparing decomposition-based candidate pool versus random alternative. Improvement percentages over the base set.

| Method | rand500 k=10 | | rand1000 k=20 | | rand500 k=50 | | rand1000 k=100 | |
| --- | --- | --- | --- | --- | --- | --- | --- | --- |
| | Medium | Long | Medium | Long | Medium | Long | Medium | Long |
| Decomposition-based (Ours) | 2.41% | 3.28% | 5.15% | 8.09% | 2.11% | 3.89% | 3.83% | 8.07% |
| Random Alternative | 0.23% | 1.02% | 3.03% | 5.74% | 0.43% | 2.39% | 2.84% | 5.79% |

We evaluate both local improvement strategies across medium and long inference modes. The results, Table 1, demonstrate that our strategic candidate selection provides consistent improvements over random selection, validating that our local improvement procedure benefits specifically from the candidate pool construction from our decomposition rather than simply having access to additional nodes for local improvement.

### G.4 Ablation study on short+Gurobi

We also tested two versions of the "short+Gurobi" approach on the random datasets, one with a time limit near our "medium" method, and for some settings, an version given the time close to our "long" method. When the instance size and the cardinality constraint are small (k = 10 on Random 500), "short+Gurobi" performs slightly better than other methods. For larger instances like Random 500 with k = 50 and Random 1000, our "medium" and "long" approaches show their advantages and achieved similar or better results, indicating that our local improvement technique effectively constrains the solution space and works better even compared to the well-optimized Gurobi Solver. In addition, the short+Gurobi method shows much better cost-efficiency compared to the pure Gurobi method, showing the potential of our method to serve as an efficient initial solution for structured solvers, significantly reducing overall computation time. The results of these tests can be found in Table 5 and Table 6.

## G.5 Ablation tests for UCOM2

As stated in Section 5.2, we categorize UCOM2 as a non-learning method and conduct several ablation studies to substantiate this classification. Specifically, we evaluate two variants: setting the neural network output to zero (UCOM2-zerostart-short), and to random uniform values between $0$ and $1$ (UCOM2-randomstart). Both ablations disregard the output of the neural network. We find that UCOM2-zerostart-short achieves results nearly identical to the standard short variant, while UCOM2-randomstart attains similar performance to all three UCOM2 variants, depending on the number of random samples generated. These results demonstrate that UCOM2's effectiveness is in large part due to their greedy module and the fact that the maximum coverage objective is a monotone submodular function.

Table 2: Ablation results for UCOM2 variants on Random500. Running time (time): smaller is better. Objective (obj): larger is better. Standard deviations capture variability.

| Method | Random500, $k = 10$ | | Random500, $k = 50$ | |
|---|---|---|---|---|
| | Time↓ | Obj↑ | Time↓ | Obj↑ |
| UCOM2-zerostart-short | $0.1089 \pm 0.0309$s | $15551.01 \pm 367.12$ | $0.0885 \pm 0.1032$s | $44311.87 \pm 819.96$ |
| UCOM2-randomstart-short | $0.8862 \pm 0.1238$s | $15294.04 \pm 408.52$ | $0.6957 \pm 0.1369$s | $44237.92 \pm 824.05$ |
| UCOM2-randomstart-medium | $3.8521 \pm 0.2770$s | $15586.95 \pm 360.89$ | $15.4367 \pm 0.0493$s | $44867.19 \pm 741.85$ |
| UCOM2-randomstart-long | $35.9862 \pm 2.5628$s | $15672.89 \pm 351.74$ | $30.5812 \pm 0.0850$s | $44923.74 \pm 754.78$ |
| UCOM2-short | $1.0411 \pm 0.0827$s | $15253.35 \pm 370.00$ | $0.7392 \pm 0.0150$s | $44208.95 \pm 768.68$ |
| UCOM2-medium | $4.1891 \pm 0.1150$s | $15589.25 \pm 363.43$ | $16.0005 \pm 0.0408$s | $44852.82 \pm 765.11$ |
| UCOM2-long | $39.2018 \pm 0.9474$s | $15779.45 \pm 358.41$ | $31.7587 \pm 0.0811$s | $44906.50 \pm 761.75$ |

Table 3: Ablation results for UCOM2 variants on Random1000. Running time (time): smaller is better. Objective (obj): larger is better. Standard deviations capture variability.

| Method | Random1000, $k = 20$ | | Random1000, $k = 100$ | |
|---|---|---|---|---|
| | Time↓ | Obj↑ | Time↓ | Obj↑ |
| UCOM2-zerostart-short | $0.2275 \pm 0.0164$s | $30991.19 \pm 528.97$ | $0.1682 \pm 0.0360$s | $88693.99 \pm 1248.72$ |
| UCOM2-randomstart-short | $1.6408 \pm 0.0329$s | $29878.53 \pm 601.10$ | $1.5426 \pm 0.0449$s | $88549.61 \pm 1342.89$ |
| UCOM2-randomstart-medium | $9.4358 \pm 0.0415$s | $30437.43 \pm 532.00$ | $8.8802 \pm 0.0522$s | $89077.84 \pm 1244.72$ |
| UCOM2-randomstart-long | $89.8093 \pm 0.3444$s | $30752.49 \pm 529.91$ | $84.3725 \pm 0.4054$s | $89386.56 \pm 1245.85$ |
| UCOM2-short | $1.7342 \pm 0.0677$s | $29791.66 \pm 678.02$ | $1.6216 \pm 0.0644$s | $88472.61 \pm 1261.36$ |
| UCOM2-medium | $9.5523 \pm 0.0620$s | $30420.69 \pm 552.66$ | $8.9956 \pm 0.0890$s | $89072.92 \pm 1248.58$ |
| UCOM2-long | $90.7475 \pm 0.3342$s | $30744.61 \pm 512.61$ | $85.2663 \pm 0.6164$s | $89408.28 \pm 1232.74$ |

On the twitch dataset, UCOM2 uses zero initialization (UCOM2-zerostart) in its original implementation, fully ignoring the trained model. To assess the impact of the neural network, we also evaluate UCOM2-withmodel on twitch, which yields markedly inferior results. This further supports our observation that all learning-based methods perform poorly on Twitch due to the architecture of the encoder network.

Table 4: Ablation results for UCOM2 variants on the Twitch dataset. Running time (time): smaller the better. Objective (obj): the larger the better. The standard deviations are captured to show the variability of the dataset and the method.

| Method | Twitch, $k = 20$ | | Twitch, $k = 50$ | |
|---|---|---|---|---|
| | Time↓ | Obj↑ | Time↓ | Obj↑ |
| Ucom2-withmodel-short | $73.2402 \pm 80.5553$s | $18.67 \pm 45.72$ | $115.8045 \pm 126.7061$s | $573.67 \pm 376.52$ |
| Ucom2-withmodel-medium | $333.3645 \pm 363.4380$s | $102.00 \pm 41.71$ | $532.9243 \pm 585.9112$s | $3018.17 \pm 5208.31$ |
| Ucom2-withmodel-long | $659.8775 \pm 719.8658$s | $150.00 \pm 48.88$ | $1054.5567 \pm 1160.6106$s | $3029.67 \pm 5202.21$ |
| Ucom2-zerostart-short | $17.8031 \pm 18.4783$s | $25853.00 \pm 1112.59$ | $17.3739 \pm 18.1940$s | $30526.00 \pm 13043.86$ |
| Ucom2-zerostart-medium | $19.6806 \pm 19.8771$s | $25858.17 \pm 11206.89$ | $21.6114 \pm 21.3478$s | $30544.17 \pm 13052.91$ |
| Ucom2-zerostart-long | $22.0589 \pm 21.6869$s | $25858.17 \pm 11206.89$ | $27.2170 \pm 25.6604$s | $30544.17 \pm 13052.91$ |

## G.6 Detailed results

Below are the plots and tables showing the full detailed test results. The raw numerical results are given in Table 5 to Table 8. The standard deviation are captured to show the variability of the dataset and the method.

Figure 3: Performance comparison of our method against baseline approaches across Learning without TTO, on multiple datasets. Metrics used are inference time (lower is better) and objective value (higher is better). In the Learning without TTO setting, our short version consistently outperforms all baselines in both inference time and objective value, demonstrating strong learning capability.

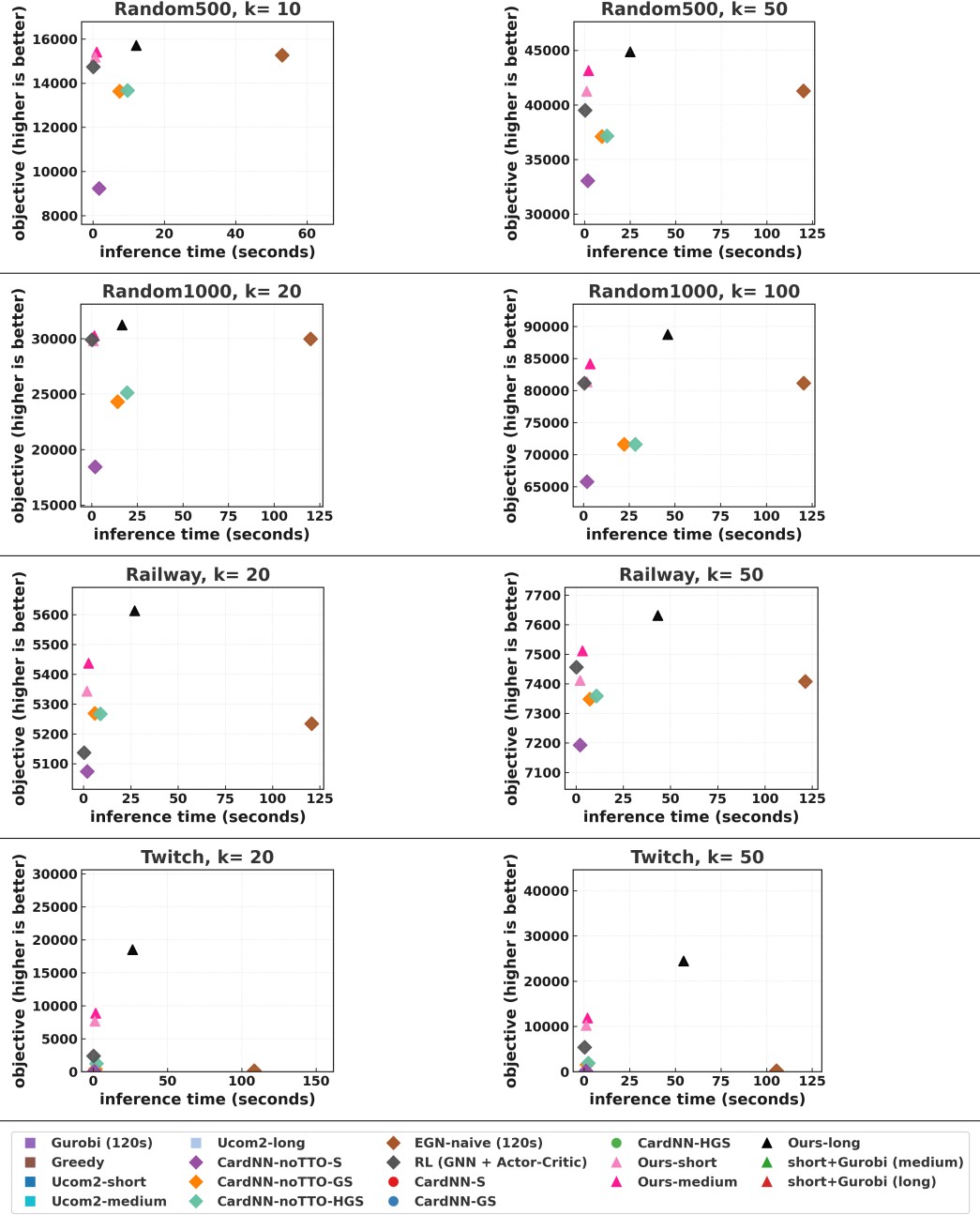

Figure 4: Performance comparison of our method against baseline approaches across Learning with TTO, on multiple datasets. Metrics used are inference time (lower is better) and objective value (higher is better). When extended to medium and long versions, our method surpasses most TTO-based baselines across datasets, with the exception of the Twitch dataset.

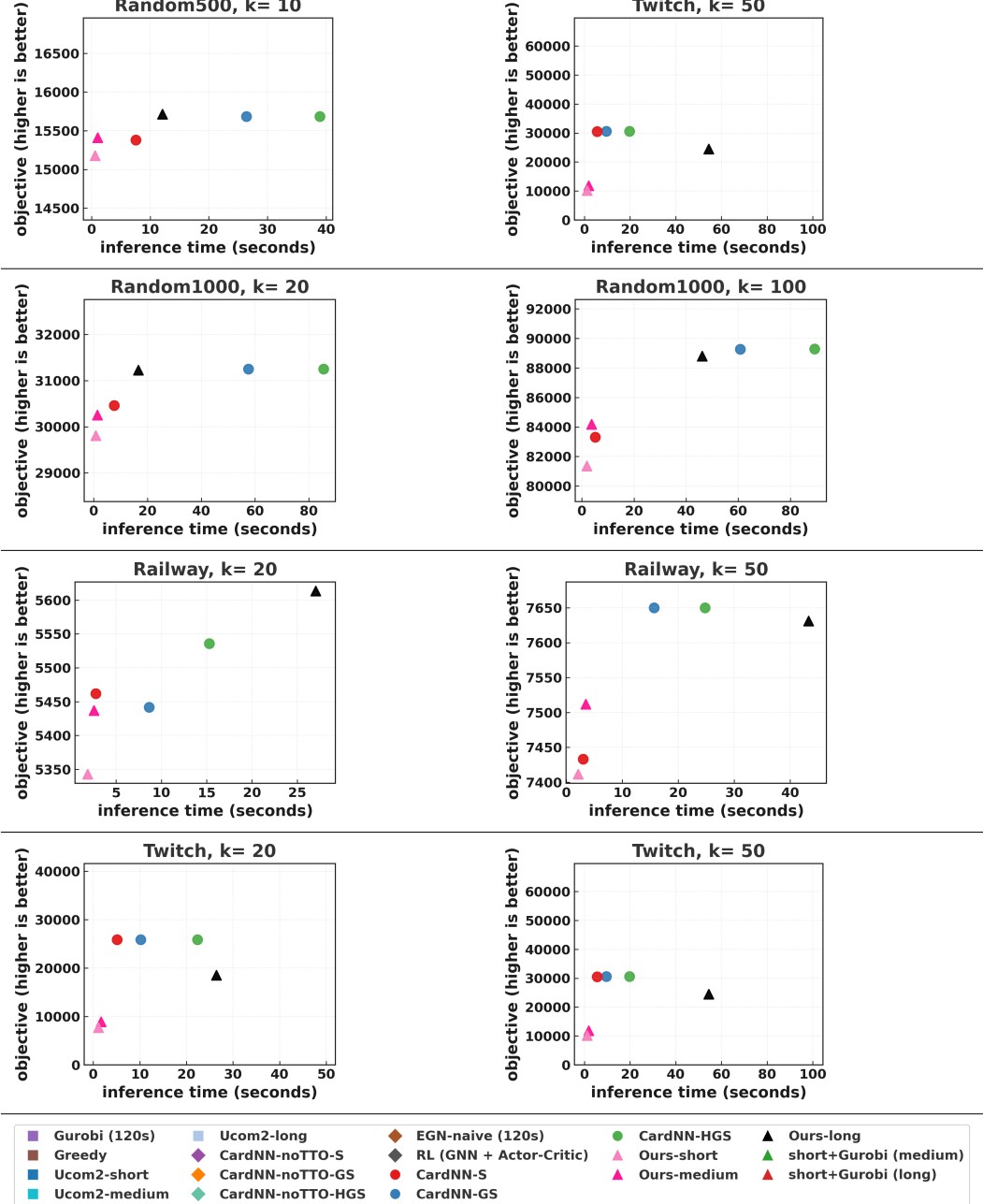

Figure 5: Performance comparison of our method against baseline approaches across non-Learning/traditional methods on multiple datasets. Metrics used are inference time (lower is better) and objective value (higher is better). While greedy is an efficient baseline with a strong approximation guarantee, our method performs competitively and is capable of outperforming it on datasets like Random500 for larger values of $k$. We are also able to outperform UCOM in several cases (e.g., Random1000 and Railway), though there are instances where greedy and/or UCOM perform the best, such as the Twitch dataset.

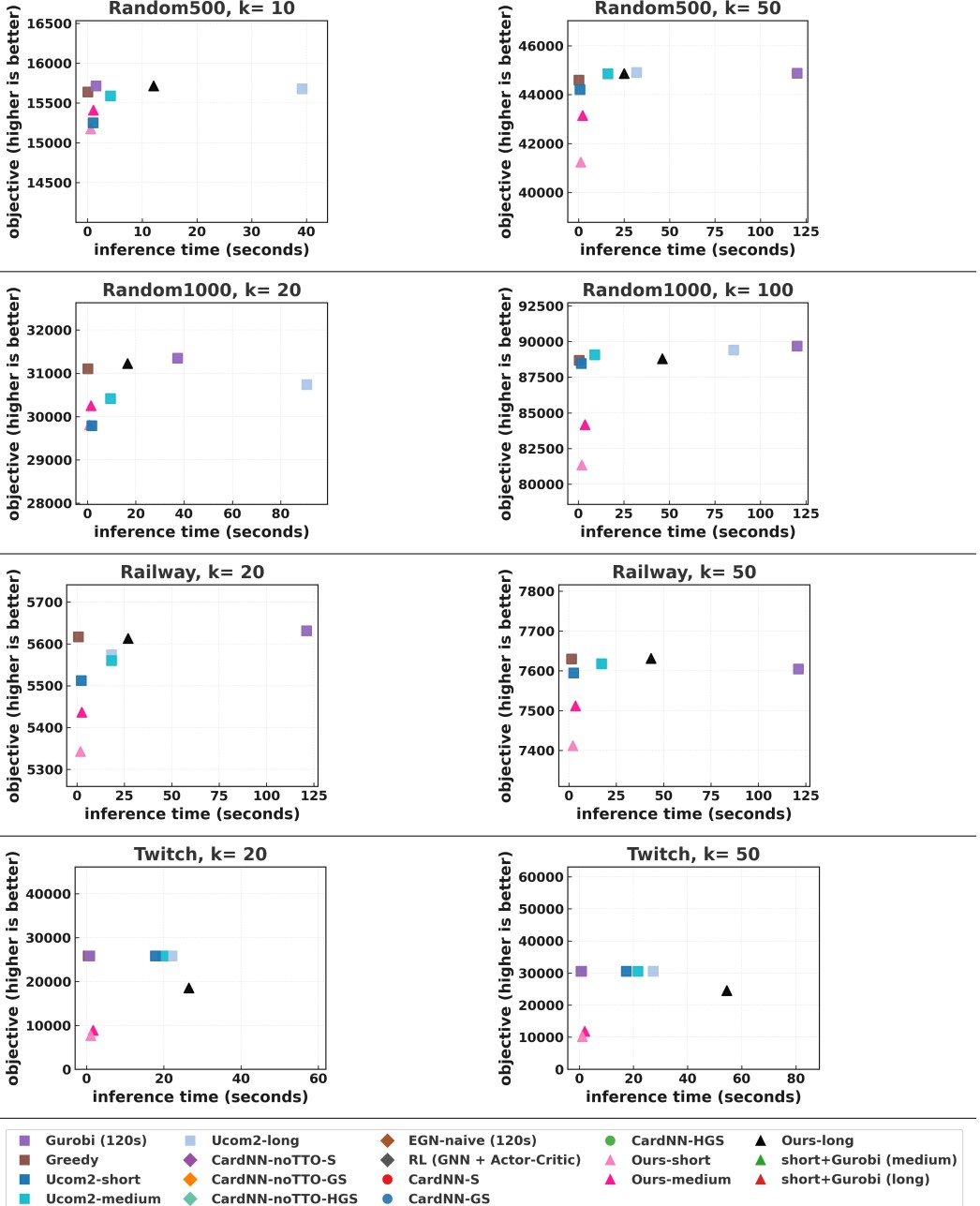

Figure 6: Performance comparison of our method against all baseline approaches. Metrics used are inference time (lower is better) and objective value (higher is better).

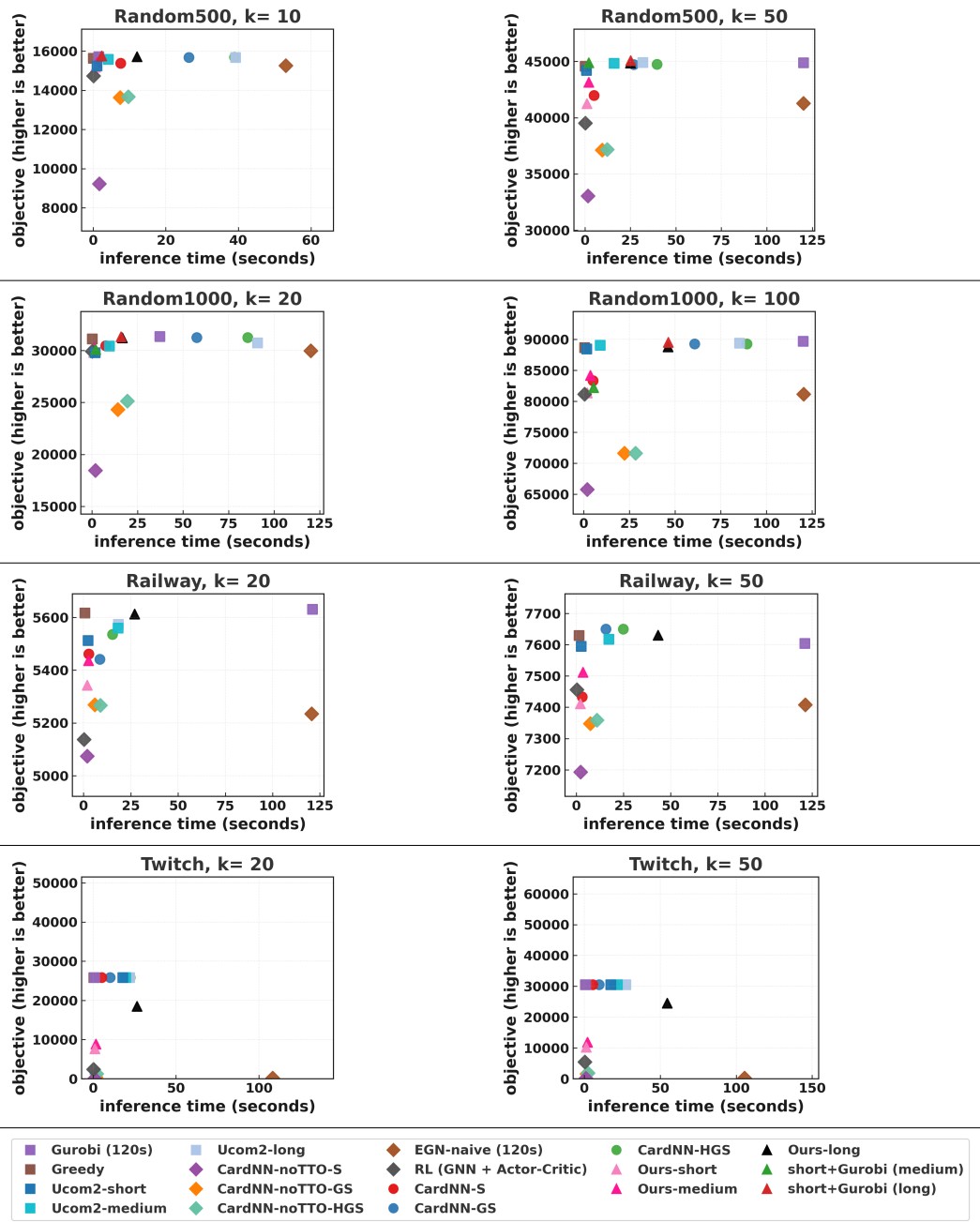

Table 5: Maximum Coverage on **rand500**. Two budgets: $k=10$ and $k=50$. Time: lower is better; Obj: higher is better.

| Setup / Method | $k = 10$ | | $k = 50$ | |
|---|---|---|---|---|
| | Time↓ | Obj↑ | Time↓ | Obj↑ |
| **Non-learning / Traditional** | | | | |
| Random (240s) | 240.0000s ± 0.0000s | 13372.76 ± 308.83 | 240.0000s ± 0.0000s | 36786.89 ± 655.45 |
| Gurobi (120s) | 1.5744s ± 0.5376s | 15714.90 ± 346.84 | 120.0651s ± 0.0268s | 44880.59 ± 7.13 |
| Greedy | 0.0572s ± 0.0502s | 15640.99 ± 360.60 | 0.1210s ± 0.0563s | 44597.56 ± 848.28 |
| Ucom2-short | 1.0411s ± 0.0827s | 15253.35 ± 370.00 | 0.7392s ± 0.0150s | 44208.95 ± 768.68 |
| Ucom2-medium | 4.1891s ± 0.1150s | 15589.25 ± 363.43 | 16.0005s ± 0.0408s | 44852.82 ± 765.11 |
| Ucom2-long | 39.2018s ± 0.9474s | 15679.45 ± 358.41 | 31.7587s ± 0.0811s | 44906.50 ± 761.75 |
| **Learning (no TTO)** | | | | |
| CardNN-noTTO-S | 1.6880s ± 0.0791s | 9231.54 ± 827.09 | 1.7211s ± 0.0661s | 33055.87 ± 1211.03 |
| CardNN-noTTO-GS | 7.4571s ± 0.0300s | 13635.05 ± 303.36 | 9.5240s ± 0.5770s | 37120.46 ± 610.25 |
| CardNN-noTTO-HGS | 9.6954s ± 0.0429s | 13671.99 ± 302.84 | 12.2502s ± 0.1068s | 37164.81 ± 678.19 |
| EGN-naive (120s) | 53.0409s ± 5.3879s | 15262.76 ± 401.10 | 120.1361s ± 0.0796s | 41272.68 ± 737.36 |
| RL (GNN+Actor-Critic) | 0.0696s ± 0.0028s | 14741.26 ± 322.51 | 0.2471s ± 0.0290s | 39510.96 ± 975.21 |
| **Learning (with TTO)** | | | | |
| CardNN-S | 7.5387s ± 0.6576s | 15381.60 ± 394.79 | 4.9843s ± 0.2941s | 41971.22 ± 814.87 |
| CardNN-GS | 26.3746s ± 0.0831s | 15683.40 ± 350.79 | 26.8944s ± 0.1131s | 44724.42 ± 773.70 |
| CardNN-HGS | 38.8875s ± 0.1065s | 15685.39 ± 349.83 | 39.6041s ± 0.1122s | 44745.20 ± 769.93 |
| **Ours** | | | | |
| Ours-short | 0.5786s ± 0.0094s | 15177.77 ± 344.68 | 1.1078s ± 0.0149s | 41247.80 ± 965.48 |
| Ours-medium | 1.0546s ± 0.0146s | 15410.23 ± 353.91 | 2.1212s ± 0.0096s | 43152.87 ± 1004.95 |
| Ours-long | 12.0878s ± 0.1365s | 15714.63 ± 354.22 | 24.9885s ± 0.3543s | 44866.84 ± 876.78 |
| short+Gurobi (medium) | N/A | N/A | 2.0622s ± 0.1122s | 44894.62 ± 951.89 |
| short+Gurobi (long) | 2.3185s ± 0.5136s | 15758.62 ± 343.40 | 25.0653s ± 0.0609s | 45090.41 ± 887.32 |

Table 6: Maximum Coverage on **rand1000**. Two budgets: $k=20$ and $k=100$. Time: lower is better; Obj: higher is better.

| Setup / Method | $k = 20$ | | $k = 100$ | |
|---|---|---|---|---|
| | Time↓ | Obj↑ | Time↓ | Obj↑ |
| **Non-learning / Traditional** | | | | |
| Random (240s) | 240.0000s ± 0.0000s | 24133.50 ± 390.27 | 240.0000s ± 0.0000s | 70527.31 ± 1051.87 |
| Gurobi (120s) | 37.2985s ± 30.0300s | 31347.62 ± 509.75 | 120.1395s ± 0.0526s | 89696.83 ± 1231.76 |
| Greedy | 0.1905s ± 0.0998s | 31105.89 ± 506.23 | 0.4609s ± 0.1074s | 88685.40 ± 1225.67 |
| Ucom2-short | 1.7342s ± 0.0677s | 29791.66 ± 678.02 | 1.6216s ± 0.0644s | 88472.61 ± 1261.36 |
| Ucom2-medium | 9.5523s ± 0.0620s | 30420.69 ± 552.66 | 8.9956s ± 0.0890s | 89072.92 ± 1248.58 |
| Ucom2-long | 90.7475s ± 0.3342s | 30744.61 ± 512.61 | 85.2663s ± 0.6164s | 89408.28 ± 1232.74 |
| **Learning (no TTO)** | | | | |
| CardNN-noTTO-S | 1.8693s ± 0.0018s | 18458.92 ± 1283.61 | 1.8810s ± 0.0453s | 65793.40 ± 1478.50 |
| CardNN-noTTO-GS | 14.1629s ± 0.0301s | 24309.37 ± 465.91 | 22.3352s ± 0.0942s | 71620.38 ± 1026.81 |
| CardNN-noTTO-HGS | 19.3888s ± 0.0532s | 25135.52 ± 436.21 | 28.3368s ± 0.1218s | 71604.61 ± 1010.17 |
| EGN-naive (120s) | 120.0105s ± 0.3980s | 29968.04 ± 563.02 | 120.3565s ± 0.1600s | 81166.12 ± 1391.02 |
| RL (GNN+Actor-Critic) | 0.1021s ± 0.0029s | 29912.14 ± 571.12 | 0.4652s ± 0.0029s | 81158.54 ± 1215.23 |
| **Learning (with TTO)** | | | | |
| CardNN-S | 7.6054s ± 0.6608s | 30461.73 ± 590.50 | 5.0909s ± 0.3016s | 83319.37 ± 1324.81 |
| CardNN-GS | 57.5445s ± 0.0670s | 31250.56 ± 528.72 | 60.7239s ± 0.0665s | 89269.41 ± 1253.68 |
| CardNN-HGS | 85.4505s ± 0.0952s | 31250.47 ± 526.03 | 89.3385s ± 0.1235s | 89280.17 ± 1261.34 |
| **Ours** | | | | |
| Ours-short | 0.8248s ± 0.0312s | 29810.12 ± 586.05 | 1.9076s ± 0.0284s | 81357.29 ± 1277.84 |
| Ours-medium | 1.4175s ± 0.0633s | 30257.05 ± 606.07 | 3.6669s ± 0.0582s | 84186.79 ± 1336.48 |
| Ours-long | 16.6007s ± 0.4632s | 31230.26 ± 476.25 | 46.1200s ± 0.6998s | 88796.03 ± 1327.83 |
| short+Gurobi (medium) | 2.0868s ± 0.1001s | 30098.64 ± 767.38 | 5.2979s ± 0.1202s | 82200.65 ± 2609.87 |
| short+Gurobi (long) | 15.8728s ± 2.8134s | 31332.18 ± 471.59 | 46.2701s ± 0.3331s | 89509.90 ± 1316.43 |

Table 7: Maximum Coverage on **railway**. Two budgets: $k=20$ and $k=50$. Time: lower is better; Obj: higher is better.

| Setup
Method | $k = 20$ | | $k = 50$ | |
|---|---|---|---|---|
| | Time↓ | Obj↑ | Time↓ | Obj↑ |
| **Non-learning / Traditional** | | | | |
| Random (240s) | 240.0000s ± 0.0000s | 5291.67 ± 73.92 | 240.0000s ± 0.0000s | 7367.00 ± 76.97 |
| Gurobi (120s) | 121.0590s ± 0.0318s | 5631.67 ± 28.77 | 121.0329s ± 0.0613s | 7604.67 ± 62.38 |
| Greedy | 0.7271s ± 0.0251s | 5617.00 ± 49.00 | 1.3196s ± 0.5774s | 7630.00 ± 72.19 |
| Ucom2-short | 2.2988s ± 0.2026s | 5512.67 ± 56.89 | 2.4537s ± 0.3920s | 7594.67 ± 68.72 |
| Ucom2-medium | 18.2307s ± 2.4105s | 5560.33 ± 44.56 | 17.1635s ± 2.2504s | 7618.00 ± 79.76 |
| Ucom2-long | 18.3035s ± 2.4544s | 5574.33 ± 58.79 | 17.1988s ± 2.2001s | 7617.00 ± 81.19 |
| **Learning (no TTO)** | | | | |
| CardNN-noTTO-S | 1.9287s ± 0.2161s | 5074.33 ± 59.41 | 2.1892s ± 0.2505s | 7193.00 ± 80.72 |
| CardNN-noTTO-GS | 5.9863s ± 0.5143s | 5269.00 ± 68.83 | 7.3275s ± 0.3261s | 7348.33 ± 80.31 |
| CardNN-noTTO-HGS | 8.8791s ± 0.7784s | 5266.67 ± 76.01 | 10.8211s ± 0.6323s | 7359.00 ± 70.89 |
| EGN-naive (120s) | 120.6763s ± 0.2931s | 5234.67 ± 52.47 | 121.3351s ± 0.9133s | 7408.33 ± 72.82 |
| RL (GNN+Actor-Critic) | 0.2191s ± 0.0030s | 5137.00 ± 52.01 | 0.2191s ± 0.0031s | 7456.67 ± 57.14 |
| **Learning (with TTO)** | | | | |
| CardNN-S | 2.7580s ± 0.2705s | 5462.00 ± 56.00 | 3.0035s ± 0.2782s | 7433.33 ± 63.34 |
| CardNN-GS | 8.6626s ± 0.5672s | 5441.67 ± 58.77 | 15.6866s ± 1.0987s | 7650.00 ± 73.33 |
| CardNN-HGS | 15.2983s ± 0.5441s | 5535.67 ± 64.52 | 24.7941s ± 0.9907s | 7650.00 ± 88.71 |
| **Ours** | | | | |
| Ours-short | 1.8559s ± 0.0458s | 5343.00 ± 89.17 | 2.0838s ± 0.0142s | 7411.67 ± 55.08 |
| Ours-medium | 2.5628s ± 0.0418s | 5437.00 ± 66.84 | 3.4385s ± 0.0532s | 7512.00 ± 63.85 |
| Ours-long | 27.0344s ± 0.9617s | 5613.33 ± 55.58 | 43.2777s ± 1.3728s | 7631.00 ± 75.19 |

Table 8: Maximum Coverage on **twitch**. Two budgets: $k=20$ and $k=50$. Time: lower is better; Obj: higher is better.

| Setup
Method | $k = 20$ | | $k = 50$ | |
|---|---|---|---|---|
| | Time↓ | Obj↑ | Time↓ | Obj↑ |
| **Non-learning / Traditional** | | | | |
| Random (240s) | 240.0000s ± 0.0000s | 12889.17 ± 5636.10 | 240.0000s ± 0.0000s | 16050.50 ± 6715.32 |
| Gurobi (120s) | 0.8221s ± 0.5700s | 25864.00 ± 10223.00 | 0.7961s ± 0.6577s | 30560.33 ± 11922.45 |
| Greedy | 0.3970s ± 0.2536s | 25855.50 ± 11207.33 | 0.6553s ± 0.3883s | 30542.33 ± 13055.67 |
| Ucom2-short | 17.8031s ± 18.4783s | 25833.00 ± 11212.59 | 17.3739s ± 18.1940s | 30526.00 ± 13043.86 |
| Ucom2-medium | 19.6806s ± 19.8771s | 25858.17 ± 11206.89 | 21.6114s ± 21.3478s | 30544.17 ± 13052.91 |
| Ucom2-long | 22.0589s ± 21.6886s | 25858.17 ± 11206.89 | 27.2170s ± 25.6604s | 30544.17 ± 13052.91 |
| **Learning (no TTO)** | | | | |
| CardNN-noTTO-S | 0.8636s ± 0.2075s | 77.00 ± 51.47 | 0.9635s ± 0.4585s | 170.50 ± 158.15 |
| CardNN-noTTO-GS | 1.5749s ± 0.3842s | 376.00 ± 194.67 | 1.6334s ± 0.6782s | 1574.33 ± 631.04 |
| CardNN-noTTO-HGS | 2.1831s ± 0.7328s | 1242.33 ± 499.23 | 2.3674s ± 0.7858s | 1898.00 ± 506.67 |
| EGN-naive (120s) | 108.3030s ± 24.3569s | 152.67 ± 12.22 | 105.4605s ± 20.7886s | 247.17 ± 31.88 |
| RL (GNN+Actor-Critic) | 0.1694s ± 0.0028s | 2379.50 ± 1471.50 | 0.2905s ± 0.0030s | 5435.50 ± 3318.67 |
| **Learning (with TTO)** | | | | |
| CardNN-S | 5.1258s ± 0.6503s | 25863.83 ± 11198.96 | 5.5767s ± 0.5135s | 30556.00 ± 13063.12 |
| CardNN-GS | 10.1648s ± 1.1928s | 25864.00 ± 11198.73 | 9.5794s ± 1.5230s | 30560.67 ± 13061.60 |
| CardNN-HGS | 22.3602s ± 1.2962s | 25864.00 ± 11198.73 | 19.7813s ± 3.4102s | 30560.83 ± 13061.35 |
| **Ours** | | | | |
| Ours-short | 1.0975s ± 0.7446s | 7689.00 ± 5712.56 | 1.1485s ± 0.7596s | 10227.00 ± 5537.29 |
| Ours-medium | 1.6693s ± 0.9268s | 8887.83 ± 5087.20 | 1.9401s ± 1.0147s | 11858.50 ± 4793.61 |
| Ours-long | 26.4401s ± 15.9632s | 18526.83 ± 8671.00 | 54.4778s ± 32.5091s | 24509.83 ± 11171.33 |

# H   Max-k-Cut Ablation

**Datasets.**   We conduct experiments on two datasets. The IMDB-BINARY dataset [92] consists of 1000 graphs. The Erdős–Rényi dataset consists of 1000 synthetic graphs generated under the $G(n, p)$ model. For each graph, the number of nodes $n$ is drawn uniformly from $\{50, \ldots, 100\}$ and each candidate edge is included independently with probability $p = 0.15$. Our experiments use the following split for both datasets. 60% of the graphs are allocated for training, 20% for validation, and the final 20% for testing. Within each split, any graphs with fewer than $k$ nodes are excluded. We compare in two settings, one of small and one of large $k$, where $k$ is selected as a fraction of the average number of nodes in the dataset. For small $k$ we pick the fraction to be 0.25 of the node average and for the large $k$ we pick 0.75.

**Random sampling + decomp.**   The purpose of this baseline is to highlight the contribution of optimization and the neural network in the performance. In this baseline, we randomly sample a single point in $\Delta_{n,k}$, decompose it, and then report the value of the best performing set in the decomposition.

**Direct optimization + decomp.**   In the direct optimization approach, we optimize a vector $\mathbf{x} \in \mathbb{R}^n$ using the perturbation method described in Section 4.1, i.e., we treat it as an additive perturbation on the vector $(k/n, k/n, \ldots, k/n)$ in $\Delta_{n,k}$. We optimize the perturbation $\mathbf{x}$ by decomposing the perturbed point and optimizing the value of the extension with gradient descent (Adam [40]). For IMDB-BINARY, we set the learning rate to 0.015, and for Erdős–Rényi, the learning rate was set to 0.012. Each optimization run consists of 150 update steps.

**NN + Optimization + Decomp.**   The neural network approach proceeds in a similar fashion as the direct optimization one. Here, we use a neural network to generate the perturbation vector $\mathbf{x} \in \mathbb{R}^n$ for the perturbation approach described in Section 4.1. The perturbed interior point is decomposed and we updated the parameters of the neural network by gradient descent on the extension. Here, the neural network is an eight-layer GatedGraphConv network [49] followed by two linear layers. Node features include random-walk positional encodings generated with AddRandomWalkPE [19] with walk length 10. The neural network in this case is optimized directly on the test data. The results are presented in the table below.

**Standard SSL.**   In the standard SSL baseline, we train the same architecture on separate training data and do model selection using a validation set. Then we report the performance of the selected model (without any finetuning) on the test set. We observe that while the trained SSL baseline is not able to match the performance of a neural net directly optimized on the test set, it consistently produces performance that is competitive with direct optimization and greedy for large $k$.

Table 9: Test-set performance (mean $\pm$ std) on IMDB-BINARY graphs (avg. 20 nodes) and Erdős–Rényi graphs (avg. 75 nodes, edge density 0.15).

| Method | IMDB-BINARY | | Erdős–Rényi | |
| --- | --- | --- | --- | --- |
| | $k = 5$ | $k = 15$ | $k = 15$ | $k = 60$ |
| Greedy Algorithm | $1.000 \pm 0.002$ | $0.801 \pm 0.195$ | $0.985 \pm 0.015$ | $0.900 \pm 0.047$ |
| Random Sampling + Decomp | $0.881 \pm 0.140$ | $0.850 \pm 0.179$ | $0.753 \pm 0.038$ | $0.761 \pm 0.041$ |
| Direct optimization + Decomp | $0.960 \pm 0.064$ | $0.922 \pm 0.158$ | $0.833 \pm 0.025$ | $0.844 \pm 0.043$ |
| NN + Optimization + Decomp | $0.971 \pm 0.041$ | $0.932 \pm 0.089$ | $0.910 \pm 0.045$ | $0.902 \pm 0.041$ |
| Standard SSL + Decomp | $0.956 \pm 0.049$ | $0.905 \pm 0.117$ | $0.899 \pm 0.035$ | $0.889 \pm 0.044$ |

