# OpenReview forum: "Geometric Algorithms for Neural Combinatorial Optimization with Constraints"
_NeurIPS.cc/2025/Conference — NeurIPS 2025 poster_

### Official Review · Reviewer_gbDc · 2025-06-20

**Clarity:** 3
**Significance:** 2
**Originality:** 3
**Rating:** 2
**Confidence:** 3

**Summary:**

This paper proposes a new pipeline to tackle combinatorial optimization problems with neural networks. The key novelty is to use a Caratheodory-type geometric algorithm in order to convert continuous solutions efficiently into sparse convex combinations of discrete solutions. This can be used both during training (in order to define a meaningful objective) and inference (in order to define a meaningful rounding procedure).

**Questions:**

no further questions

**Ethical Concerns:**

["NO or VERY MINOR ethics concerns only"]

**Final Justification:**

See my response to the rebuttal

**Limitations:**

no concerns

**Paper Formatting Concerns:**

no concerns

**Quality:**

2

**Strengths And Weaknesses:**

The general idea of the paper makes a lot of sense and seems to be promising. Blindly applying neural nets to CombOpt problems certainly doesn't work well - and incorporating geometric insights is the right way to make things work.

I am not entirely familiar with the practical side of neural combinatorial optimization. The results look decent, even if not totally mind-blowing. I cannot judge that much, how novel the work is compared to previous work on neural combinatorial optimization. What I can however judge well is the mathematical foundations of this work. On this side, while I think the general approach makes a lot of sense, I have do have doubts with respect to the broad applicability. I also think that the presentation is insufficient. My concrete concerns (with respect to that and more generally) are as follows:

- While the authors make bold claims about "incorporating geometric algorithms in deep learning pipelines", all they do in the end is a standard Caratheodory decomposition. I recognize that it is a contribution to incorporate them into deep learning pipelines, but I would like to emphasize that this is not groundbreaking on a mathematical level and has been used in CombOpt algorithms for a long time.
- Algorithm 1 is insufficiently described. Many points remain unclear: what exactly is the support of x_t? How to choose S_t? What is g? Shouldn't, by Caratheodory's theorem, the algorithm terminate after n+1 steps, with epsilon = 0, so why do you need this epsilon-test? Why should it output a *unique* Caratheodory decomposition (which is necessary to interpret the coefficients as probabilities)? How does this unique decomposition differ from all other possible decompositions?
- Theorem 4.1 contains many undefined terms and is not helpful in this formulation. I understand that it is an informal version, but it does not help to build any intuition if formulated like this. Also it remains unclear why suddenly n terms are sufficient, while Caratheodory requires n+1.
- It is questionable whether the proposed pipeline is broadly applicable to difficult problems. The example problems in the paper (optimizing over a uniform matroid, partition matroid, spanning trees) are all easy problems from a complexity perspective. Independent set is a hard problem, but here the authors actually discuss a relaxation instead of the actual problem and it remains unclear how to transform a vertex of the fractional stable set polytope into an actual independent set. In conclusion, I am not convinced that this pipeline generalizes to CombOpt problems with more complicated underlying polytopes.
- line 107: I find it questionable to defer related work discussions to the appendix. If the work is relevant, put it in the main text. If not, leave it out.
- As far as I understand, the authors have not submitted code for their experiments.

Minor (no need to address in rebuttal):
- line 20/21: "These problems are often non-convex" --> this statement is misleading. Are you talking about the domain? Or the objective function? The former is discrete, which induces challenges, but talking about convexity doesn't make much sense. The latter is often comparably simple, e.g., linear (not necessarily, but claiming it would be "often non-convex" is too general).
- line 22-24: this sentence needs citations
- line 24: eschew --> maybe use a more standard word?
- line 27: "discrete objective" --> what exactly do you mean with that? The objective itself is usually not discrete, just the set of feasible solutions is. (same in line 49)
- Figure 1: the graphic has low quality. Use vector graphic instead.
- beginning of Sec. 3: I don't think it makes much sense to introduce the problem with general D if you then only focus on D={0,1} in the paper. A simpler problem setup would have been sufficient.
- line 127: what are positional encodings? I don't think this term has been introduced before.
- line 137: if you follow this strategy, do you need training data at all? Why?
- line 139: add comma before "which"
- around line 159: in general, there are many ways to write down a Caratheodory decomposition of an interior point in a polytope. So, in order to view the coefficients as probabilities, you need to define a *unique* way of finding a Caratheodory decomposition! I suppose you do that, but make it more explicit here.
- line 183: if this method comes from GLS, why do you cite two papers by other authors? Is there no original reference corresponding to these three names?
- line 195: and --> an (??)
- In Alg. 1 and the text in section 4: please define what "the support of x_t" is supposed to be.
- line 240/241: which problems are you referring to here? Please name them and add citations.
- I would have liked more details in the main submission on the spanning tree case and fewer on the cardinality case. Reason: the cardinality case is a quite simple one, while spanning trees already possess interesting and non-trivial combinatorial structure.
- line 285: "There is no compact description of the polytope of independent sets of a graph" --> this requires a citation. I suspect that this statement refers to an extension complexity lower bound (which would be the right thing to cite here).
- line 312: one space too much before semicolon
- lines 326-328: I don't understand what issue you are referring to here. Please explain better.
- Table 1 should be referenced somewhere in the main text.
- Why is the time limit for different approaches different? (I.e. Gurobi 240s, Random 120s)

---

> ### Author Rebuttal · Authors · 2025-07-29
>
> We would like to thank the reviewer for their detailed response.
> > While the authors make bold claims about "incorporating geometric algorithms in deep learning pipelines", all they do in the end is a standard Caratheodory decomposition.
>
> The reviewer adds
> >but I would like to emphasize that this is not groundbreaking on a mathematical level and has been used in CombOpt algorithms for a long time.
>
> - We believe that the statement of 'incorporating geometric algorithms in deep learning pipelines' is appropriate since the proposed method leverages constructive results from the literature on geometric algorithms to implement a differentiable pipeline for combinatorial optimization with neural nets.  The theorem that forms the basis of our approach is in the GLS book 'Geometric algorithms and combinatorial optimization'. Our wording choice in the line quoted by the reviewer is a direct reference to the title of the book. We cite the book and mention the theorem explicitly in the paper (see line 192). We do not attempt to present this as some kind of outstanding mathematical achievement. The contribution lies in leveraging such algorithms for the purpose of neural comb opt. In that context, the constructive Caratheodory decomposition is able to effectively deal with both loss function design and rounding. We also provide extensive benchmarking to show the empirical benefits of the method (see also response to reviewer 7pF1). Our ablation also shows how this can even work using direct optimization with Adam.
>
> - The reviewer acknowledges that such decompositions have been used in combinatorial algorithms for a long time. That is one of the strengths of this approach, since one can leverage results from the literature to adapt our method to other problems (e.g., scheduling, networking, etc.). To the best of our knowledge, the specific decompositions we describe have not been published but by no means constitute mathematical breakthroughs. That's because their existence essentially follows from the GLS result (theorem 4.1). They are valuable elements of our contribution, but they are not the central contribution of the paper. It is worth noting that the design of such efficient decompositions has been acknowledged as an interesting question in the literature (see for example section 4.3 in [1] or the introductory remarks in [2]).   Introducing this problem as an element of machine learning algorithm design and making progress on it could be beneficial for both machine learning communities and communities working on algorithms. We certainly do not intend to mislead with our claims so we welcome any corrections of specific sentences that the reviewer believes would alleviate the issue.
>
> >Algorithm 1 is insufficiently described...
>
> also
>
> >Theorem 4.1 contains many undefined terms and is not helpful in this formulation. I understand that it is an informal version...
>
> - Here are are some clarifying comments that we will make sure to include in the final version of the paper:
>
>   - By support of $\mathbf{x}$, we mean the nonzero coordinates of the vector.
>
>   - The function $g$ is the (a.e.) differentiable function that computes the coefficient of the current iterate given the current iterate and a vertex of the polytope. We will make sure to introduce it earlier in the paper since this was also confusing to another reviewer. For example, the cardinality constraint section of the paper provides a concrete example for $g$ (see Equation 6).
>
>   - Finding a vertex for the current iterate is problem dependent. In the most general case, according to the GLS theorem, we need to find a vertex of the smallest face of the polytope that contains the current iterate. This is further elaborated on in appendix B. For the problems we considered, a greedy algorithm can be sufficient. For instance, in the cardinality constraint we fetch the vertex corresponding to the the top-k coordinates. In some cases we may have to solve a linear program.
>
> > Why should it output a unique Caratheodory decomposition...
>
> - The reviewer raises a valid point about the uniqueness of the decomposition. The uniqueness is required in order to ensure a.e. differentiability of the extension and we achieve uniqueness by designing a deterministic decomposition algorithm that maps a point from the polytope to a distribution. The details of such deterministic algorithm are problem specific and we provide several examples such as cardinality constraints and other matroids. For example, if we have a problem defined on graphs, we need to fix an ordering of the vertices of the input graph so as to consistently obtain a vertex and a coefficent corresponding to the current iterate. The decomposition then yields a deterministic mapping from the point in the polytope to the coefficients. In the case of the cardinality constraint, the set corresponding to the top-k coordinates of each iterate is chosen. Each iterate is a deterministic function of the previous iterate. Hence, the collection of sets (corresponding to top-k coordinates) and their coefficients are consistently obtained from the inputs. The ordering may only become relevant in case of tie-breaks (a rare occurrence in practice).
>
>
> > Shouldn't, by Caratheodory's theorem, the algorithm terminate after n+1 steps, with epsilon = 0, so why do you need this epsilon-test?
>
> - The number of steps is always at most $n+1$ in theorem 4.1, which indeed has epsilon=0. Iterations should start from 1 and end at $n+1$ in the GLS theorem (Theorem 4.1). We apologize for the typo. Terms like strong optimization oracle are explained in the Appendix B but we will make sure to clear this up in the updated version.
>
> - In appendix $C$ the role of the $ \epsilon $ test is discussed in more detail. The main idea is to obtain a decomposition that contains potentially more than $n+1$ sets by rescaling the coefficient choice at every iteration. This allowed for better exploration (and convergence) during training and led to improved results. Furthermore,  the $\epsilon$ test is motivated by the existence of approximate versions of the Caratheodory decomposition that do not depend on the ambient dimension of the space. We mention those in the last paragraph of the related work section of the main paper (references [3,12,47]). We consider this as a promising future direction for this line of work.
>
> > It is questionable whether the proposed pipeline is broadly applicable to difficult problems. The example problems in the paper (optimizing over a uniform matroid, partition matroid, spanning trees) are all easy problems from a complexity perspective.
> -  First, we want to acknowledge that depending on the problem, it may indeed be challenging to implement our approach. We mention this in the conclusion and in the extended discussion in Appendix B. We want to emphasize that cardinality constrained problems like maximum coverage or max-k-cut are known to be NP-Hard. However, the computational complexity of the problem is not sufficient to decide whether our method is applicable. As another example, there exist efficient decompositions for permutation polytopes and permutation polytopes can be used to solve hard (in terms of complexity) combinatorial problems like travelling salesperson. More generally, the viability of our approach primarily depends on whether one can find vertices and coefficients for each iteration of the decomposition efficiently. Hence, implementing our approach will be challenging for cases where linear optimization over the polytope is computationally hard.
> This is what is troublesome for independent set.  We included the independent set example to highlight that even in cases where it is not obvious how to construct an efficient decomposition due to the potential intractability of the polytope, there are compromises that can be made to yield a viable method (i.e., a relaxation like FSTAB).
>   FSTAB's corners are either integral vertices corresponding to independent sets or half-integral vertices. The decomposition will terminate but it may include some infeasible points (half-integral vertices). We may simply set the value of the objective to 0 for all non-integral points so that for an FSTAB extension the optimization will prioritize integral corners. Admittedly, this is not as mathematically clean as in other constraints but we wanted to bring up how one could deal with those harder cases.
>
> > I find it questionable to defer related work discussions to the appendix. If the work is relevant, put it in the main text. If not, leave it out.
>
> - Given the space limitations, it is not viable to include almost 2 pages of related work in the main paper. We chose to prioritize the papers from the literature that we find are the most pertinent in the context of our contribution. The approach we take to those problems is not widely known, so our view was that the additional discussion around related work that we provide in the appendix could help interested readers contextualize our contribution and improve the overall accessibility of the paper.
>
> >As far as I understand, the authors have not submitted code for their experiments.
>
>
> - We have submitted the code and the link can be found on page 17 of our corresponding NeurIPS checklist answer to this question. We will make sure to mention this in the updated main body of the paper too.
>
>
>  References:
>  1. Cunningham, William H. "On submodular function minimization." Combinatorica 5.3 (1985): 185-192.
>
>  2. Hoeksma, Ruben, Bodo Manthey, and Marc Uetz. "Efficient implementation of Carathéodory’s theorem for the single machine scheduling polytope." Discrete applied mathematics 215 (2016): 136-145.

---

> > ### Comment · Reviewer_gbDc · 2025-08-01
> >
> > I thank the authors for their extensive and insightful rebuttal.
> > - I remain sceptical about the title: you are essentially borrowing the name from the title of a book because one algorithm of that book is used.
> > - I remain sceptical about the broader applicability of the methods. I understand that the applicability cannot directly be related to computational complexity, but I still think that the positive examples in the paper are hand-picked problems for which the authors where lucky that it worked. The authors admit in their rebuttal that it becomes difficult as soon as the linear program over the polytope is difficult - which is the case even for some problems in P (compare [Thomas Rothvoss. 2017. The Matching Polytope has Exponential Extension Complexity. J. ACM 64, 6, Article 41]).
> > - I remain sceptical about the writing, compare my original comments. I appreciate that the authors clarified many points in the rebuttal, but NeurIPS does not have a proper revision cycle and I think there are too many points where the paper is not precise enough for being accepted without a new version being peer-reviewed.
> > - I do not want to further judge about having large parts of related work in the appendix. I don't like the practice, but this is not a major point for me.
> > - Sorry for missing the link to the code in the appendix. Yes, I think it is good if you have the reference in the main text.
> >
> > Overall, for the mentioned reasons, I continue to vote against acceptance of the paper.

---

> > > ### Author Response · Authors · 2025-08-01
> > >
> > > >You are essentially borrowing the name from the title of a book because one algorithm of that book is used.
> > >
> > > We use a theorem that serves as the foundation for a class of algorithms and we have cited those as they appear in the literature while also providing new examples for specific cases. Our title is also suitably modified compared to the book to reflect the use of those algorithms in the specific use case of neural combinatorial optimization. We explained our reasoning for the choice but we also requested specific suggestions from the reviewer that would alleviate the issue. The reviewer has not provided any. Regardless of our disagreement, it is not difficult to fix a title so we would appreciate if you could follow up on this.
> > >
> > > > I do not want to further judge about having large parts of related work in the appendix. I don't like the practice, but this is not a major point for me.
> > >
> > > We believe this should not have any impact on the assessment of our submission. If there are specific elements of related work that the reviewer believes should be in the main submission we are more than willing to accommodate their request and make the appropriate modifications. If there is an issue here, it is also quite easy to fix.
> > >
> > > > but I still think that the positive examples in the paper are hand-picked problems for which the authors where lucky that it worked.
> > >
> > > As researchers and practitioners who work on the topic can appreciate, it is quite difficult to get new methods to work on NP-Hard combinatorial problems. Progress in the topic by the community has been steady but slow over several years. Having the ability to beat classic baselines and general purpose solvers like gurobi is primarily a matter of extensive effort in designing a viable approach and doing careful engineering to get it to work. We provided thorough documentation of that through sections in the appendix and the code that we provide. We find the reviewer's assessment here completely unfair.
> > >
> > > >  The authors admit in their rebuttal that it becomes difficult as soon as the linear program over the polytope is difficult - which is the case even for some problems in P (compare [Thomas Rothvoss. 2017. The Matching Polytope has Exponential Extension Complexity. J. ACM 64, 6, Article 41]).
> > >
> > > It is important to be precise with the wording when discussing this issue. Our specific claim in the rebuttal is:
> > >
> > > > 'Hence, implementing our approach will be challenging for cases where linear optimization over the polytope is computationally hard.'
> > >
> > > The reviewer  provides a reference about the *extension complexity* (i.e., the number of inequalities needed to describe the linear program) in their argument.  The extension complexity does *not* determine how applicable our approach is because it does not necessarily determine whether *linear optimization* over the polytope is computationally hard. For the matching polytope that they cite, there are already efficient Caratheodory decompositions that have been applied to practical problems (see algorithm 3 in [1]). This is because one can have efficient optimization/separation oracles despite exponential extension complexity (see also the GLS book). On the other hand, there are polytopes like the TSP polytope with exponential extension complexity that do not admit polytime optimization oracles (unless P=NP). However, one could still use our approach for a problem like TSP by switching to a permutation polytope that admits a fast oracle (e.g., view TSP as a quadratic assignment problem). Therefore, even there, the complexity of linear optimization is *not* sufficient to rule out our method.
> > >
> > >   Generally, a difficult polytope implies that a different formulation is required where hardness has been offloaded to the objective (e.g., the polytope admits a P oracle but the objective is a nonconvex quadratic; another example is the Motzkin formulation for maxclique).  In that sense, yes, our approach becomes more challenging because it requires some additional design decisions from the practitioner.
> > >
> > > However, treating this as grounds for rejection given the full scope of the contribution of the paper is, in our view, overly harsh but ultimately up to the judgement of the reviewer.
> > >
> > > >  but NeurIPS does not have a proper revision cycle and I think there are too many points where the paper is not precise enough for being accepted without a new version being peer-reviewed.
> > >
> > > We have provided clarifications. These amount to a few sentences that can be added to the main body of the paper in the final version. We are in the process of peer review; If we provided an incorrect explanation/clarification somewhere, we welcome corrections by the reviewer.
> > >
> > >
> > > 1. Y. H. Ezzeldin, M. Cardone, C. Fragouli and G. Caire, "Polynomial-time Capacity Calculation and Scheduling for Half-Duplex 1-2-1 Networks," 2019 IEEE International Symposium on Information Theory (ISIT), Paris, France, 2019, pp. 460-464, doi: 10.1109/ISIT.2019.8849671.

---

> > > > ### Comment · Reviewer_gbDc · 2025-08-03
> > > >
> > > > I thank the authors for their latest response. We have different opinions on the title and and how to treat related work, but I think we agree that these are not the big points that should influence the decision too much.
> > > >
> > > > Thanks also for the clarifications with the polytopes -- I would be very curious to learn more about the polyhedral structure necessary to make this approach applicable. If it is applicable to the matching problem, why haven't you mentioned this in the paper? Could you really solve, e.g., min-weight perfect matching in a general graph efficiently with your approach? Similarly for the approach you outline with respect to the TSP polytope: if this is a viable approach to improve existing TSP algorithms, why isn't it included in the paper?
> > > >
> > > > Besides my doubts on the general applicability, the one thing that I am firm about and which prevents me from recommending acceptance is the insufficient presentation. I'd like to see clean presentation of Algorithm 1 and Thm. 4.1, and other places I commented on in my original review. Since NeurIPS doesn't have a proper revision cycle, I think this paper should go through another round of peer-review before being accepted, hopefully after my comments on mathematical precision (including my minor comments) have been addressed.

---

> ### Author Response · Authors · 2025-08-04
>
> Thank you for engaging with our rebuttal and responses!
>
> >the one thing that I am firm about and which prevents me from recommending acceptance is the insufficient presentation. I'd like to see clean presentation of Algorithm 1 and Thm. 4.1, and other places I commented on in my original review.
>
> Could you please specify what is the exact issue with 4.1 or algorithm 1? Is there some specific problem with the answers we provided? To be clear, thm 4.1 in the paper is just a slightly abstracted version of the GLS theorem.  This was done purely to save space. The full theorem from the book can be found in our appendix, together with the preliminaries required, but we reproduce the essential components here to facilitate discussion:
>
>
> Let $\mathcal{P}\subseteq\mathbb{R}^n$ be a polyhedron and let $\varphi$ and $\nu$ be positive integers.
>
> We say that $\mathcal{P}$ has facet‐complexity at most $\varphi$ if there exists a system of inequalities with rational coefficients that has solution set $\mathcal{P}$ and such that the encoding length of each inequality of the system is at most $\varphi$.
>
> A well‐described polyhedron is a triple $(\mathcal{P}; n, \varphi)$ where $\mathcal{P}\subseteq\mathbb{R}^n$ is a polyhedron with facet‐complexity at most  $\varphi$.
>
> The strong optimization problem for a given rational vector $\mathbf{x}$ in $n$ dimensions and any well described  $(\mathcal{P}; n, \varphi)$ is given by  $ \max{\mathbf{c}^\top \mathbf{x}}, \quad \mathbf{c} \in \mathcal{P}$.
>
> A strong optimization oracle in our case is just an oracle for the above problem.
> The theorem is:
>
>   There exists a polynomial-time algorithm that for any well-described polyhedron $(\mathcal{P}; n, \varphi)$ given by a strong optimization oracle, for any rational vector $ \mathbf{x} $, finds affinely independent vertices $ \mathbf{1}_{S_1},  \mathbf{1} _{S_2} , \dots, \mathbf{1} _{S_k} $ and positive rational numbers  $\lambda _1, \lambda _2, \dots, \lambda _k$  such that   $\mathbf{x}=  \sum  _{t=1} ^k   \lambda _t \mathbf{1} _ {S_t} $.
> Here $k\leq n+1$.
>
> If the paper is accepted, we are given an additional page of content so including a few additional lines with definitions is easy.  We want to emphasize that all of the above are contained in the standard textbook by GLS. It is difficult to provide a complete primer into polyhedral geometry within the page limit that covers all of the required background, since one could further inquire about the details of the encoding length, the precise notion of an oracle, etc.
>
> Similarly for algorithm 1, could you please explain what the exact issue is?
>
> > If it is applicable to the matching problem, why haven't you mentioned this in the paper? Could you really solve, e.g., min-weight perfect matching in a general graph efficiently with your approach? Similarly for the approach you outline with respect to the TSP polytope: if this is a viable approach to improve existing TSP algorithms, why isn't it included in the paper?
>
> There is no obstruction when it comes to applying our method to the matching problem or TSP. However, that is also the case for several other problems. In each case, we would have to provide some writeup (and ideally experiments) if we're going to present it. The paper runs over 40 pages already so we had to draw the line somewhere.  For instance, suppose we decide to talk about TSP.
>
> In that case, we could use the Birkhoff polytope, together with a Sinkhorn algorithm to obtain an interior point. Then we could use a fast matching oracle and results from [1] (for example)  to perform the decomposition. The extension would then optimize over a distribution of tour lengths that is obtained via the distribution of permutations encoded by an interior point of the polytope. Obviously, while we can summarize this here in a few words, it would require a careful writeup in the appendix.
>
> More generally, our approach makes more sense when no exact fast algorithm is available. If a problem is optimally solved by a fast greedy algorithm, then it's harder to justify using our approach.
>
> > I would be very curious to learn more about the polyhedral structure necessary to make this approach applicable.
>
> The main sufficient condition is the existence of a polytime/fast oracle as seen in Theorem 4.1. With that it is possible to set up the algorithm. But as we discussed in the independent set example it is not necessary since one can work around it using a relaxation. How successful the approach is will depend on many implementation details such as the choice of model used (see also the k-cut ablation where the benefit of the model is shown).
>
> We'll respond to the minor comments in a separate comment.
>
> 1. Dufossé, Fanny, et al. "Further notes on Birkhoff–von Neumann decomposition of doubly stochastic matrices." Linear Algebra and its Applications

---

> > ### Comment · Reviewer_gbDc · 2025-08-04
> >
> > My main concern is mathematical preciseness in many instances in the paper, as reflected in the original review. Thm. 4.1 and Algo 1 are just specific instances. I would like to have the paper peer-reviewed in a revised version before I can recommend acceptance.
> >
> > Regarding applicability: yes, you describe a pipeline for TSP, and it is conceivable that it might work. Yet, the examples in the orgininal paper didn't seem overall convincing to me, and that is what I remarked in the original review already.
> >
> > From my side there's no need to respond to the minor comments. It won't make a difference for my evaluation. I simply hope you will take care of them, let it be in a final NeurIPS version (if accepted) or in a future submission.

---

### Official Review · Reviewer_vkDa · 2025-06-25

**Clarity:** 3
**Significance:** 3
**Originality:** 4
**Rating:** 5
**Confidence:** 3

**Summary:**

The paper proposes a theoretically grounded and practically effective framework for solving constrained combinatorial optimization (CO) problems using neural networks. The main contribution is an end-to-end differentiable method that leverages convex geometry and Carathéodory’s theorem to project neural outputs onto the convex hull of feasible discrete solutions. This ensures constraint satisfaction while maintaining differentiability. The method also includes a decomposition algorithm (Algorithm 1) to iteratively represent neural predictions as convex combinations of feasible sets. The authors provide useful theoretical guarantees (Theorems 4.1 and 4.2, Proposition 4.3), empirical evaluation on cardinality-constrained Maximum Coverage problem. Experimental results demonstrate improvements in both inference time and objective values over learning-based baselines, and competitive performance against non-learning methods.

**Questions:**

- What exactly is the MIP-based solution mentioned in line 310?
- What value of $\epsilon$ is used in Algorithm 1 in the experiments? How does performance vary with $\epsilon$, and how was the appropriate $\epsilon$ chosen to balance speed and accuracy?
- In Theorem 4.1 or 4.2, does the value of $\epsilon$ affect the theorem's validity? Can the theorems hold with $\epsilon$ set to zero, or must $\epsilon$ be a certain positive number?

**Ethical Concerns:**

["NO or VERY MINOR ethics concerns only"]

**Final Justification:**

As I mentioned in the response for the authors rebuttal, I decided to maintain my rating.

**Limitations:**

- As mentioned above, the primary limitation of the method is that any change in the target CO problem requires adapting the projection and decomposition procedures specific to the new problem’s polytope.
- While the proposed approach achieves strong results compared to learning-based methods, its performance can be less favorable when compared to non-learning methods or TTO-based baselines. However, as the authors suggest, there is potential to improve this by designing better neural architectures tailored to specific problems.

**Paper Formatting Concerns:**

There is no formatting issues in this paper.

**Quality:**

4

**Strengths And Weaknesses:**

Strengths
---
- The paper presents a well-structured framework based on solid theoretical foundations for handling CO problems with constraints.
- To guarantee that neural network outputs lie on the polytope of feasible solutions, the network predicts perturbations on an n-dimensional hypercube rather than outputting directly. This clever approach resembles the reparameterisation trick in variational methods, which separates random perturbations from learned distribution parameters for gradient-based learning.
- The method demonstrates strong performance compared to learning-based baselines. Inference is fast, and the objective values found by the method often surpass those of other approaches.

Weaknesses
---
- In Algorithm 1, $x_0 \leftarrow x$ should be replaced by $x_1 \leftarrow x$. (I am not 100% sure with the replacement, but Algorithm 1 currently raises UndefinedError since $x_1$ is not defined in Step 3.) Some definitions (e.g., Steps 3–5) are not fully introduced until later in the paper. It would be helpful to provide earlier hints or comments so that readers are not confused when encountering them initially.
- It would be useful to clarify the meaning of TTO (Test-Time Optimization) in Table 1. Table 1 may be more appropriately named as Figure 1. The figures are small to read. In the last graph (12th), the points for Greedy and Gurobi appear to coincide—this overlap should be explicitly noted.
- The method’s most significant limitation is that if the target CO problem changes, the projection onto the polytope and decomposition algorithm must be adapted accordingly. However, the authors alleviate this concern by demonstrating that for a broad class of CO problems—including those with cardinality constraints, partition matroids, spanning trees, and independent sets—the projection and decomposition can be implemented efficiently.

---

> ### Author Rebuttal · Authors · 2025-07-29
>
> We sincerely appreciate the reviewer’s thorough evaluation and constructive feedback. Below, we address the bulleted questions in the order they were presented.
> - Thank you for pointing this out. You are absolutely right—the initialization should indeed be $\mathbf{x}_1\gets\mathbf{x}$, and we have corrected this accordingly. We also appreciate your suggestion regarding clarity: to improve readability, we will move the relevant definitions earlier in Section 4.1 so that Steps 3–5 are easier to follow when first encountered.
>
>  - The details of TTO are given in Lines 346 to 357. The caption of the Table has been updated to include the meaning of TTO (Test-Time Optimization). Table 1 is now renamed as Figure 1. The figures have been enlarged for readability; an enlarged version of the plots was already included in the appendix. In the last graph (12th), the overlap between Greedy and Gurobi is now explicitly noted.
>
>   - The MIP-based solution refers to the exact solution obtained by formulating the problem as a Mixed Integer Program (MIP) and solving it using Gurobi.
>    - A detailed discussion of how $\epsilon$ serves as a tolerance parameter to control the size of the decomposition is provided in Appendix Section C. The main idea is to 'undershoot' each step in the Caratheodory decomposition by *rescaling* the coefficient choice at every iteration in order to control the number of iterations and hence the number of sets that are used in the decomposition. This helped with convergence during training and led to improved results. Please see the discussion in Lines 320 to 329 around *scaling factor* and lines 1103 to 1110 in the Appendix for the choices of scaling factors. The scaling factors influence the number of terms generated in the decomposition—specifically larger scaling factor results in fewer terms in the decomposition. In the experiments $\epsilon$ is set to a very small value (e.g., $10^{-14}$) to account for numerical instability.
>
> - In both Theorem 4.1 (due to Grötschel–Lovász–Schrijver) and our Theorem 4.2, we have $\epsilon=0$. As stated in Theorem 4.2, instantiating Algorithm 1 with Equations (5) and (6) yields an exact equality; that is, $\mathbf{x}=\sum_{t}p_{\mathbf{x}_t}(S_t)\mathbf{1} _{S_t}$.

---

> > ### Comment · Reviewer_vkDa · 2025-08-05
> >
> > I appreciate the authors for their useful rebuttal. My overall impression of the paper remains unchanged after reading both the initial submission and the subsequent reviews and rebuttal. While there is still room for improvement in the presentation, it is clear that the authors have already put significant effort into making the paper accessible. The examples and experimental results help to alleviate concerns regarding the applicability to a broader class of CO problems.

---

### Official Review · Reviewer_6w4h · 2025-07-02

**Clarity:** 4
**Significance:** 3
**Originality:** 4
**Rating:** 5
**Confidence:** 3

**Summary:**

This work aims to improve self-supervised learning to predict good feasible solutions for combinatorial optimization problems. This is done by decomposing predicted continuous solutions as convex combinations of integer solutions. By doing so, we can take the best integer solution from this decomposition, which is guaranteed to have objective value at least as good as the value of the continuous solution. The paper ensures that this decomposition is differentiable, so it can be integrated into the learning process. This decomposition relies on the classical Carathéodory's theorem, which guarantees that we can express any point in a polytope as a convex combination of n+1 vertices (in this case, integer feasible solutions), where n is the dimension of the polytope. Since this decomposition is hard, this requires some strong oracles for this decomposition to be possible, but the paper shows that this can be done effectively in some more tractable cases, in particular cardinality constraints, partition matroids, spanning trees, and a continuous relaxation of independent sets. Experiments are done on the maximum coverage problem, which uses cardinality constraints, and compared with both learning and non-learning baselines.

**Questions:**

1. Request for one more baseline: In my view, learned methods for combinatorial optimization can be very good at producing high-quality solutions, but often to close the gap to optimality, optimization solvers that do a more structured search such as Gurobi can perform better in practice. I would be interested in seeing what you get when providing to Gurobi the solution to the "short" method as an initial solution. Gurobi itself can be viewed as a way to perform test-time optimization, and I would interpret this as a TTO variant of your method rather than a comparison baseline. Would you be able to perform these experiments? This also will help contextualize the improvements that you get from the "medium" and "long" approaches. I would suggest keeping the time limit here low, so that the total time limit (your "short" + Gurobi) is in line with the set of learned approaches. Of course, this experiment would be more meaningful for the larger problems that you cannot solve to optimality with Gurobi.

2. The beginning of Section 3 states that this method applies only to binary problems. It is perfectly fine to focus on binary problems, but could you please clarify what changes would be necessary if you want to apply this to a non-binary integer problem? Can this method be easily adapted, or is there any particular blocker for making this work?

3. Could you add a clearer definition of g in Algorithm 1? I encountered it for the first time when reading Algorithm 1, and it was not clear what it was other than a function that produces the right boundary point.

4. Figure 1 uses min when the rest of the paper uses max. Please make it consistent.

5. Table 1 has a lot of room for improvement in presentation:
    * I believe this should be a Figure, not a Table?
    * Please add what TTO means in the caption, especially since this table appears before it is defined.
    * Please clarify what the marker X means in the caption (average of both inference time and objective?).
    * These plots are difficult to see: especially when printed, it is small and the individual dots are barely visible. I am not sure if there is any way to make it larger given space constraints, but please see if you can find ways to improve this. In particular, avoid using very bright colors such as yellow. Perhaps experiment with variations in marker shapes as well as color to distinguish between methods of the same type. Optionally, consider increasing the transparency of the dots.
    * Please consider keeping both the x and y axis the same for each problem class. It is difficult to compare between them. If you do this, you may want to consider removing the Random (240s) result from the main text, and only keep it in the Appendix (with a note mentioning that). This is a very weak baseline dominated by everything else, and makes the plot more difficult to read by extending the axes.

6. Tables 9-11 need cleaning up. Some times have s, others do not. The standard deviations are inconsistently in different lines. Some numbers are even overlapping. If possible, try to make them all be in the same line (perhaps further split the tables, or take away a digit of precision when it makes sense).

7. I believe the references in the Appendix are not the same as the references in the main paper. For example, I see references going above 80, and I spot checked a few of them and they did not make sense. Please fix the Appendix references.

**Ethical Concerns:**

["NO or VERY MINOR ethics concerns only"]

**Final Justification:**

I continue to recommend acceptance. While I agree with other reviewers that there is room for improvement in presentation, I do not believe those issues to be sufficiently major as long as the authors can fix them for the final version. I also believe that this method has good potential for applicability, even if it requires effort to adapt the method to new problems. I acknowledge that there is room for improvement in discussing and providing supporting evidence for applicability, but the authors agreed to include further discussion on this topic.

**Limitations:**

Limitations are addressed in the paper.

**Paper Formatting Concerns:**

No formatting concerns.

**Quality:**

4

**Strengths And Weaknesses:**

The decomposition idea explored in this paper is very interesting. While the existence of this decomposition is a classical result, it is rare to see a practical way to apply this decomposition as far as I know, since it is generally considered to be too expensive in practice for most types of constraints that appear in NP-hard problems. However, it can be made tractable for simple constraints. This makes sense in this learning context, where, 1, even handling simple constraints can be sometimes challenging, and 2, we can sometimes embed the "hard part" of the problem into the objective function, like it is done in the maximum coverage example, and leave only a simple constraint to consider. Furthermore, this framework also has the nice property that it provides a generic perspective that can be applied to multiple classes of constraints, which makes it easier for this work to be extended. On the other hand, along these same lines, perhaps a drawback of this method is that it may require a lot of effort to make this work effectively for new classes of constraints, but the paper shows positive results for at least maximum coverage.

The overall writing is careful, well-structured, and comprehensive, especially the theoretical and methodological portion. Unfortunately, I did not have time in this review to go through all the theoretical proofs in detail in the appendix, but it looks generally comprehensive and the basic ideas at a high level make sense. The computational portion has some room for improvement, but I believe it can be improved in this review process. The computational results are positive but not remarkably strong, which I believe is fine with given that the methodological ideas introduced in this paper are interesting and can potentially be extended to other scenarios. Overall, the quality of this paper is high. I have a few requests for improvement below, mostly on the computational experiments and on readability, but at its core I believe this paper is very solid.

---

> ### Author Rebuttal · Authors · 2025-07-29
>
> We are thankful for the especially thorough review and their constructive comments. Below we respond to the bulleted questions in order.
>
> - We tested two versions of the “short+Gurobi” approach on the random datasets, one with a time limit near our "medium" method, and for some settings, an "extended time" version, given the time close to our "long" method. When the instance size and the cardinality constraint are small (k = 10 on Random 500), "short+Gurobi" performs slightly better than other methods. For larger instances like Random 500 with k = 50 and Random 1000, our "medium" and "long" approaches show their advantages and achieved similar or better results, indicating that our local improvement technique effectively constrains the solution space and works better even compared to the well-optimized Gurobi Solver.
> In addition, the short+Gurobi method shows much better cost-efficiency compared to the pure Gurobi method, showing the potential of our method to serve as an efficient initial solution for structured solvers, significantly reducing overall computation time.
>
> **Dataset: Random500**
> | Method         | Time (k=10)           | Obj (k=10)            | Time (k=50)           | Obj (k=50)            |
> |----------------|-----------------------|------------------------|------------------------|------------------------|
> | Ours-short     | 0.5786s ± 0.0094s     | 15177.77 ± 344.68      | 1.1078s ± 0.0149s      | 41247.80 ± 965.48      |
> | Ours-medium    | 1.0546s ± 0.0146s     | 15410.23 ± 353.91      | 2.1212s ± 0.0096s      | 43152.87 ± 1004.95     |
> | Ours-long      | 12.0878s ± 0.1365s    | 15714.63 ± 354.22      | 24.9885s ± 0.3543s     | 44866.84 ± 876.78      |
> | Gurobi (120s)  | 1.5744s ± 0.5376s     | 15714.90 ± 346.84      | 120.0651s ± 0.0268s    | 44880.59 ± 7.132       |
> | short+Gurobi   | 2.3185s ± 0.5136s     | 15758.62 ± 343.40      | 2.0622s ± 0.1122s      | 44894.62 ± 951.89      |
> | short+Gurobi (Extended Time)              |        N/A*               |        N/A                | 25.0653s ± 0.0609s     | 45090.41 ± 887.32      |
>
> \* Gurobi finishes search in less than 5 seconds, so extended time version is not applicable
>
> **Dataset: Random1000**
> | Method         | Time (k=20)           | Obj (k=20)            | Time (k=100)          | Obj (k=100)           |
> |----------------|------------------------|------------------------|------------------------|------------------------|
> | Ours-short     | 0.8248s ± 0.0312s      | 29810.12 ± 586.05      | 1.9076s ± 0.0284s      | 81357.29 ± 1277.84     |
> | Ours-medium    | 1.4175s ± 0.0633s      | 30257.05 ± 606.07      | 3.6669s ± 0.0582s      | 84186.79 ± 1336.48     |
> | Ours-long      | 16.6007s ± 0.4632s     | 31230.26 ± 476.25      | 46.1200s ± 0.6998s     | 88796.03 ± 1327.83     |
> | Gurobi (120s)  | 37.2985s ± 30.0300s    | 31347.62 ± 509.75      | 120.1395s ± 0.0526s    | 89696.83 ± 1231.76     |
> | short+Gurobi   | 2.0868s ± 0.1001s      | 30098.64 ± 767.38      | 5.2979s ± 0.1202s      | 82200.65 ± 2609.87     |
> | short+Gurobi (Extended Time)    | 15.8728s ± 2.8134s     | 31332.18 ± 471.59      |  46.2701s ± 0.3331s      | 89509.90±1316.43 |
>
>  - In general, Carathéodory’s theorem applies to any convex hull. However, the specifics of obtaining an almost-everywhere (a.e.) differentiable extension can vary when moving beyond the Boolean domain and may require domain-specific techniques rather than direct generalization. In particular, Steps 3, 5 and 6 of our Algorithm 1 are tailored to the $\{0,1\}$ setting and may require modification in other domains. These steps are problem-specific, and obtaining analogous steps in non-Boolean settings would be a worthwhile contribution on its own. For example, in combinatorial problems over permutations—where we are no longer dealing with $\{0,1\}$ assignments—achieving an a.e. differentiable extension typically relies on the Birkhoff decomposition. This approach requires additional problem-specific algorithmic tools. Recent work by Nerem et al. [1] explores this direction in more depth.
>
> - Thank you for the constructive feedback on the presentation—we sincerely appreciate it. We will incorporate your suggestions in the revised version of the paper.
>      - Clarification on the meaning of each "X" in the plots: In each plot, the x-axis represents runtime (where smaller values are better), and the y-axis represents the objective value (where larger values are better). Each colored cross ("X") corresponds to a method identified in the legend and indicates that method’s performance in terms of both runtime and objective. The relative positions of the crosses illustrate the trade-off between these two metrics. Across the various settings, our methods consistently appear on or near the Pareto front, demonstrating favorable performance in both dimensions.
>    - We will change both the marker shapes and colors to improve readability.
>    - We agree that standardizing the axis ranges can improve visual clarity. However, since we experiment with different values of $k$ (in the cardinality constraint), the optimal objective value naturally increases with larger $k$, which leads to differing y-axis scales across plots. This variation reflects meaningful differences in the problem instances rather than plotting inconsistencies.
> 	    That said, we appreciate the point about the Random (240s) baseline. Given its poor performance and the distortion it introduces in the axis scaling, we will move it to the appendix and add a note in the main text to guide the reader. We also note that larger versions of all plots are already available in the appendix for more detailed comparison.
>     -  We will clean up Tables 9–11 and consider splitting the tables or reducing precision to improve readability.
>     - The reason the reference numbers in the Appendix differ from those in the main paper is that we include additional related work and citations in the Appendix (Section A) that are not referenced in the main text. As a result, the numbering continues beyond what appears in the main paper. There are a few inconsistencies that we have fixed. Thanks for pointing this out.
>
>
> References:
>
> 1. Robert R Nerem, Zhishang Luo, Akbar Rafiey, and Yusu Wang. Differentiable extensions with rounding guarantees for combinatorial optimization over permutations. arXiv preprint arXiv:2411.10707, 2024

---

> > ### Comment · Reviewer_6w4h · 2025-08-03
> > **Response to rebuttal**
> >
> > Thank you for the rebuttal. All my questions have been addressed (item 3 was addressed in a different rebuttal). I appreciate the experiments: I believe this set of experiments is an interesting TTO variant for your method that helps contextualize the medium/long improvement heuristics. I suggest adding these experiments to the final version.
> >
> > Given other discussions, I would also like to comment on the theme of applicability across reviews. While first reading the paper, I did initially feel that the requirement of a tractable optimization oracle seemed too strong. However, when I read the maximum coverage application, I realized that there is still good potential for applicability because you can keep some of the difficulty of the problem in the objective function in this context, and therefore apply this to harder problems where you do not have that oracle. I believe this is crucial for this framework. In retrospect, the paper would have been stronger if it dived deeper in a couple more applications like these, but I do not believe this to be particularly necessary for a meaningful contribution. I recognize that this framework requires work to adapt to applications, but my overall opinion is still that the methodological contribution outweighs these challenges in applicability.

---

> > > ### Author Response · Authors · 2025-08-03
> > >
> > > > All my questions have been addressed (item 3 was addressed in a different rebuttal). I appreciate the experiments: I believe this set of experiments is an interesting TTO variant for your method that helps contextualize the medium/long improvement heuristics. I suggest adding these experiments to the final version.
> > >
> > > > I recognize that this framework requires work to adapt to applications, but my overall opinion is still that the methodological contribution outweighs these challenges in applicability.
> > >
> > > Thank you for the response! We are glad to see that the reviewer appreciates the methodological contribution of the paper.  We will also make sure to add those results in the final version.
> > >
> > > >However, when I read the maximum coverage application, I realized that there is still good potential for applicability because you can keep some of the difficulty of the problem in the objective function in this context.
> > >
> > > This is exactly right. The feasible set in many well known hard combinatorial problems can admit a tractable oracle or if it doesn't, one can shift to a formulation of the problem that admits one. As you point out, there the objective may become more 'difficult' (see also our response to reviewer gbDc for examples). Obviously there are countless variants of hard problems so as the reviewer acknowledges, depending on the specifics, adapting the framework to any given problem is non-trivial and requires careful design decisions. These include what formulation of the problem to use, the projection, and the optimization oracle.
> > >
> > > In retrospect, we should have probably provided some of that discussion in the main paper in order to make this clearer.
> > > Our hope was that the discussion of specific use-cases (like max-coverage or matroid constraints and independent sets) would help with building intuition around this. Space constraints made this a tough balance to strike in our original submission because we wanted to ensure that our submission is accessible to a broader machine learning audience that may not be familiar with the specifics of polyhedral combinatorics and optimization.
> > >
> > >  If the paper is accepted, we will use the additional space to provide more information in the main text about the applicability of the method and we will provide additional discussion in the Appendix. Based on the feedback we received we will also describe function $g$ clearly and earlier in the text.
> > >
> > > Again, thank you for the review and taking the time to respond to our comments.

---

### Official Review · Reviewer_7pF1 · 2025-07-07

**Clarity:** 2
**Significance:** 3
**Originality:** 3
**Rating:** 4
**Confidence:** 3

**Summary:**

The paper introduces an end-to-end differentiable framework for combinatorial optimization with discrete constraints. The paper leverages geometric insights to decompose neural network predictions into convex combination of feasible sets.  During training, it learns to output a point within the convex hull of feasible solutions. The loss is computed by decomposing the point and evaluating the objective. During test time, the prediction is decomposed and the best feasible solution is selected. In experiment, it achieves good results over machine learning baselines and classical algorithms.

**Questions:**

1. Could you comment on how well this approach scale to larger instances? How computationally heavy is differentiating through the decomposition steps?
2. Can you provide a summary of the results for other problem domains in appendix?
3. Why did you choose to present ablation study results on max-cut which is different from the coverage problem?

**Ethical Concerns:**

["NO or VERY MINOR ethics concerns only"]

**Final Justification:**

I have read the authors' rebuttal. Most concerns are resolved and want to maintain my score.

**Limitations:**

See weakness 1.

**Paper Formatting Concerns:**

No major issues. Some tables in appendix need formatting revision.

**Quality:**

3

**Strengths And Weaknesses:**

Strengths:

1. The paper presents a novel idea of decomposing an interior point of the convex hull of solutions and develops an end-to-end framework for solving combinatorial optimization based on it. The claims are supported by both theoretical guarantees and empirical evidence.
2. Overall, the presentation of the paper is easy to follow.


Weaknesses:

1. This method could only work well for problems where feasible solutions are easy to obtain. For some problems where feasible solutions are difficult to find, I can imagine that this method will struggle.
2. Overall, the results look good for the coverage problem. But the results in appendix for other problems don’t seem to be impressive (also it is kind of hard to read, maybe I misinterpreted. See weakness 3).
3. The experimental results in Table 1 are difficult to read, especially  for the last row where there is a cluster of crosses where same of them with similar colors are close to each other. It would be better to have a numerical representation of those figures. In addition, the result tables in appendix are how to read as well. For example, Table 11 is not well-formatted where some columns overlap with each other. Can you also highlight the best rows in bold and provide a text summary of the results.

---

> ### Author Rebuttal · Authors · 2025-07-29
>
> We would like to thank the reviewer for their thoughtful and constructive feedback on our work.
>
> - The reviewer makes a point about the applicability of the method to problems where feasible solutions are difficult to obtain. Indeed, not being able to efficiently construct feasible solutions poses a challenge and our proposed method works best when obtaining feasible solutions is computationally tractable. This allows us to construct a decomposition by fetching feasible sets at each iteration. On the other hand, there are plausible ways in which this approach can be naturally extended to the setting where feasible solutions are hard to construct. For instance, one could define a polytope of 'efficiently computable infeasible solutions' and optimize over them by scoring the amount of constraint violation. This could be used as warm-starts that are passed to a discrete search algorithm that then attempts to construct a feasible solution. This is an interesting future direction but it introduces an additional layer of complexity in terms of benchmarking and implementation so we decided to prioritize simpler problems.
>
> The reviewer also asked for a summary of the results in the appendix, particularly for problems other than max coverage.
>
> - To summarize the results in the appendix, our method achieves strong empirical performance on the max coverage problem and is competitive with the greedy approach on the max k-cut problem. The appendix also includes several ablation studies that demonstrate the effectiveness of our medium and long methods, as well as the quality of candidate sets returned by the trained model. Additionally, we highlight our focus on the cardinality constraint and discuss the shortcomings of the best-known neural baseline.
>
> - Below, we expand on these points.
>
>
>   - While the appendix presents extensive experiments on the max coverage problem—using various cardinality constraints and datasets—as well as results on the max k-cut problem, we are also in the process of obtaining additional results for other problems. Below we provide preliminary results for another fundamental combinatorial problem: the Facility Location problem. Here we present the performance of our short method (with no Test-Time Optimization) against the Greedy, an RL baseline, and the CardNN approach without Test-Time Optimization (TTO). We consider two cardinality constraints cases that are k=5 and k=20 and report the objective function values. Note that Facility Location is a minimization problem and smaller objective values are better. The experiments are run on synthetic data where each instance contains 500 points.
>
>   | Method | k=5 | k=20 |
>   |-------|-------|-------|
>   |Ours short  |	16.1775	| 4.1373 |
>   | CardNN-S-NoTTO-NoKmeans |	86.2816	| 17.6436 |
>   |Greedy |	17.3714	| 2.9644 |
>   |RL	| 35.9455	| 6.8085 |
>
>    - In Appendix G.3, we run an ablation study to demonstrate that in the medium and long versions of our approach, the strategic selection of nodes in our candidate pool provides meaningful benefits. The results in Table 2 demonstrate that our strategic candidate selection provides consistent improvements over random selection, validating that our local improvement procedure benefits specifically from the candidate pool construction via our decomposition, rather than simply from having access to additional nodes for local improvement.
>
>    - Note that we heavily emphasized cardinality constraints in the experiments because they contain practically important problems like max-coverage. Despite the simplicity of the constraint, we found that existing solutions in the neural combinatorial optimization literature did not provide sufficiently simple and/or effective ways of tackling the constraint while achieving strong results. The ablation study we performed in page 31 (Appendix G.4)  highlights this issue with the best known neural baseline. Through our projection-decomposition-extension scheme, we provide an intuitive and empirically compelling approach to solving the problem in a differentiable way, and our experimental results support that.
>
> > It would be better to have a numerical representation of those figures. In addition, the result tables in appendix are how to read as well.
>
> - The reviewer also raises a valid point about the presentation of the results which we completely acknowledge. We conducted a comprehensive set of experiments on our method but also on the competing baselines to ensure a fair evaluation. There are several components to those comparisons. Different datasets, different parameters for the constraint, as well as the difference between baselines that involve learning and those that don't. While preparing the paper, it was not clear to us what is the best way to present these results, because the full table (as can be seen in the appendix G.5) is also quite dense and hard to read. We are going to do our best to reformat the results to highlight some of the key takeaways according to the reviewer's comment, but we welcome any further recommendations on how we could present the results.
>
> - We will change both the marker shapes and colors to improve readability. We will clean up Tables 9–11 and consider splitting the tables or reducing precision to improve readability. We also note Appendix G.5 contains larger versions of all plots for more detailed comparison as well as raw numbers for experiments in tables 9 to 11.
>
> >Could you comment on how well this approach scale to larger instances? How computationally heavy is differentiating through the decomposition steps?
>
> - Regarding computational efficiency, the detailed cost depends on the problem and the cost of the decomposition, which in turn depends on the algorithm used to obtain feasible sets and the ability to compute intersections with the boundary of the polytope. Given a point in the polytope, the decomposition applies iteratively a transformation to the point until the termination criterion is reached. For the examples we discussed, that is roughly $O(n)$ (in the worst case) iterations given by Equation (3). For the backprop step, the gradients are passing through the coefficients of the decomposition, where we have one coefficient per iterate. Therefore, the cost of gradient computation for each coefficient depends linearly on the number of iterations it took to compute the coefficient because the chain rule for a given coefficient will have to go through all the iterates before it. Crucially, the most costly part is actually the decomposition itself and the backward pass does not increase the overall time significantly.
> As a general rule, the closer the neural net prediction is to a feasible solution, the fewer the decomposition steps and hence the method becomes more efficient. How close the predictions are to feasible depends on practical implementation, how the architecture aligns with the problem, etc.
> We were able to run our experiments containing instances with several thousands of nodes on a single 48GB GPU.  To make the point about the cost of backprop clearer, we provide some time measurements in the coverage problem.
>
> | Setting | Back-prop time  |
> |----------|----------|
> | 1500 nodes, k = 10    | 17.08s    |
> | 1500 nodes, k = 50    | 10.42s    |
> | 3000 nodes, k = 20    | 12.54s   |
> | 3000 nodes, k = 100    | 10.49s   |
> | 6000 nodes, k = 100    | 16.32s   |
> | 6000 nodes, k = 300    | 9.98s    |
>
> - For a given instance size, the decomposition will run at most $n-k$ times which explains why for smaller values of $k$ the cost is larger. Overall, we see that the cost of backprop is not significantly increased as we scale up the instances.
>
> > Why did you choose to present ablation study results on max-cut which is different from the coverage problem?
>
> - Finally, the reason for choosing max k-cut for the ablation was to highlight another problem where our approach can work. It is also worth noting that the same ablation shows that our decomposition-based extension also works with direct optimization on the input space (i.e., without a neural network). Even though the constructed function is non-convex, one can just directly run Adam on a random starting point in the hypercube to find a solution for the problem. If the paper is accepted, we are planning to include even more results along those lines to further bolster our claims.

---

> > ### Comment · Reviewer_7pF1 · 2025-08-06
> >
> > Thank you for the rebuttal. I have read the rebuttal and decided to maintain my score.

---

### Decision · Program_Chairs · 2025-09-17

**Decision:**

Accept (poster)

**Comment:**

The paper proposes a self-supervised method for predicting feasible points of combinatorial optimization problems by a decomposition approach. Reviewers note the novelty of the idea, good presentation. Reviewers have mixed opinions on the quality of the experimental results, ranging from ok but not impressive to strong. There is some excitement about the broad applicability and potential of the idea. The main negative issue is the writing. We ask the authors to incorporate the feedback from the review process.
All in all, the paper merits acceptance at NeurIPS.
The extensive rebuttal is positively acknowledged.